# Thermal structure of the southern Caribbean and NW South America: implications for seismogenesis

**Ángela María Gómez-García[1,2,+], Álvaro González[1,3], Mauro Cacace[1], Magdalena Scheck-Wenderoth[1], Gaspar Monsalve[4]**

[1] GFZ German Research Centre for Geosciences. Telegrafenberg, 14473, Potsdam, Germany.

[2] Corporation Center of Excellence in Marine Sciences (CEMarin). Bogotá, Colombia.

[3] Centre de Recerca Matemàtica (CRM). Campus UAB, Edifici C. 08193, Bellaterra (Barcelona), Spain.

[4] Universidad Nacional de Colombia, Facultad de Minas, Medellín, Colombia.

Corresponding author: Ángela María Gómez-García (angela@gfz-potsdam.de). [+]Now at Geosciences Barcelona (GEO3BCN), CSIC, Lluís Solé i Sabarís s/n. 08028, Barcelona, Spain

**Abstract.** The seismogenesis of rocks is mainly affected by their mineral composition and in-situ conditions (temperature and state of stress). Diverse laboratory experiments have explored the frictional behavior of the rocks and rock-forming minerals most common in the crust and uppermost mantle. However, it is debated how to "up-scale" these results to the lithosphere. In particular, most earthquakes in the crust nucleate down to the crustal seismogenic depth (CSD), which is a proxy to the maximum depth of crustal earthquake ruptures in seismic hazard assessments. In this study we propose a workflow to up-scale and validate those laboratory experiments to natural geological conditions relevant for crustal and upper mantle rocks. We used the southern Caribbean and NW South America as a case study to explore the three-dimensional spatial variation of the CSD (mapped as D90, the 90% percentile of hypocentral depths) and the temperatures at which crustal earthquakes likely occur. A 3D steady-state thermal field was computed for the region, using a finite element scheme using the software GOLEM, considering the uppermost 75 km of a previously published 3D data-integrative lithospheric configuration, lithology-constrained thermal parameters, and appropriate upper and lower boundary conditions. The model was validated using additional, independent measurements of downhole temperatures and heat flow. We found that the majority of crustal earthquakes nucleate at temperatures less than 350°C, in agreement with frictional experiments of typical crustal rocks. A few outliers with larger hypocentral temperatures evidence nucleation conditions consistent with the seismogenic window of olivine-rich rocks, and can be due either to uncertainties in the Moho depths and/or in the earthquake hypocenters, or to the presence of ultramafic rocks within different crustal blocks and allochthonous terranes accreted to this complex margin. Moreover, the spatial distribution of crustal seismicity in the region correlates with the geothermal gradient, with no crustal earthquakes occurring in domains with low thermal gradient. Finally, we find that the largest earthquake recorded in the region ($M_w$=7.1, Murindó sequence, in 1992) nucleated close to the CSD, highlighting the importance of considering this lower stability transition for seismogenesis when characterizing the depth of seismogenic sources in hazard assessments. The approach presented in this study goes beyond a statistical approach in that the local heterogeneity of physical properties is considered in our simulations and additionally validated by the observed depth distribution of earthquakes. The coherence of the calculated hypocentral temperatures with those expected from laboratory measurements provides additional support to our modelling workflow. This approach can be applied to other tectonic settings worldwide, and it could be further refined as new, high-quality hypocentral locations and heat flow and temperature observations become available.

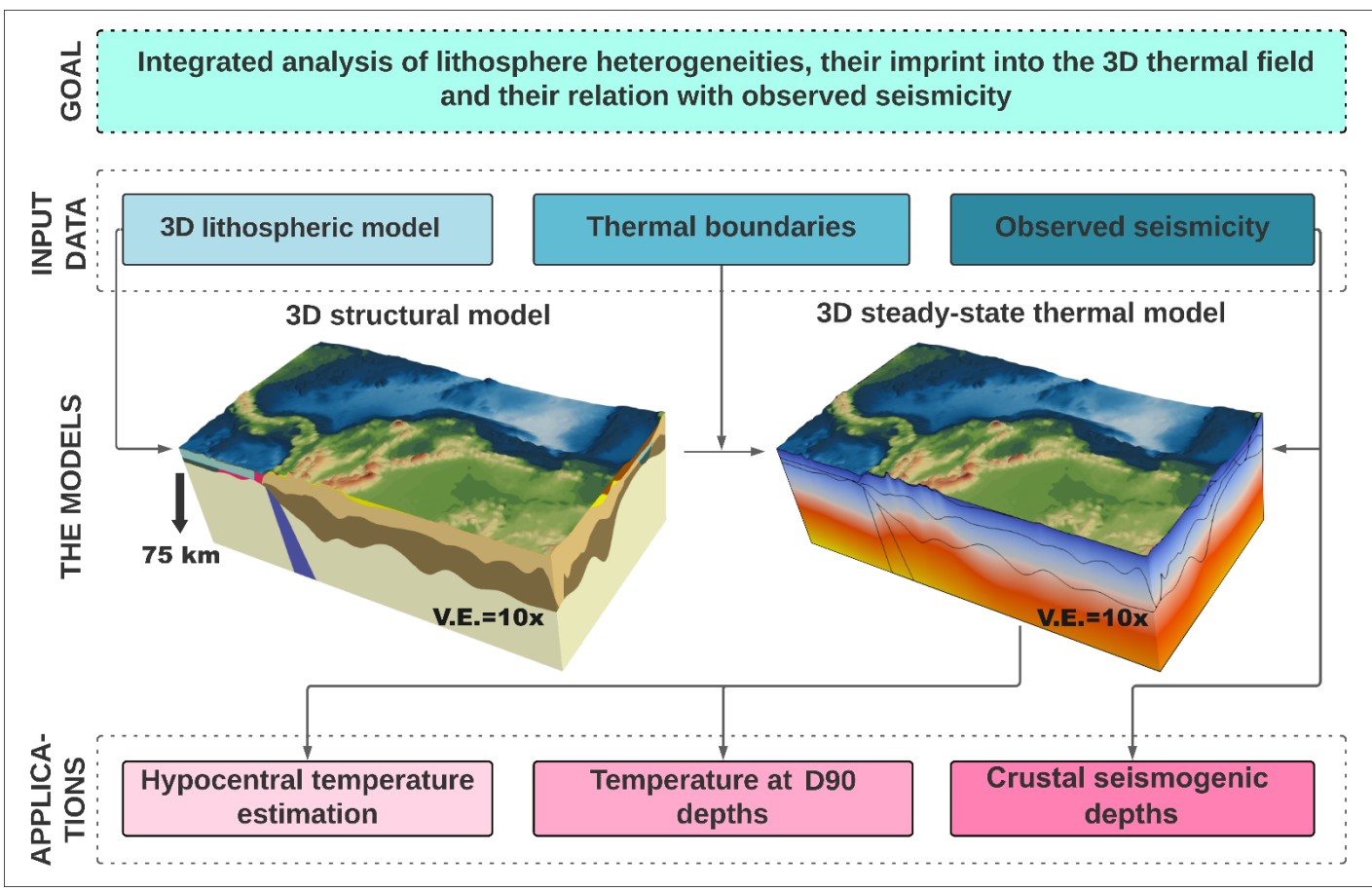

# 1 Introduction

The spatial distribution of seismicity is controlled by the mechanical properties of the hosting rocks, and therefore by factors such as mineral composition and grain size, as well as by the in-situ temperature, pressure and strain rate conditions (e.g. Chen et al., 2013; Zielke et al., 2020). Laboratory experiments indicate a range of limiting temperatures for seismogenesis. Granitic rocks exhibit seismic behavior at temperatures between 90-350°C, gabbro between 200-600°C, and olivine gouge between 600-1000°C (King and Marone, 2012; Scholz, 2019). These ranges, however, are only a proxy to natural conditions, where rocks are more heterogeneous than the laboratory samples and may have a more complex behavior. For example, mixtures of 65% illite and 35% quartz might exhibit a seismogenic window between 250 and 400°C, while replacing the illite with muscovite implies a new window between 350 and 500°C (Scholz, 2019).

The upper temperature threshold for seismogenesis in mantle-forming minerals is highly debated. Some authors defined a rather strict limit 600°C (e.g. Craig et al., 2012; McKenzie et al., 2005), but new evidence suggests higher values. For example, Ueda et al. (2020) found that the brittle-to-ductile transition in peridotite occurs at ~720 °C, based on thermobarometry of equilibrium mineral assemblages in fault-related deformed rocks (pseudotachylytes, cataclasites, and mylonites). Similarly, Grose and Afonso (2013) studied the evolution of the oceanic lithosphere using more realistic thermal models than those assumed by McKenzie et al. (2005) (i.e. including the effects of hydrothermal circulation, oceanic crust, and temperature-pressure-dependent thermal properties, as well as mineral physics), and found a brittle-ductile transition closer to the 700-800°C isotherms, depending on the estimated mantle temperature. So, apart from subduction zones, it is generally considered that earthquakes nucleate within the crust at T< 350±50°C, and at T< 700±100°C in the mantle (see review by Chen et al., 2013).

As a result of these physical constrains, seismicity is typically distributed within the crust down to a maximum depth (Marone and Scholz, 1988; Marone and Saffer, 2015; Wu et al., 2017; Scholz, 2019). Such a lower limit, the crustal seismogenic depth (CSD) is usually quantified as the depth above which a large percentage (such as 90% or 95%) of the crustal hypocenters have been recorded (Omuralieva et al., 2012; Sibson, 1982; Tanaka, 2004; Wu et al., 2017). This estimate has also been discussed to provide a conservative, minimum bound to the depth to the brittle-ductile transition (Zuza and Cao, 2020). In converging margins, such percentiles may be calculated from crustal seismicity alone, in order to avoid mixing different statistical earthquake depth distributions arising from the subduction interface or the underlying slab with that of the upper plate (Tanaka, 2004; Wu et al., 2017).

In an attempt to scale up the results of laboratory experiments, previous studies tried to model the thermal field of active systems and to determine the temperature ranges at which earthquakes likely nucleate (Gutscher et al., 2016; Oleskevich et al., 1999; Zuza and Cao, 2020). The results from these efforts indicate that in intracontinental faults, the brittle-ductile transition seems to be controlled by variations in the geothermal gradient (Zuza and Cao, 2020). Other works (Omuralieva et al., 2012; Tanaka, 2004), support that high heat flow correlates with a shallow CSD, especially for regions with high geothermal gradient (e.g. > 100°C/km). However, in regions where the geothermal gradient is not particularly high, the relationship between the CSD and the thermal state of the crust is not clear (see Fig. 1 in Tanaka, 2004). As a major limitation, most of these approaches consider a simplified lithospheric structure, disregarding in particular tectonic assemblages that can considerably affect the three-dimensional configuration of the thermal field, such as the heterogeneous geometries and properties of the lithospheric layers and the thermal blanketing effect of the sediments (e.g. Cacace and Scheck-Wenderoth, 2016).

This paper focuses on the thermal structure of southern Caribbean and NW South America (Fig. 1a) and its implication for seismogenesis, using a 3D data-integrative numerical modelling approach, and relying on an earthquake catalogue which selects the best-located earthquakes reported in global databases since 1980. We develop a thermal model for the lithosphere down to 75 km depth, to calculate the temperatures at which crustal earthquakes likely nucleate. Also, we systematically map the spatially variable CSD, and its temperature according to the model. We do not attempt to account for transient effects in the seismogenic zone configuration, but rather focus on its regional variations as averaged over time.

The complex tectonic setting of the study area poses a challenge to model a realistic thermal field that allows scaling up the predicted conditions of seismogenesis from laboratory experiments to the lithosphere. This includes the convergence of at least four tectonic plates, several tectonic blocks, the accretion of allochthonous terranes, and the presence of continental basins with sediment thicknesses up to 8 km (Mora-Bohórquez et al., 2020) (Fig. 1b).

As the CSD is influenced by factors that vary in space, such as lithology and temperature (Hirth and Beeler, 2015; Zielke et al., 2020), we computed the 3D steady-state thermal field using a recently published 3D structural and density model of the study area, which is consistent with different geological and geophysical observations, including gravity (Gómez-García et al., 2020, 2021). A steady-state approach can be regarded as appropriate for this analysis since: 1) we preferentially target crustal earthquakes. 2) The subducting segments of the Nazca and Caribbean slabs in the study area are flat (Gómez-García et al., 2021; Kellogg et al., 2019; Sun et al., 2022), implying that the subduction velocities might be lower than in steep slab segments (Currie and Copeland, 2022; Schellart and Strak, 2021). So, the transient effects of dynamic changes and mantle wedge cooling due to subduction occur on much longer timescales than those of the heat transport in the upper lithosphere and of the earthquake cycle. And 3) we are already considering the mantle imprint on the temperature field at 75 km depth as a lower boundary condition. The novelty of our study is to consider how spatial heterogeneity in the lithology of the lithosphere and mantle temperature influences the temperature distribution and seismicity within the crust.

Few earthquakes with magnitude $M > 7.0$ have been recorded in northern South America since the deployment of modern seismological networks, but there are historical records of earlier ones, for example, the shock which destroyed the city of Santa Marta, Colombia, in 1834. Similarly, paleoseismological studies in western Venezuela indentified the fault rupture of other events with estimated magnitudes $M > 7.0$ (e.g. Audemard, 1996; Pousse-Beltran et al., 2018). Overall, there is a substantial seismic hazard in the region (Johnson et al., 2023; Arcila et al., 2020), with highly populated centers located close to shallow active faults, which can generate devastating earthquakes (Veloza et al., 2012). A better understanding of the regional seismogenesis can significantly contribute to improve future seismic hazard and risk assessments. In particular, the CSD is a proxy to the maximum depth of seismic ruptures in crustal faults (e.g. Ellis et al., 2023; Zeng et al., 2022), which in turn may limit the rupture areas and the maximum magnitudes of the earthquakes that these faults may host.

**2 Study area**

The study area (5°-15°N and 63°-82°W, Fig. 1a) encompasses a domain where the Caribbean and Nazca (Coiba) flat slabs interact at depth (Gómez-García et al., 2021; Kellogg et al., 2019; Sun et al., 2022). This interaction results in a complex tectonic setting at the scale of the whole lithosphere, and in large uncertainties in the estimated depths of the Moho interface (Avellaneda-Jiménez et al., 2022; Poveda et al., 2015; Reguzzoni and Sampietro, 2015). The present-day flat slab geometry has been established since about 6 Ma, when the Nazca tear developed separating the north (flat) and south (steep) segments. As a result, the volcanic activity has ceased in the continental crust of the overriding plate of the north segment, which spatially

corresponds to our study area (Wagner et al., 2017). This allows us to consider that the propagation of heat within the crust is mainly driven by conduction (e.g. Liu et al., 2021).

Figure 1a depicts the selected crustal earthquakes in the region (details in Sect. 3.2.1) and active fault traces. For the sake of clarity, in the remainder of the study we will focus on three specific sub-regions, marked by blue boxes in Fig. 1a. Our choice stems from the fact that these regions have contrasting tectonic environments, a heterogeneous spatial distribution of crustal seismicity and diverse allochthonous terranes accreted to the NW margin of South America (Fig. 1b). Such terranes resulted from the migration of the Caribbean Large Igneous Plateau (C-LIP) from the Pacific towards the present-day Caribbean plate location. The collision of the C-LIP with the continental margin of South America defined not only a broad sheared margin

(with remnants of continental slivers and ophiolitic sutures, Kennan and Pindell, 2009), but also extended fragments of mafic and ultramafic rocks associated to mantle-plume processes, and emplaced oceanic crust and remnants of island arcs (see Boschman et al., 2014; Kennan and Pindell, 2009; Montes et al., 2019). As a consequence, large-scale sutures (faults) act as major boundaries between these terranes (Kennan and Pindell, 2009), so they have to be addressed, as they may potentially limit domains with different thermal and/or seismogenic behavior.

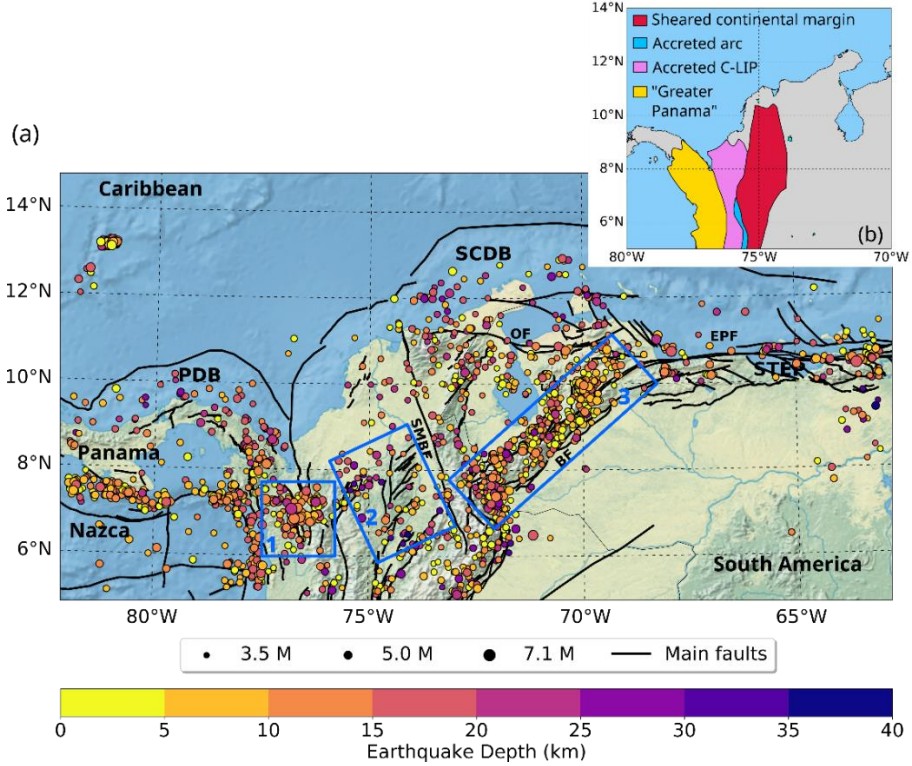

**Figure 1.** (a) Crustal earthquakes and active faults in the study area. Only the earthquakes with the best determined hypocentral depths in the region are represented (Sect. 3.2.1). Blue boxes: Sub-regions discussed in the main text. Black lines: Active fault

traces as compiled by (Styron et al., 2020; Veloza et al., 2012). PDB = Panama deformed belt, SCDB = South Caribbean Deformed Belt and STEP = Subduction-Transform-Edge-Propagator fault system. Main fault systems are: BF = Boconó Fault, SMBF = Santa Marta - Bucaramanga Fault, EPF = El Pilar Fault, and OF = Oca-Ancón Fault. (b) Simplified map of accreted terranes in NW South America (after Boschman et al., 2014; Kennan and Pindell, 2009).

Region 1 corresponds to the area around the Murindó seismic cluster (Dionicio and Sánchez, 2012). In this region, the Uramita fault system (UF, Fig. 3) acts as the suture between the (mainly) oceanic terranes of the western Cordillera, and the "Greater Panama" block (Fig. 1b), also called Panamá-Chocó block, dominated by oceanic plateau and magmatic arc terranes (Montes et al., 2019; Mosquera-Machado et al., 2009). Diverse active faults have been described in this area, including the Atrato, Mutatá and Murindó systems (MF, Fig. 3). The latter has been considered responsible for the $M_w = 6.6$ foreshock and $M_w = 7.1$ mainshock events, that occurred on 17th and 18th October 1992, respectively (Mosquera-Machado et al., 2009), the largest earthquakes recorded in the study region since the 1980s. The mainshock caused widespread liquefaction, landslides, complete destruction of the center of Murindó town and even building damages in Medellín, a city located more than 130 km away from the epicenter (Mosquera-Machado et al., 2009). In terms of recorded seismicity, this region is characterized by a dense occurrence of earthquakes at depths shallower than 25 km.

Region 2 includes the Otú, Palestina and El Espíritu Santo fault systems (Paris et al., 2000). The Palestina fault is a NE-SW strike-slip, right-lateral system that cuts the Central Cordillera, and its formation may have been associated to the oblique subduction of the oceanic lithosphere during the Late Cretaceous (Acosta et al., 2007). This system can be interpreted as the northward continuation of a large-scale brittle suture of different terranes (Kennan and Pindell, 2009). Hereafter, we will refer to the Palestina and Otú-Pericos faults altogether as the Otú-Palestina fault system (OPF, Fig. 3), even though those two structures might be genetically different (Restrepo and Toussaint, 1988). The right-lateral Espíritu Santo fault (ES, Fig. 3) can be considered as a part of the large-scale suture zone defined by the Romeral Fault System (RFS, Fig. 3, Noriega-Londoño et al., 2020). This region concentrates most of the deepest seismic events of the study area.

Region 3 comprises the Venezuelan Andes including the NE-SW Boconó fault system (BF, Fig. 1a). This active fault network accommodates most of the displacement of the Maracaibo block (delimited by the OF, BF and SMBF fault systems, Fig. 1) with a right-lateral strike-slip motion, and serves as its boundary with South America (Pousse-Beltran et al., 2018 and references therein). The seismicity is deeper in the SW portion of the fault system and shows a smooth transition where it shallows towards the NE.

## 3 Methods

### 3.1 Steady-state 3D thermal model and input data

The main mechanism of heat transport within the lithosphere is thermal conduction. Considering the short temporal scales of the seismic events compared to the scales at which the thermal field evolves in the crystalline crust, a first-order calculation can be obtained by a steady-state approach (Turcotte and Schubert, 2014), described by the following equation:

$$H = \nabla(\lambda_b \nabla T) \qquad \text{Eq. (1)}$$

where $H$ is the radiogenic heat production, $\nabla$ is the nabla operator, and $\lambda_b$ the bulk thermal conductivity. The steady-state 3D thermal field is here computed using a numerical model based on the finite-element method with the software GOLEM (see details in Cacace and Jacquey, 2017). We used the uppermost 75 km of an available 3D data-constrained structural and density model (Gómez-García et al., 2020, 2021) (Fig. 2a) as the main input, where dominant lithologies were assigned to individual layers. In the computed thermal field (Fig. 2b), the heat transport within the lithosphere depends on the temperatures used as boundary conditions (Fig. 2c and 2d) and on the thermal properties of each lithospheric layer ($H$ and $\lambda_b$), the values of which have been assigned based on the main lithology as explained in more detail later in the text. The thermal model is published as a separate database (Gómez-García et al., 2023).

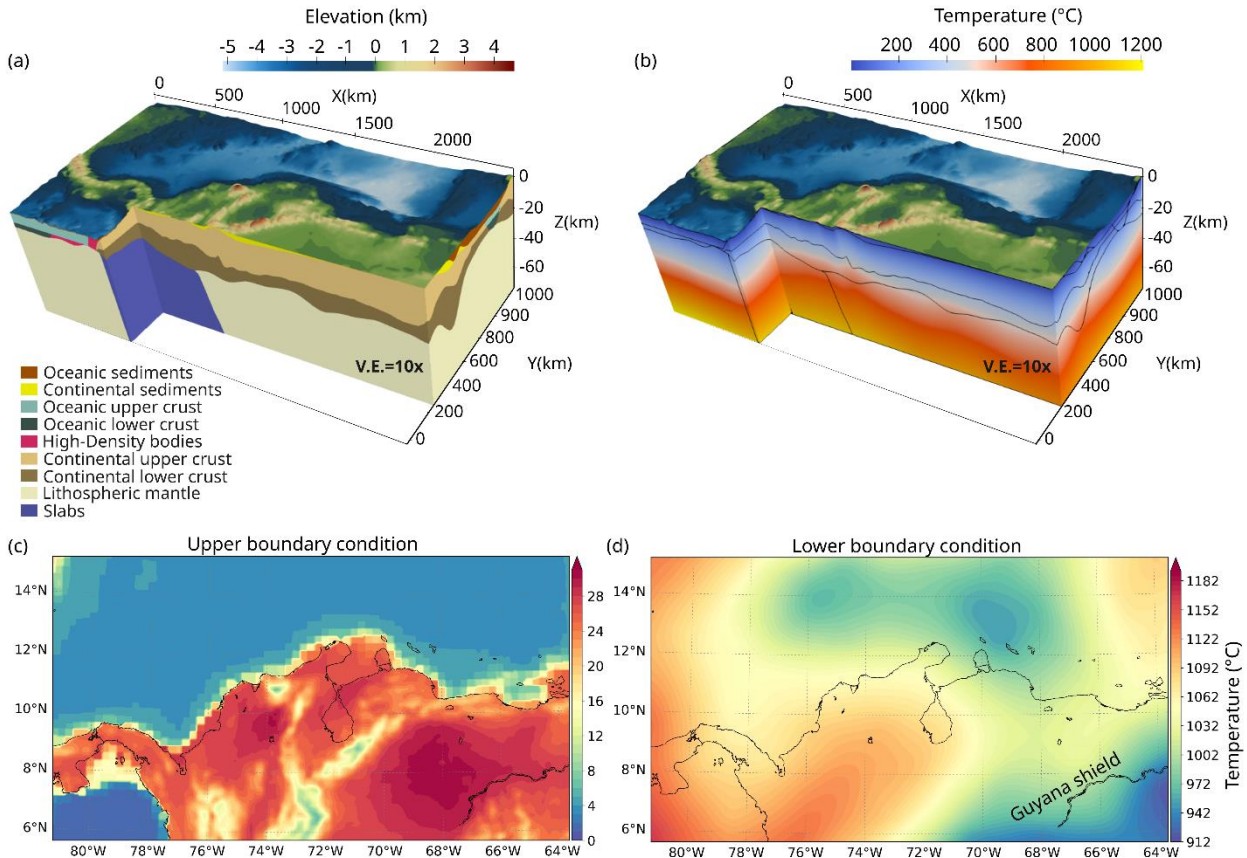

**Figure 2.** 3D thermal model with its structural layers and boundary conditions. The thermal calculation was based on a 3D data-integrative model of the study area, which includes the thermal signature of the heterogeneities from the lithospheric mantle (75 km depth) to the surface. (a) 3D structural model (Gómez-García et al., 2020, 2021) used to compute the 3D thermal field. The model includes fifteen different layers, although the figure only depicts those visible in the 3D view. Lithology-constrained thermal parameters were assigned to each of them (Table 1). (b) 3D steady-state thermal field, with the boundaries

between layers of the structural model depicted as black thin lines. Both the structural and thermal models are shown with a 10X vertical exaggeration. (c) Upper boundary condition, which integrates the temperatures over the continent from the ERA5-Land dataset (Copernicus Climate Change Service, 2019) with those on the seabed from the GLORYS dataset (Ferry et al., 2010). (d) Lower boundary condition, set as the temperatures at 75 km depth, after converting S-wave velocities into temperatures (details in Sect. 3.1.2).

### 3.1.1 Lithospheric structural model and definition of thermal properties

The data-integrative and gravity-constrained structural and density model of the South Caribbean margin (as detailed in Gómez-García et al., 2020, 2021) (Fig. 2a) represents the first-order geological complexity of the Caribbean realms by including fifteen different layers (Table 1). In order to achieve a detailed spatial resolution for the thermal calculations, the structural model has been refined to a 5 km x 5 km horizontal cell size.

The density of each layer (as constrained by Gómez-García et al. (2021)) helps inferring its main lithology, which in turn, allows defining its own values of thermal properties, namely the thermal conductivity and radiogenic heat production (e.g. Ehlers, 2005; Hasterok et al., 2018; Vilà et al., 2010), as detailed in Text S1. Text S2 presents a sensitivity analysis in which we explored the response of 25 different models which considered a range of feasible variations in the thermal properties. The model fitting approach followed, for simplicity, a local optimization in which the initial average values of some
thermal properties were tuned only if necessary, in order to reproduce with minimum misfit the independent measurements of temperatures in boreholes (as discussed in Sect 4.1).

Table 1 lists the lithologies inferred for each layer, compatible with derived densities and with the geologic and tectonic setting of the southern Caribbean region and the northern Andes, the final thermal properties used for the modelling (best fitting model), and the rationale behind each choice.

### 3.1.2 Upper and lower boundary conditions

The thermal upper boundary condition (Fig. 2c) was defined as the temperature field on the solid Earth surface, obtained by integrating the average onshore surface temperatures from the ERA5-Land dataset, from January 2015 to April 2019 (Copernicus Climate Change Service, 2019) and the average temperatures at the seafloor from GLORYS reanalysis (Ferry et al., 2010) for the year 2015. In the modelled domain, the integrated temperature field ranges from ~1°C on the
seafloor of the Pacific Ocean to a maximum of ~30°C over the Venezuelan territory. As expected, the temperatures over the mountains are the lowest within the continental realm, with an average of ~8°C.

The lower boundary condition was defined as the temperature field at 75 km depth (Fig. 2d). It was calculated from a conversion of the S-wave velocities from the SL2013sv tomographic model (Schaeffer and Lebedev, 2013) to temperatures, following the approaches of Goes et al. (2000) and Meeßen (2017), and adopting the reference composition listed in Table S3.
This thermal boundary depicts two cold domains: the Guyana shield, with minima ~912°C, and within the Caribbean region, with a mean value of ~972°C. In contrast, in the region of the Nazca and Caribbean flat slabs, the temperatures are higher than the surroundings, reaching up to ~1100°C. All lateral borders of the model are assumed to be closed.

**Table 1.** Thermal properties defined for each lithospheric layer and their densities after Gómez-García et al. (2020, 2021). $\lambda_b$: Bulk thermal conductivity. $H$: Radiogenic heat production. C-LIP: Caribbean Large Igneous Plateau. See details in Texts S1 and S2. Reference abbreviations: [a]Turcotte and Schubert (2014). [b]Vilà et al. (2010). [c]Neill et al. (2011). [d]Kerr (2014). [e]Montes et al. (2019).

| Layer | Density (kg m⁻³) | $\lambda_b$ (W m⁻¹ K⁻¹) | $H$ (µW m⁻³) | Rationale for $\lambda_b$ | Reference for $H$ |
|---|---|---|---|---|---|
| Oceanic sediments | 2350 | 2.55 | 1.1 | Average between sandstone, limestone and shale[a] | Mean value for sedimentary rocks[b] |
| Continental sediments | 2500 | 3.5 | 1.19 | Assuming sandstones[a] | Mean value for detritic sedimentary rocks[b] |
| Oceanic upper crust | 3000 | 2.1 | 0.358 | Mean value for basalts[a] | Mean value for basalts[b] |
| Low density bodies in the upper oceanic crust (Aves Ridge) | 2900 | 2.6 | 1.07 | Average for basalts and granites[a] following the composition by[c] | Eq. S1, using the average concentration of U, Th and K for Aves Ridge samples[c] |
| High density bodies in the upper oceanic crust | 3250 | 2.93 | 0.057 | Average for basalts, gabbros and peridotites[a] assuming a C-LIP mixed composition | Eq. S1, using the average concentration of U, Th and K for C-LIP samples[d] |
| Oceanic lower crust | 3100 | 2.95 | 0.468 | Mean value for gabbros[a] | Mean value for gabbros[b] |
| Low density bodies in the lower oceanic crust (Aves Ridge) | 3000 | 2.6 | 1.07 | Average for basalts and granites[a] following the composition by[c] | Eq. S1, using the average concentration of U, Th and K for Aves Ridge samples[c] |
| High density bodies in the lower oceanic crust | 3250 | 2.93 | 0.057 | Average for basalts, gabbros and peridotites[a] assuming a C-LIP mixed composition | Eq. S1, using the average concentration of U, Th and K for C-LIP samples[d] |

| | | | | | |
|---|---|---|---|---|---|
| Continental upper crust | 2750 | 2.4 | 0.6 | Assuming a granitic composition[a] | Assuming a granitic composition[b] |
| Low density bodies in the upper continental crust | 2600 - 2650 | 2.1 | 0.4 | Assuming a basaltic composition[a] | Assuming a basaltic composition[b] |
| High density body in the upper continental crust (Santa Marta massif) | 3000 | 2.95 | 0.667 | Mean value for gabbros[a] assuming a magmatic composition[e] | Assuming a gabroic composition[b] |
| Continental lower crust | 3070 | 2.4 | 0.5 | Assuming a granitic composition[a] | Assuming a granitic composition[b] |
| High density subcrustal bodies | 3242 | 4.15 | 0.01 | Mean value for dunites[a] assuming a depleted, high-density mantle material | Value for depleted peridotites[b] |
| Slab | 3163 | 3.3 | 0.001 | Assuming a prevalence of peridotites[a] | Eq. S1, using the average concentration of U, Th and K reported for depleted mantle[a] |
| Lithospheric mantle | 3D solution | 3 | 0.012 | Assuming a peridotitic composition[a] | Eq. S1, using the average concentration of U, Th and K reported for mantle[a] |

### 3.1.3 Data available for validating the thermal model


We validated the 3D thermal model (Sect. 4.1) by comparing available measurements of downhole temperatures (ANH, 2020) and surface heat flow (Lucazeau, 2019) -not used as model inputs, against the corresponding modelled values. The locations of the control points are shown in Fig. 3. Our goal was to minimize the misfit between the observed and modelled values. We found the modelled temperatures at the downhole sites to be particularly sensitive to changes in the thermal properties, thus allowing tuning these (Text S2 and Fig. S1). Heat flow values are, by definition, less reliable than direct temperature measurements. Thus, we have used only the heat flow with the highest qualities (error range between 10% and 20%, Lucazeau, 2019) for a secondary check. In general, the measured heat flow is lower within the Caribbean Sea (40-80 mW m$^{-2}$) than in the Pacific Ocean (>80 mW m$^{-2}$). Minima (10-40 mW m$^{-2}$) are found close to the area of influence of the Magdalena Fan depocenter (MFD, Fig. 3), likely as a result of the thermal blanketing by this thick sedimentary sequence (Scheck-Wenderoth and Maystrenko, 2013).


### 3.1.4 Geothermal gradient


We showcase the spatial variations in the geothermal gradient to demonstrate that 3D modelling is necessary to realistically calculate the thermal field in the study area. The geothermal gradient ($\nabla T$) was obtained considering the modelled temperature ($T_{i,j}$) at different depth levels ($z_{i,j}$), following Eq. 2. As the geothermal gradient is not constant with depth, we mapped its variation for depths ranging from the solid Earth surface (z=0) down to z=30 km, with incremental steps of 3 km (Fig. S6). Additionally, we mapped the geothermal gradient from z=0 to z=20 km, the latter being approximately the average crustal seismogenic depth in the region (Sect. 4.3).


$$\nabla T(z) = \frac{T_i - T_j}{z_i - z_j} \qquad \text{Eq. (2)}$$

A similar approach for calculating the geothermal gradient based on 3D thermal models was followed by Gholamrezaie et al. (2018), also using a 3D modelling scheme in which the geological heterogeneities of the system were included. This is particularly useful in complex tectonic settings such as the study area, where the application of a 1D geotherm approach for calculating the thermal field would not be representative of the present-day configuration.


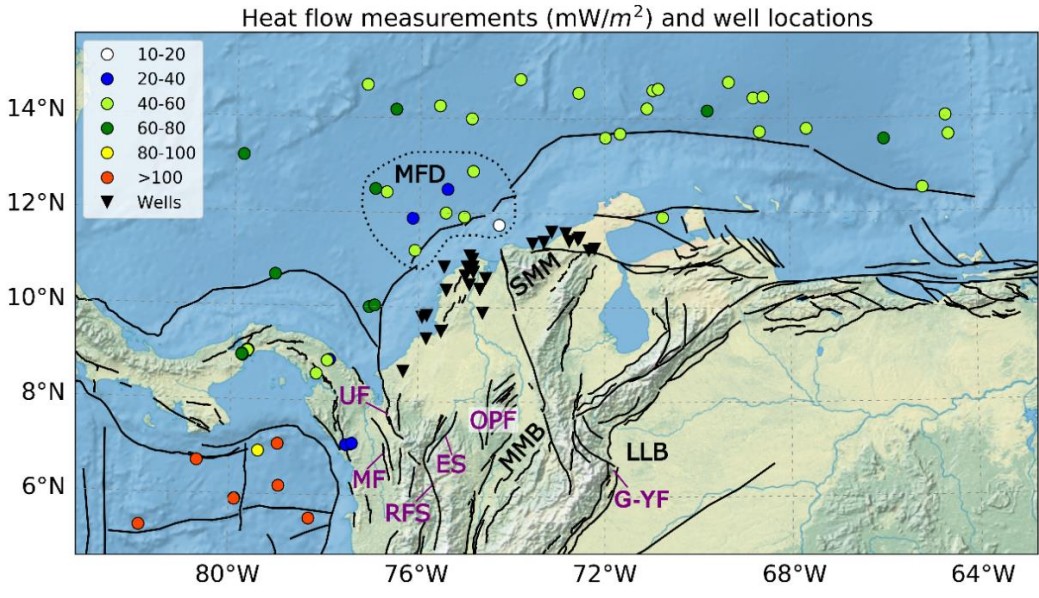

**Figure 3.** Measurements used for validating the thermal model. Color-coded dots: heat flow values with the highest qualities (Lucazeau, 2019). Black triangles: wells from the oil industry with measured downhole temperatures (ANH,

2020). Active fault traces (black lines) as in Fig. 1a. ES = Espíritu Santo Fault. G-YF = Guaicaramo and Yopal Faults. OPF = Otú-Palestina Fault system. RFS = Romeral Fault System. MF = Murindó Fault. UF = Uramita Fault. The dotted polygon highlights the heat flow values close to the Magdalena Fan depocenter (MFD). Additional features discussed in the text: LLB = Llanos Basin. MMB = Middle Magdalena Basin. SMM = Santa Marta Massif.

### 3.2 Crustal seismogenic depths

The crustal seismogenic depths were calculated from the earthquake catalogue, as described below.

### 3.2.1    Earthquake catalogue

A composite earthquake catalogue was compiled for the study area and surroundings (±1.5° of latitude and longitude), using public sources, with preference given to global databases over national ones. This last choice stems from the study area covering several countries, hindering a proper homogeneization of the local catalogues. For each event, the best location source, that is, the one with the most reliable depth, was chosen by using the following order of preference: 1) The gWFM database (Wimpenny and Watson, 2020), based on synthetic body-waveform modeling, and updated to version 1.2, which includes earthquake locations calculated in the region by Wimpenny, (2022) and Wimpenny et al. (2018). 2) Locations calculated by full-waveform modelling (with the ISOLA code; Sokos and Zahradnik, 2008) using records obtained at regional or local distances by the Colombian Geological Survey (Dionicio et al., 2023; Servicio Geológico Colombiano, 2023) and by Quintero et al. (2023). 3) A high-precision hypocentral relocation for the 2008 Quetame mainshock by Dicelis et al. (2016). 4) Locations with free (not fixed) hypocentral depth from the ISC-EHB dataset (Weston et al., 2018; Engdahl et al., 2020), which is compiled and curated by the International Seismological Centre (2023a). And 5) The prime locations reported in the reviewed ISC Bulletin (International Seismological Centre, 2023b) which has been completely rebuilt for the period 1964-2010 (Storchak et al., 2020), adding additional earthquakes and relocating hypocenters with the same location procedures used from 2011 onwards (Bondár and Storchak, 2011). Prime hypocenters are those relocated by ISC, or provided by regional agencies and considered by ISC as the best determined ones (Di Giacomo and Storchak, 2016). The resulting catalogue covers the period from January 1980 (when the ISC Bulletin became more homogeneous, e.g. Woessner and Wiemer (2005) until June 2021 (the last month fully revised in the ISC Bulletin at the time of writing).

Most locations in the catalogue were provided either by the ISC-EHB dataset or the reviewed ISC Bulletin. Only for 34 earthquakes, the location was provided by full-waveform inversion. In these cases, the location refers to the centroid (the center of seismic moment release), instead of the hypocenter (where the earthquake rupture starts). This adds some heterogeneity to the catalogue, as these two locations may not be the same for a given earthquake. However, high-precision full-waveform inversion locations are better constrained than hypocentral locations calculated from wave phase arrivals (e.g. McCaffrey and Abers, 1988; Nábělek, 1984), and most of these earthquakes had moment magnitude $M_w \leq 5.0$, with small rupture dimensions, which considering the location uncertainties leads to a negligible difference between the actual hypocentral and centroid location.

For each event, a preferred magnitude value was assigned. The original $M_w$ values reported from the sources 1 to 3 in the list above were used for the corresponding earthquakes. For those with locations provided by ISC (sources 4 and 5), $M_w$ was used if reported, preferably from the Global CMT catalogue (Dziewonski et al., 1981; Ekström et al., 2012), or alternatively from other agencies as reported by ISC (Di Giacomo et al., 2021). If no $M_w$ was available, we adopted the hierarchy proposed by ISC for selecting the most reliable, preferred magnitude type (Di Giacomo and Storchak, 2016). Earthquakes without reported magnitudes were disregarded.

In order to assess the magnitude of completeness ($M_c$) of the composite catalogue, we first checked the time series of magnitude values (Fig. S2, e.g. Gentili et al., 2011; González, 2017). This evidenced that small earthquakes were systematically better detected since June 1993, when regional seismic monitoring improved (Arcila et al., 2020). Therefore, we use this date to split the catalogue into two sub-periods with different mean $M_c$ (calculated with the maximum curvature method, Wiemer and Wyss, 2000; Woessner and Wiemer, 2005): $M_c = 4.6$ from January 1980 to May 1993, and $M_c = 3.5$ from June 1993 – March 2020 (Fig. S3). Spatial variations of $M_c$ within these periods were mapped and considered negligible for our analysis (Text S5).

The $M_c$ values for each period were used as minimum thresholds for the subsequent analysis despite in our study we do not investigate the corresponding magnitude-frequency distribution. For a given magnitude, deep earthquakes typically generate smaller amplitudes of ground motion, so they are more difficult to detect by seismometers and preferentially missing in the earthquake catalogues. Indeed, $M_c$ increases with depth when such dependence is quantified (e.g. Schorlemmer et al., 2010). So, if we would have relied on an incomplete catalogue
(considering also earthquakes with magnitude $<M_c$) the statistical results would be biased towards those resulting from shallow earthquakes, which are more likely to be detected and included in the catalogue. Also, the apparent spatial earthquake distribution would be distorted by numerous earthquakes with magnitude $< M_c$ located, e.g. in the vicinity of recording stations. Pruning the catalogue from earthquakes below $M_c$ should avoid such biases.

        Next, earthquakes with non-reported depths, as well as those with depths reported as 0 km or fixed, or with
reported depth error > 15 km were also excluded from the analysis. This selection allowed pruning the worst located earthquakes but preserving a sufficient number of events to perform our analysis. Note that the hypocentral depth errors reported in the ISC or ISC-EHB Bulletin format are wide, since they cover the 90% uncertainty range (Biegalski et al., 1999). The impact of the remaining hypocentral depth uncertainties on the results will be quantified later.

        The datum (reference surface used as depth=0) in the ISC Bulletin is the WGS84 reference ellipsoid (István
Bóndar & Dimitri Storchak, pers. comm., 2020; see also Bondár and Storchak, 2011). Our thermal model considers sea level as the reference surface, so hypocentral depths were referred to the EGM2008-5 geoid model (Pavlis et al., 2012; Hanagan and Mershon, 2021), which approximates well the sea level in the study area. The reference depth for locations provided by full-waveform inversions was considered as the solid Earth surface (e.g. Wimpenny, 2022), so their depths below sea level were calculated considering the topo-bathymetry only.

When selecting crustal seismicity, we disregarded earthquakes mislocated above the solid Earth's surface (according to the GEBCO topographic model, Weatherall et al., 2015), or located below the crust-mantle (Moho) boundary of the GEMMA model (Reguzzoni and Sampietro, 2015) interpolated to a homogeneous grid of 5 km × 5 km. We preferred the GEMMA model over others available in the region (e.g. Avellaneda-Jiménez et al., 2022; Poveda et al., 2018) because either these studies do not cover the entire study area, or portray large regions with data
gaps, as they relied on available seismic stations. The uncertainty of this model is represented in Fig. S4.

        The data repository (Gomez-Garcia et al., 2023) provides the catalogue subset of the best located crustal earthquakes in the study region, selected according to the criteria above, with their calculated hypocentral temperatures (Sect. 4.3). The histograms of their depth errors are shown in Fig. S5. For this subset, the scalar seismic moment ($M_0$, in N·m) was calculated (if not already provided by the original sources of the earthquake catalogue), from the standard
IASPEI formula for the moment magnitude $M_w$ (see Bormann, 2015 after Kanamori, 1977). If the preferred magnitude was not $M_w$, it was first converted to $M_w$ using the relations detailed in Text S6.

### 3.2.2 Crustal seismogenic depths and uncertainty quantification

        A robust statistical estimate of the crustal seismogenic depth at each location requires defining a given percentile of the observed distribution of nearby reliable earthquake depths. Simply considering the deepest
earthquake (percentile 100%) in the vicinity, despite sometimes used, is not robust, since the available sample of earthquakes is finite, and future ones will have some chance of being deeper than the deepest ones observed so far. A more stable statistical measure is the 90% depth percentile, D90 (Ellis et al., 2023; Marone and Scholz, 1988; Sibson, 1982) of the sample of nearby earthquakes, so that only 10% of them are deeper than this threshold. Indeed, D90 or D95 (the 95% percentile) are commonly used as proxies to the bottom depth (lower seismogenic depth) of seismogenic
sources in seismic hazard assessments (e.g. Bommer et al., 2023; Ellis et al., 2023). Alternatively, the average depth of the earthquakes deeper than D90 has also been used for this purpose (Zeng et al., 2022). Which of the two percentiles (D90 or D95) is more statistically meaningful depends on how many earthquakes are available in the considered sample. A sample of 20 earthquakes suffices to calculate D90 (e.g. Chiarabba and De Gori, 2016), so that two of them will be deeper. In contrast, with this sample size, D95 will eventually be less reliable than D90, as it again depends on
the deepest earthquake recorded. Larger percentiles, such as 99%, can be used only if there are many earthquakes in

the sample used, such as in high-seismicity regions with dense seismic monitoring (e.g. Marone and Scholz, 1988; Marone and Saffer, 2015; Wu et al., 2017; Scholz, 2019).

Here, D90 and D95 were spatially mapped considering the subset of crustal earthquakes with the best hypocentral depth determinations (see previous section). We used the median-unbiased percentile estimator of Hyndman and Fan (1996) at each node of a latitude-longitude grid with a spacing of 0.1°, considering the closest earthquakes to each node (at least 20) as the sample dimension for the corresponding D90 and D95 values. The resolution radius (distance to the furthest earthquake considered in each sample) was set to a minimum of 5 km (in order to cover at least one grid cell of the model). If there were <20 earthquakes within this distance, the radius was increased up to the 20-th closest earthquake. The percentiles were not calculated for nodes with resolution radius >120 km, where the spatial density of epicentres was deemed too low to obtain reliable results. To avoid boundary effects, we also considered earthquakes outside the study area, applying the same selection procedure, after checking that $M_c$ was not larger in this extended region. Given the sample size (20 events), the results based on D90 will be considered robust and interpreted here, while those of D95 will be only provided as supplementary information.

This way of spatially sampling a minimum number of the closest earthquakes around each map location is novel for calculating hypocentral depth percentiles, but it has been frequently used for mapping $M_c$ and $b$-values of the Gutenberg-Richter distribution (firstly by Wiemer and Wyss, 1997). Zeng et al. (2022) used a similar sampling method with variable resolution, but with a larger minimum sampling radius (50 km) and a minimum sample size not specified. The reason for our choice is that it maximizes the mapping detail, that is, the resolution radius will be small in locations with high spatial earthquake density, and large only if necessary, in those locations with sparse seismicity. We avoided the use of a larger earthquake sample for each node, as it would imply enlarging the resolution radius, considering earthquakes located further away from the nodes, and thus smoothing out the spatial variations of D90 (or D95).

In order to quantify the uncertainty of D90 (and D95) at each node, we relied on a combined Monte Carlo and bootstrap procedure. The Monte Carlo simulation accounts for the uncertainty due to reported errors in earthquake depth determination, while the bootstrap quantifies the uncertainty due to the finite size of the sample. In each of the 200 Monte Carlo runs used, a random depth was assigned for each earthquake. For this, it was assumed that its depth uncertainty followed a Gaussian distribution truncated at the solid Earth surface, with a mean given by the best depth estimate, and a standard deviation such that the reported error covers 90% of the uncertainty range, as stated for the locations provided by ISC (Biegalski et al., 1999), which constitute the bulk of the catalogue. Next, the spatial sampling described above was applied to the randomized set of hypocentres, only if they were located within the crust, according to the local depth of the GEMMA Moho model. Then, for each node, 100 random bootstrap samples (Efron, 1979) were generated out of the corresponding sample with at least 20 depth values. Thus, for each node there was a set of 20,000 (= 200 × 100) values of D90 (or D95) from which the average and standard deviation were calculated.

The resulting D90 (and D95) values and their corresponding standard deviations and resolution radii are provided in the data repository (Gomez-Garcia et al., 2023) and discussed further in Sect. 4.4. Due to the spatial sampling method used, in most nodes of the map the calculated D90 (or D95) lies within the crust, but there are some areas where the percentile may be located below the crust (such as in regions with abrupt changes in the Moho depth). Only the crustal D90 or D95 values (i.e. those whose depths are not deeper than the Moho) were considered.

## 4 Results and discussion

### 4.1 Thermal model validation

In Fig. 4a we compare the modelled and measured temperatures at boreholes (see Fig. 3), based on the selected, best fitting thermal model resulting from the sensitivity test (Text S2). The histogram of residuals (Fig. 4a, right) indicates that most misfits range between -10 and 10°C, with a mean of 5°C, the same magnitude as the common error for borehole temperature estimates. There is a general agreement between modelled temperatures (cyan dots) and measured ones (black dots). However, larger misfits occur at shallower depths (< 1km), which could be explained

by shallow advective processes of heat transport (e.g., by groundwater), not considered in our model, especially given the rather small spatial scales at which these processes occur, compared to our regional-scale approach.

Figure 4b compares the modelled and observed heat flow values. In general, heat flow measurements are usually affected by local, non-conductive processes of heat transport (such as hydrothermal circulation), making their
interpretation difficult in terms of a purely conductive, lithospheric-scale model, as pointed out elsewhere (Scheck-Wenderoth and Maystrenko, 2013; Klitzke et al., 2016). The modelled heat flow is generally lower than the measured one (Fig. 4b), except in the area of influence of the Magdalena Fan (Fig. 3). High observed values in the Pacific Ocean could also be attributed to additional advective heat transport, because they are located in an area of intense faulting (Marcaillou et al., 2006), close to the Panama Fracture Zone. Considering that the associated error in the heat flow
data used in this analysis ranges between 10 and 20% (Lucazeau, 2019), it can be concluded that the model fits the regional trend, except in those two areas previously mentioned.

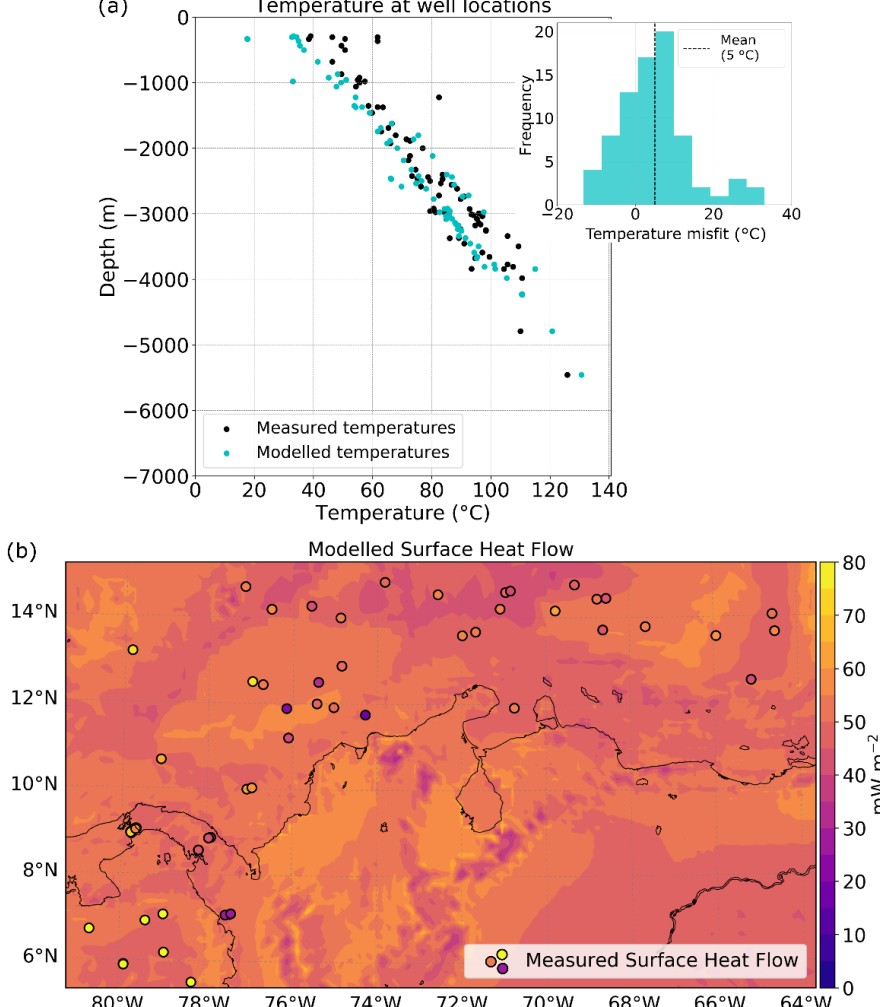

**Figure 4.** Validation of the 3D thermal field against measurements of downhole temperatures and surface heat flow. (a) Modelled borehole temperatures show a good agreement with the observed temperatures. The largest misfits
(histogram of the right panel) occur at depths shallower than 1km. (b) Calculated surface heat flow (background) and measured values (colored dots, with the same color scale).

### 4.2 Geothermal gradient: 3D variations and correlation with seismicity

Relying on a multi-1D geotherm approach, as commonly done, to compute the thermal field implies a spatially homogeneous setting where lateral variations in heat fluxes, as they do naturally occur, are disregarded. Implications of oversimplifying such three-dimensional interactions via 1D and 2D model representations have been already discussed, for instance, by Cacace and Scheck-Wenderoth (2016).

The thermal field within the lithosphere is influenced by factors such as: 1) the imprint from deep mantle sources; 2) the geometries of the different layers that compose the lithosphere; 3) their corresponding thermal conductivities; 4) the heat produced by the radioactive decay of elements present, especially in the (heterogeneous) crystalline crust; and 5) the thickness of sedimentary depocenters. Given that all these factors are not homogeneously distributed in space, a 3D thermal approach enables to better resolve all those interactions while preserving the heterogeneous subsurface configuration.

In particular, regional geothermal gradient variations can provide insights about the thermal state of the lithosphere (e.g. Gholamrezaie et al., 2018). Figure 5a shows the computed geothermal gradient for the regional seismogenic zone (from the surface down to 20 km below it, see Sect. 4.4). Long-wavelength spatial variations are observed both in the oceanic and continental realms, with minima offshore (13°C km$^{-1}$) and maxima underneath the Colombian Andes (up to 23°C km$^{-1}$).

Moreover, the geothermal gradients can also be used as an indirect indicator of crustal rheology. In Fig. 5a, it is possible to observe the correlation between the spatial distribution of seismicity and the geothermal gradients in this region. The crustal earthquakes occur at locations with a mean geothermal gradient of 19.4±1.2 °C/km$^{-1}$, preferentially clustering in specific zones, e.g. in the North Andes block and the Panama microplate. Seismicity is almost absent in cold lithospheric areas such as the Guyana craton and the Caribbean Large Igneous Plateau. This again, is an indication that a 1D geotherm approximation will not be robust enough to model the thermal configuration of the heterogeneous study area.

A quantitative measure of the correlation between the spatial distribution of seismicity and the geothermal gradients can be made with the so-called Molchan (or error) diagram (Molchan, 1990, 1991; Molchan and Kagan, 1992), already used to test the skill of geodynamic variables at forecasting the spatial distribution of seismicity (e.g. Becker et al., 2015).

The Molchan diagram (blue curve, Fig. 5b) results from considering all possible thresholds of the geothermal gradient in the map of Fig. 5a, following the procedure proposed by Zechar and Jordan (2008) for continuous 2-D forecast functions. Each point of the diagram shows the fraction of missed events (earthquakes occurred at or below a given threshold) versus the fraction of geographic area of the map covered above that threshold. For example, the lowest gradient at which an earthquake is observed in the map is 15.85°C km$^{-1}$; the areas where the gradient is at least at this threshold occupy 84% of the map (fraction of occupied space = 0.84) and below this threshold no earthquake occurred (miss rate = 0). Another example is the threshold at 18.73°C km$^{-1}$: exactly 30% of the map area has a gradient larger than this (fraction of occupied space = 0.30), and in those regions 72% of the earthquakes took place (miss rate = 28% = 0.28). The lower the threshold of geothermal gradient used, the lower the fraction of missed earthquakes, and the higher the fraction of occupied space.

A purely random guess with no skill would yield a curve close to the diagonal shown as a dashed line in Fig. 5b. For example, randomly choosing 20% of the map area should, on average, hit 20% of the earthquakes by chance, and miss 80% of them. A skilful correlation (or forecast) would yield a curve below this diagonal, with larger departures being more statistically significant (Zechar and Jordan, 2008). Fig. 5a contains *N*=1969 crustal earthquakes with well-determined depth (according to the criteria described in sect. 3.2.1), and for such a number, this departure of the curve is indeed statistically significant.

The area above the curve can be used as an overall measure of the skill (Zechar and Jordan, 2008). The latter can be quantified by the score *S* (Becker et al., 2015) given by the area above the Molchan diagram minus 0.5. This

exercise with the calculated geothermal gradient yields a score *S*=0.261. Albeit the results from different geographic regions cannot be directly compared, we should note that the value found is similar to those obtained by considering geodynamic variables (i.e.: shear strain rates and rates of topography change), as tested in western North America by Becker et al. (2015). We therefore can conclude that our correlation between the earthquake spatial distribution and the geothermal gradient is physically meaningful, also given that they are variables independent from each other.

As the geothermal gradient is a function of the temperatures at given depths (Eq. 2), it changes according to the depth interval used for its calculation; therefore, we explored its variation considering depth intervals of 3 km, from the surface down to 30 km depth (Fig. S6). Besides a general decrease in the geothermal gradient with depth, the most remarkable result is that in the continental realm there is not a constant pattern at all depths. In the elevated Andes mountains, the geothermal gradient reaches its maxima in the uppermost 6 km (Figs. S6a and S6b), but this
trend shifts at larger depths, where the highest gradients spatially correlate with thick sedimentary basins (Fig. S7). This behavior is consistent with an increase in the amount of radiogenic heat production associated with the thick crystalline crust of the Andes, and with the thermal blanketing effect of the sediments, which retains heat in the underlying crust (Scheck-Wenderoth and Maystrenko, 2013; Cacace and Scheck-Wenderoth, 2016).

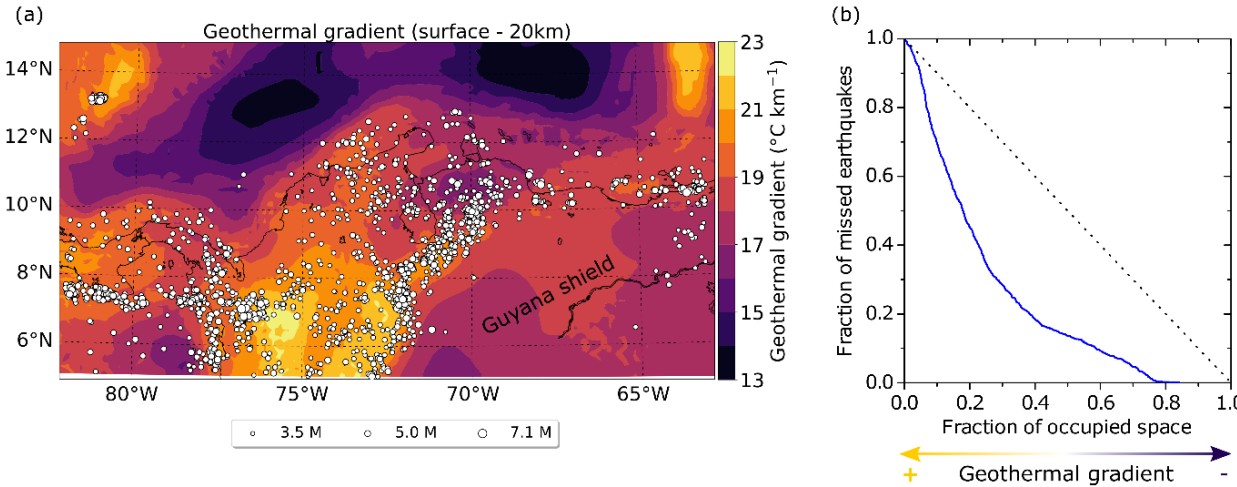

**Figure 5.** Correlation between geothermal gradient and the spatial distribution of crustal seismicity. (a) Geothermal gradient in the study area computed in the uppermost 20 km of the lithosphere (about the regional average of the crustal seismogenic depth, Sect. 4.3). Large spatial variations are observed both onshore and offshore. White dots: Crustal seismicity analyzed in this study (Sect. 3.2.1). (b) Molchan diagram showing the skill of the spatial distribution of the geothermal gradient at forecasting the distribution of crustal earthquakes.

**4.3 Relation between lithology, hypocentral temperature and seismic moment release**

The modelled hypocentral temperature distribution of the selected earthquake dataset is shown in Fig. 6. We focus our discussion around the three sub-regions as previously defined in Fig. 1a. Seismicity is frequent in region 1, as it hosts the Murindó seismic cluster, including the largest earthquake in the selected dataset ($M_w$ = 7.1), with a hypocentral depth of 21.1 km (Fig. 1a), and an associated modelled temperature of ~453°C. In the Otú-Palestina and
470 El Espíritu Santo fault systems (region 2), the deepest hypocentral depths in the crust are reported (> 30 km) (Fig. 1a), giving as a result modelled hypocentral temperatures of more than 600°C. In the Venezuelan Andes, bounded by the Boconó Fault (region 3), seismicity is spatially denser than in the rest of the North Andes region, and shows a shallowing pattern from the southwest towards the northeast (Fig. 1a). Such a trend implies a transition from higher hypocentral temperatures close to the Colombian-Venezuelan border towards lower hypocentral temperatures in the
475 Falcon basin (see location of this and other basins in Fig. S7).

A synthesis of modelled temperatures for the entire study area is presented in Fig. 7, where we also depict the seismogenic window as typically associated with granite (90-350°C; Blanpied et al., 1992; Scholz, 2019), gabbro (200-600°C; Mitchell et al., 2015; He et al., 2007; Scholz, 2019) and olivine gouge (600-1000°C; King and Marone, 2012; Scholz, 2019). Granitic rocks are typically regarded as the representative lithology in the crystalline continental crust. However, the study area has a variety of allochthonous terranes that have been attached to the margin, including large ophiolite sequences, associated to oceanic plateaus, and magmatic arcs (Fig. 1b) (Montes et al., 2019; Boschman et al., 2014; Kennan and Pindell, 2009); therefore, the seismogenic windows of gabbro and olivine are also considered.

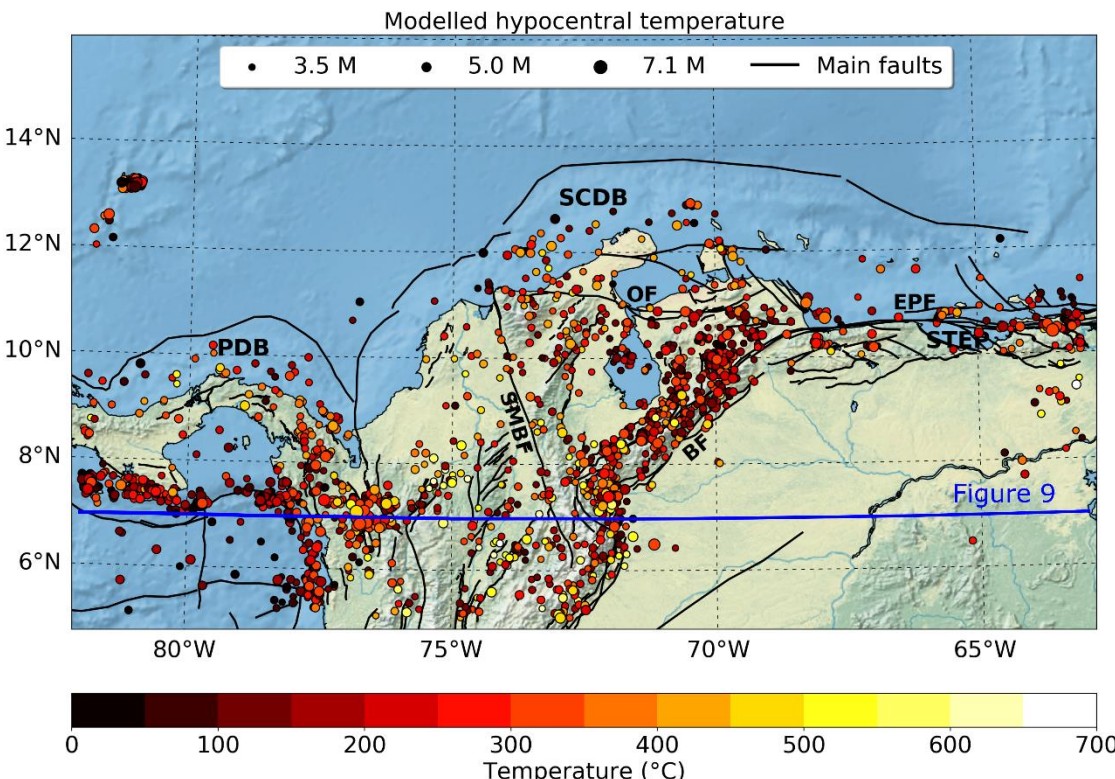

**Figure 6.** Modelled hypocentral temperature for crustal earthquakes. Acronyms and active fault traces (black lines) as in Fig. 1a. The surface projection of the vertical profile of Fig. 9 is shown as a blue line.

Most crustal earthquakes have hypocentral temperatures of less than 350°C (Fig. 7a), within the observed seismogenic window of granite and/or partially overlapping with that of gabbro (Fig. 7b). Nevertheless, modelled temperatures range from 1°C (offshore events) to almost 700°C, with only 13 events reaching the seismogenic window reported for olivine gouges at > 600°C. Such temperatures support early findings based on laboratory experiments (King and Marone, 2012; Scholz, 2019, and references therein), as well as more recent ones, suggesting that the brittle-to-ductile transition in mantle-forming minerals might occur at higher temperatures (> 600°C) than previously expected (e.g. Chen et al., 2013; Grose and Afonso, 2013; Ueda et al., 2020).

Despite relying only on the best-located earthquakes (see Sect. 3.2.1), uncertainties in the hypocentral depths still remain (Fig. S5). In any case, the overall trend of hypocentral temperatures is expected to be robust despite these uncertainties, as it is based on almost 2000 events. The hypocentral depths show a unimodal distribution, with the peak at about 5 km (Fig. 7c). Computing D90 associated to the whole catalogue of selected crustal earthquakes results in a regional average seismogenic depth for crustal earthquakes of about 20.5 km (blue dotted line in Fig. 7c).

Given a thermal model, errors in focal depths propagate into uncertainties in the hypocentral temperatures. The values represented in Figs. 6 and 7 are the most likely ones, corresponding to the best estimates of hypocentral locations; uncertainties have been omitted for clarity. For each earthquake, the possible temperature range can be measured directly from the 3D thermal model (Gómez-García et al., 2023), considering the depth range resulting from the best depth estimate plus/minus the formal 90% depth error. Also, an approximate estimate of its temperature uncertainty can be obtained by multiplying the depth error times the local geothermal gradient at the hypocentral location (e.g. Figs. 5 and S6). For deeper crustal earthquakes, both the formal depth errors (Fig. S5) and the local geothermal gradients (Fig. S6) are typically smaller than those for shallower events, implying typically smaller temperature uncertainties. Note that real hypocentral depth errors may be larger than the formal ones reported in the catalogues (e.g. Wimpenny and Watson, 2020), due to systematic errors in earthquake location procedures, such as in the assumed seismic velocity model (e.g. Husen and Hardebeck, 2010). Consequently, eventual improvements in velocity models and earthquake location accuracy will directly reduce the uncertainties in hypocentral temperature estimates.

Our analysis indicates that the 18 October 1992 Murindó mainshock (darkest blue dot in Fig. 7b) nucleated close to the regional base of the seismogenic crust (D90), and in particular, at the D90 depth at its location. This behavior supports early findings broadly debated in the literature (e.g. Tse and Rice, 1986), and suggests that ruptures which initiated within deep and high-stress regions are able to propagate through the entire seismogenic zone and probably reach the surface, resulting in a large rupture area, and therefore, in a large magnitude event. This event dominates the seismic moment release recorded so far in the study area, as can be observed on the seismic moment histogram as a function of depth (Fig. 7d). Its geological effects suggest a surface rupture exceeding 100 km in length (Mosquera-Machado et al., 2009), on the order of the overall rupture length deduced from the source-time functions of the earthquake sub-events (Li and Toksoz, 1993) and the size of the aftershock distribution (Arvidsson et al., 2002). Thus, we infer that the mainshock most likely ruptured the whole seismogenic crust, from its base up to the surface.

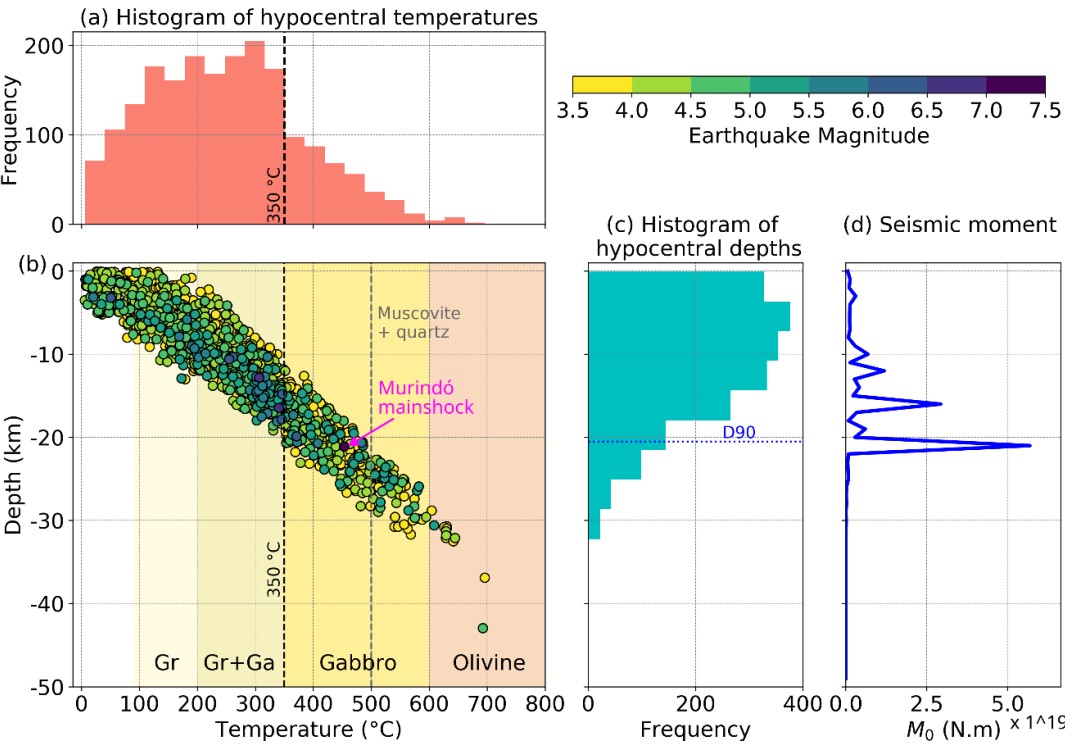

**Figure 7.** Synthesis of the modelled hypocentral temperatures. (a) Histogram of hypocentral temperatures. (b) Modelled temperature versus depth and preferred magnitude (color coded according to the scale shown in the upper

right). Colored domains in the graph represent the seismogenic windows of different rocks or minerals as reported by laboratory experiments (see main text). Gr = Granite. Gr+Ga = shared seismogenic window between granite and gabbro. (c) Histogram of hypocentral depths with regional (average) D90 = 20.5 km. (d) Histogram of seismic moment release ($M_0$, in N·m) as a function of depth, with depth bins of 1 km.

### 4.4 Depths and temperatures of the base of the seismogenic crust

The D90 depths, the associated temperatures, the D90 uncertainty estimates and the resolution radius are shown in Fig. 8. The D90 depths vary in space, ranging between 9.8 and 42.7km, and several abrupt changes can be traced to known crustal structures. Our results suggest a trend from shallower D90 depths and colder temperatures in the Greater Panama terrane (oceanic, with island arc affinity) towards deeper and hotter values in the sheared continental margin (Fig. 1b and blue polygons in Fig. 8b). In particular, in the D90 estimates, the Romeral Fault System (RFS, Fig. 3) seems to act as a boundary between the oceanic plateau-like affinity and the sheared continental environment. The latter is characterized by the deepest D90 values of NW South America, reaching up to 35 ± 3.5 km (corresponding temperatures ∼ 650°C). These maxima within our study region 2 are bounded by the Otú-Palestina and El Espíritu Santo fault systems to the west, and the western thrust front of the Eastern Cordillera to the east, where ophiolitic sutures are likely present (Kennan and Pindell, 2009). We interpret that the observed variability in D90 between the Central and Eastern Cordilleras and the Middle Magdalena Basin (MMB, Fig. 1b) evidences significant rheological contrasts between these areas. These major terranes are likely separated by crustal-scale faults (Kennan and Pindell, 2009).

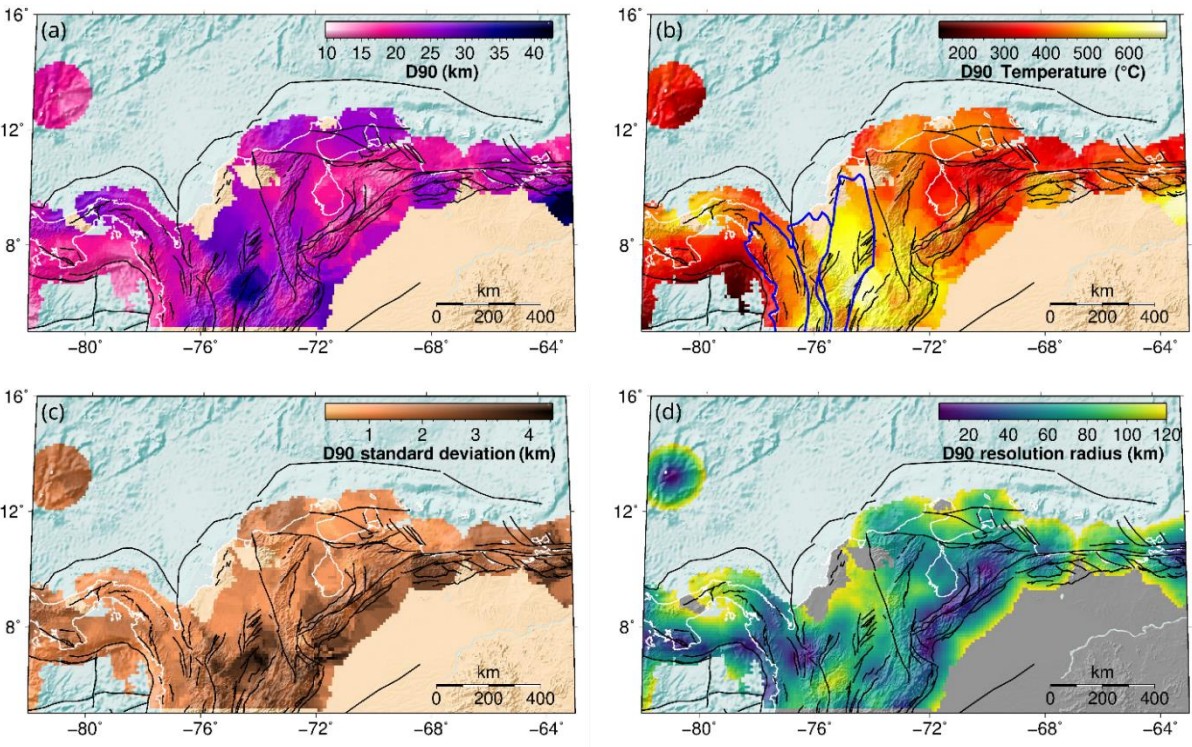

**Figure 8.** Results on D90, showing how the crustal seismogenic depth, and its associated temperature, varies spatially across the region. (a) D90 depths. (b) D90 temperature. Blue polygons: terranes in NW South America after Boschman et al. (2014) and Kennan and Pindell (2009), as represented in Fig. 1b. (c) Uncertainties in D90. (d) Resolution radius used to compute D90. Black lines: active fault traces, as in Fig. 1a. Coastlines are depicted as white lines.

D90 minima (9.8 ± 1.7 km) are located in the Venezuelan Andes (region 3) bounded by major faults along the Boconó system (BF, Fig. 1a). The associated temperatures indicate a transition from hot (deep) CSD in the SW of this region towards colder (shallower) values in the NE. In northern South America, the Oca-Ancon and El Pilar strike-slip faults (OF and EPF, Fig. 1a) seem to separate (to the north and south) tectonic blocks with diverse D90 depths and associated temperatures, suggesting a different rheological behavior. In fact, this margin was also highly affected by the C-LIP migration (e.g. Boschman et al., 2014).

Our results suggest that the temperatures at the base of the seismogenic crust in the continental realm span a relatively wide range (143°C to 690°C). In most of the study area, we found values larger than those reported as the onset of quartz plasticity (~300°C, Zielke et al., 2020) and in some cases larger than the temperature range consistent with quartz ductile behavior (350±100°C – see a detailed review by Chen et al., 2013). The D90 temperatures are also higher than the seismogenic window of rocks and mineral assemblages typically found in the continental crust (see Fig. 7 and Sect. 4.3), especially in region 2, which includes sutures of different (ultra)mafic, C-LIP related terranes (see Sect. 2).

The general patterns previously described are also present in the resulting D95 depths and estimated temperatures, although with a different range (12,4 km to 43, 9 km and 208 °C to 720 °C, respectively, Fig S8). The calculated uncertainties of D90 and D95 (due to the hypocentral depth errors and the size of the available sample of earthquakes) are very similar to each other (Fig. S9). The mean standard deviation of D90 is only 1.8 km (as for D95) and the maximum is 4.4 km (5.0 km for D95). Differences between D95 and D90 are found to be of little statistical significance overall, since, in most cases, these percentiles are between 1.0 and 1.5 standard deviations from one another. Such results indicate that the calculated D90 values are a robust proxy to the CSD.

Moreover, uncertainties in D90 (or D95) are similar to (or even smaller than) independent estimates of the Moho depth uncertainties (Fig. S4), evidencing that the estimation of D90 (and D95) is robust given all available data. The errors associated with the Moho geometry (Fig. S4) are significant across the Nazca and South American realms, resulting in uncertainties about the location of the events either in the lithospheric mantle (including both the mantle wedge and the subducting slab), or in the lower continental crust.

The spatial resolution of the D90 results highly depends on the spatial density of available earthquakes. This can be observed in the resolution radius map (Fig. 8d), which shows the search radius required for reaching a minimum of 20 seismic events to compute D90. As we allowed a maximum radius of 120 km, the map is truncated at this value. It is possible to observe how regions with dense seismicity required a small radius for reaching the 20 events, including the Murindó cluster (region 1) and the Venezuelan Andes (region 3).

Figure 9 shows a longitudinal profile along 7°N (see Fig. 6 for spatial location). Here, it is possible to observe the thermal response of the system, considering the spatially heterogeneous lower boundary condition at 75 km depth. Underneath the Pacific Ocean, the 600°C isotherm bounds the majority of the seismic events located within the crust and uppermost mantle (black filled and open dots), as previously suggested by Chen and Molnar (1983) and McKenzie et al. (2005), while the isotherm gradually shifts upward underneath western South America.

The thermal structure of the continental realm is usually more complex than that of the oceanic lithosphere. Our results suggest that the lithospheric mantle underneath the Colombian Andes is hotter than the surroundings, as indicated by a shallowing of the 600°C isotherm (Fig. 9). As a response, most of the crustal seismicity there preferentially occurs at shallower depths.

Nevertheless, deep events below the Moho interface (open dots) are also present in this area, especially associated to the Coiba (Nazca) slab, and perhaps the mantle wedge. Although direct estimations of the Moho depth are available at specific locations in the study area (Poveda et al., 2015; Avellaneda-Jiménez et al., 2022; Mojica et al., 2022), given the regional scope of our analyses, as already noted, we preferred to use the Moho of the GEMMA model (Reguzzoni and Sampietro, 2015). Considering the uncertainties in the hypocentral depths, and also in the

Moho estimates (up to ~7 km along this profile), it is especially challenging to make a clear statement about these upper mantle events. However, we can hypothesize that the subducting Coiba plate can host such intraplate events. Alternatively, the occurrence of upper mantle earthquakes is nowadays broadly recognized (e.g. Chen et al., 2013) as also dehydration reactions can trigger seismicity at temperatures above the normal brittle-ductile transition (e.g.
Bishop et al., 2023; Rodriguez Piceda et al., 2022; Jackson et al., 2008; Mackwell et al., 1998).

Two regions with prominent seismic activity at a crustal scale are recognized: the suture of the Panamá-Chocó block ("Greater Panama" terrane) with NW South America, around the Murindó cluster; and close to the Guaicaramo and Yopal faults (G-YF, Fig. 3), the boundary between the North Andes block (Eastern cordillera) and the Guyana shield. As previously mentioned, along this profile, most of the seismic activity in these areas is bounded
by the 600°C isotherm.

Variations of the base of the seismogenic crust (magenta dotted line in Fig. 9) are not necessarily correlated with variations in Moho depths. Between ~74°W and ~76°W (approximately corresponding to region 2, Fig. 1a), there is an abrupt deepening of D90, which correlates with a thick lower crust and with the shallowing of the 600°C isotherm due to the thermal imprint of a hot upper mantle. This deepening of D90 causes its correspondingly high temperatures
in region 2, as already discussed. Again, to explain that crustal earthquakes occur down to deeper locations despite of the hot lower crust, it is necessary to hypothesize that the latter has a mafic composition, with a deeper brittle-ductile transition. As mentioned in Sect. 4.3, this profile shows the close proximity of the 1992 Murindó's mainshock hypocenter (cyan star) to the base of the seismogenic crust.

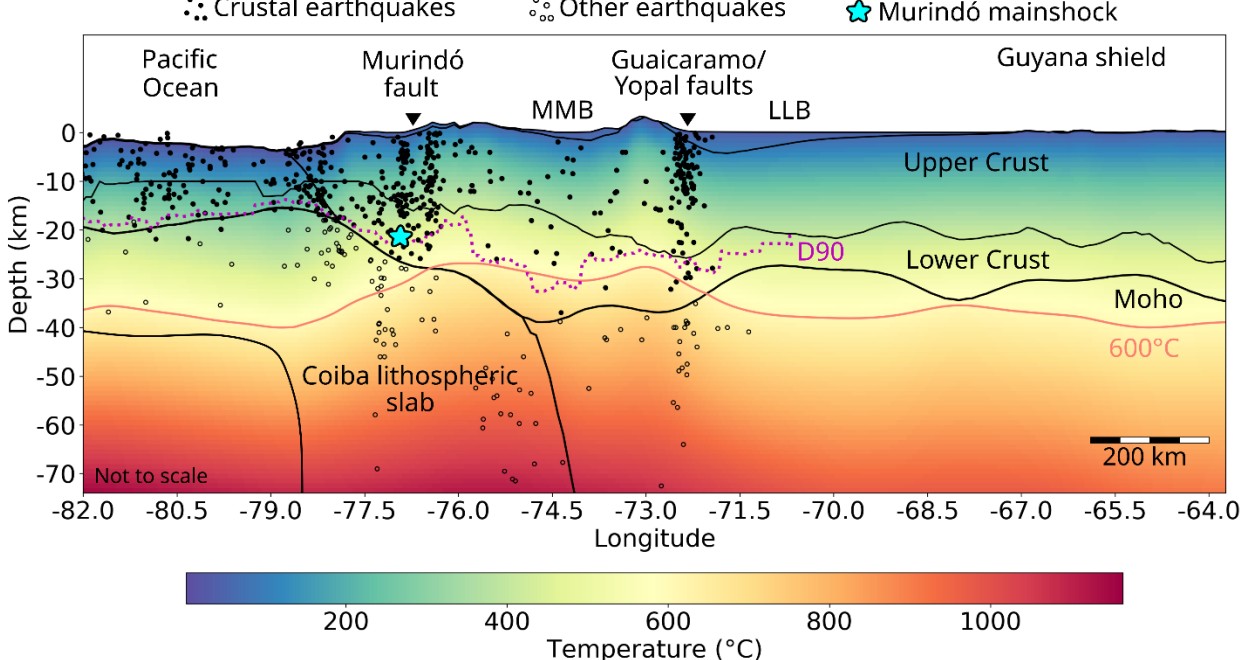

**Figure 9.** W-E profile at 7°N (see location in Fig. 6) showing the modelled temperatures and their relation to the lithospheric structure, topography and seismicity. Vertical exaggeration 8.5X. Black lines: Boundaries of the lithospheric layers of the structural model as integrated by (Gómez-García et al., 2020, 2021). Pink continuous line: 600°C isotherm. Magenta dotted line: D90. Black filled dots: Crustal earthquakes used in this study. Open dots: Earthquakes deeper than the Moho interface, not used for calculating D90. The earthquakes projected in the profile
include those from 6.5°N to 7.5°N. Cyan star: Hypocenter of the largest earthquake (Murindó mainshock). LLB = Llanos Basin. MMB = Middle Magdalena Basin (which spatially corresponds to region 2).

Regarding the occurrence of crustal earthquakes at temperatures higher than the seismogenic windows expected for typical crustal rocks, it can be remarked that: 1) The earthquake dataset includes aftershocks (as otherwise the number of events for analysis would be further reduced), which may nucleate at depths larger than the base of the background seismogenic zone (e.g. Zielke et al., 2020). Thus, the calculated D90 values may be affected by transient deepening of the seismogenic crust during aftershock sequences. These deeper values would yield a larger temperature for the CSD than the long-term one. 2) The diverse allochthonous terranes accreted to NW South America, and the variety of autochthonous crustal blocks, include (ultra)mafic, olivine-rich rocks, which could host seismicity at larger temperatures. 3) The lower crust under part of the Andes may be mafic, able to host earthquakes at the relatively high modelled temperatures, which are due to a hot upper mantle together with a thick upper crust (which generates additional heat due to the decay of radioactive elements, Vilà et al., 2010).

## 5 Summary and conclusions

We present a three-dimensional, data-integrative model of the thermal field in the north Andean region and the transition to the Caribbean, which displays spatial temperature variations that would have been overlooked by simplified 1D or 2D models. The model fits the available observations of borehole temperatures, and approximates the first-order trend of heat flow values.

This modelling workflow provides an opportunity to compare limiting temperatures for seismogenesis provided by laboratory experiments against real-case scenarios, by considering geological complexities, including a realistic lithospheric structure and the mantle imprint on crustal temperatures.

We have mapped the variable crustal seismogenic depth (CSD) in the region, based on an earthquake catalogue compiled from global sources. The sampling procedure used allows to identifying the variations of the CSD with greater detail in areas with higher spatial earthquake density. Some of those variations are shown to correlate with crustal-scale faults in the region, which acts as tectonic boundaries for crustal domains with different seismogenic behaviors.

Most crustal seismic events in the study area have modelled hypocentral temperatures < 350°C, and are located at depths < 20 km. Although most of the hypocentral temperatures range in the reported seismogenic window of rocks and mineral assemblages typically found in continental crust, some of the deepest hypocenters have associated temperatures > 600°C, reaching the seismogenic window of olivine. We suggest that the diverse allochthonous crustal blocks, which have been attached to the NW South American margin, and which include large ophiolite sequences, contain olivine-rich, ultramafic rocks, thus explaining these deeper earthquakes. Alternatively, these high-temperature events can be explained by a thick, mafic lower crust (with a relatively deep brittle-ductile transition), or by the depth uncertainties of the Moho (up to 7 km in the study area) and those of the hypocenters, which could imply that some of those events actually occurred in the upper mantle. The overall coherence of the calculated hypocentral temperatures with those expected from laboratory measurements provides additional, indirect support to the model, and *vice versa*.

We additionally found that the spatial distribution of seismicity strongly correlated with the geothermal gradients in the uppermost 20 km of the lithosphere. The Molchan diagram indicates that the geothermal gradient may be as skillful at forecasting the spatial distribution of seismicity as other geodynamic indicators (i.e. strain rates) usually adopted in previous studies. To our knowledge, this skill test had not previously been quantified elsewhere, so we encourage further studies in other regions to explore the systematic nature of the correlation found in our analysis.

Our results evidence that the rupture of the largest event in the region since 1980 ($M_w = 7.1$, Murindó sequence of 1992) propagated from the base of the crustal seismogenic zone. This highlights the importance of considering this transition while defining the lower boundary of seismogenic sources in any seismic hazard assessment.

The estimated CSD in the Otú-Palestina and El Espíritu Santo fault systems is one of the deepest in the study area (up to ~35 ± 3.5 km), as most of the deepest events have been recorded beneath these regions. This suggests that

these fault systems likely behave as crustal-scale structures, which might have the potential of rupturing large fault areas, thus likely resulting in large-magnitude, hazardous events.

Future analysis will benefit from improved and enlarged thermal and seismic datasets. Additional measurements of heat flow and borehole temperatures (especially within the continent) would better constrain the thermal model. Moreover, as time passes and new seismic stations are installed, more earthquakes are being recorded (particularly of smaller magnitudes than those considered in the present study, $M < 3.5$) with improved depth accuracy. Therefore, uncertainties in hypocentral locations and on the crustal seismogenic depths (and associated temperatures), could be eventually reduced.

## Data availability

The results of this publication are available in the data repository Gómez-García et al. (2023). It includes the calculated 3D thermal model, the catalogue of selected earthquakes with their modelled hypocentral temperatures, geothermal gradient and seismic moment, and the mapped depths, uncertainties and temperatures of the CSD (D90 and D95).

The thermal calculations were computed using the software GOLEM (Cacace and Jacquey, 2017) available at Jacquey and Cacace (2017). The figures were created using diverse Python packages (Python Software Foundation. Python Language Reference, version 2.7, available at http://www.python.org) and GMT (Wessel & Smith, 1991).

## Author contribution

AMG developed the research idea, processed the data, performed the thermal calculations and wrote the manuscript. AGG contributed to the research idea, compiled the earthquake catalogue, calculated the crustal seismo-genic depths and wrote the manuscript. MC developed the software GOLEM and provided support for the thermal calculations. MS and GM supervised the implementation of the thermal calculations. All authors contributed to the discussion and interpretation of the results, revised the manuscript and were partially responsible for obtaining the financial support to develop this research.

## Competing interest

The authors declare that they have no conflict of interest.

## Acknowledgments

AMGG was partially supported by grants from the German Academic Exchange Service (DAAD, 57314023 and 57440918), the Corporation Center of Excellence in Marine Sciences (CEMarin), Fundación para la Promoción de la Investigación y la Tecnología (Banco de la República de Colombia), the Centre de Recerca Matemàtica (CRM) in Barcelona and the ESM-project of the Helmholtz Impulse and Networking Funds. Grants from MCIN/AEI/10.13039/501100011033, and NextGenerationEU/PRTR were also granted to AMGG (FJC2021-047434-I) and ÁG (FEDER, IJC2020-043372-I and PID2021-125979OB-I00). AMGG is grateful to Antoine Jacquey for his advice during early versions of the thermal models and with Maximilian Frick for his guidance on the use of Paraview. We would like to acknowledge the contributions of Sam Wimpenny and an anonymous reviewer for their constructive feedback.

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
