# Peer review of "Thermal structure of the southern Caribbean and NW South America: implications for seismogenesis"

_EGUsphere, 2023_

## Referee Comment (RC1)

**Review of "Crustal Seismogenic Thickness and Thermal Structure of NW South America" by Gómez-Garciá et al., submitted to Solid Earth.**

**Review by:** Dr Sam Wimpenny, University of Leeds, UK.

**Manuscript Overview**

Gómez-Garciá et al., present an analysis of the spatial variability in the depth of seismicity within the NW South America and its relationship with the thermal structure of the lithosphere. Their work touches on a long-running research question regarding the controls on the depth of seismicity within the continents [e.g. Chen and Molnar 1983; McKenzie et al., 2005; Jackson et al., 2008]. Gómez-Garciá et al., argue that the majority of seismicity in the NW Andes nucleates in rocks at temperatures <350 degrees Celsius, with a smaller number of earthquakes nucleating in crustal or mantle rocks at temperatures higher than 350 degrees Celsius. They argue that earthquakes nucleating in rocks at temperatures >350 degrees Celsius indicate that they may have more mafic lithologies, or that the depth estimates of these events may be inaccurate. The authors also find that there is a strong correlation between areas of elevated geothermal gradient and seismicity, but do not provide a physical interpretation for the correlation.

I found the manuscript relatively easy to read and well annotated, though in a number of places the text needs adjusting to remove some ambiguous statements and definitions (I have outlined these in the line-by-line comments).

I do have methodological concerns with the analyses presented in the manuscript, including with the accuracy of the routine depth estimates used, the method used to compute the seismogenic thickness, and the assumptions made in computing the temperature structure. These concerns make it difficult (at least in my view) to assess how robust the major conclusions in the manuscript currently are. Nevertheless, I have outlined how the authors might address these concerns.

A criticism I have with this manuscript is that it has mostly overlooked the extensive literature on the relationship between earthquake depths, geology, and temperature within the continents that is directly relevant to the arguments they present [Jackson 2002; Jackson et al., 2004; McKenzie et al., 2005; Jackson et al., 2008; Sloan et al., 2011; Craig et al., 2012]. I recommend the authors more explicitly address how their conclusions advance on the findings made in the aforementioned papers, as this will help contextualise the contribution that this paper makes to the field.

Overall, because of the technical concerns I have with this manuscript, I would recommend that the manuscript be returned for **major corrections** in order for the authors to address these concerns.

**General Comments**

1. **Inaccuracies in the ISC earthquake hypocentral depths limit the earthquake depth analyses presented:** The authors use the ISC catalogue's hypocentral depth estimates to map out the depth distribution of seismicity. The ISC locates earthquakes using the reported travel times of body waves, and for events less than ~40 km deep the ISC location procedure can be incorrect by tens of kilometers in depth. These errors are not necessarily reflected in the formal uncertainties in the catalogue, which is a well-known limitation associated with using the ISC earthquake catalogue to study shallow seismicity [Maggi et al., 2000; Chen et al., 2009; Weston et al., 2011] and is something the authors acknowledge. Most studies use waveform modelling of the earthquakes to accurately determine earthquake depths and thereby draw robust conclusions regarding the depth of seismicity in the continents [see examples for South America in: Suarez et al., 1983; Chinn and Isacks 1983; Devlin et al., 2012]. Without accurate earthquake depth estimates, it is unclear to me to what extent variability in the calculated seismogenic thickness, D10 or D90 represent errors in the earthquake depths in the ISC catalogue, or real spatial variability in the depth of earthquake generation.

To address this comment, I would recommend that the authors re-analyse the depths of earthquakes in their study area using waveform modelling of teleseismic [e.g. Devlin et al., 2012; Wimpenny 2022] or regional [e.g. Alvarado et al., 2005] seismic data. Alternatively, if the author's believe that the depths of crustal earthquakes in the ISC catalogue are accurate enough for their purposes, then they need to demonstrate this by comparing the ISC event depths with an independent source of earthquake depths (e.g. from local seismic deployment), because in most other settings the ISC event depths are not accurate enough for these purposes.

2. **The method used to compute the seismogenic thickness does not necessarily account for the real depth-distribution of events in each grid area:** The authors say that the D10/D90 statistics are calculated on a 0.1x0.1 degree grid by considering the nearest 20 events to each grid centre as the sample from which D10/D90 are computed. This method appears biased to me. Consider a case where there are 100 events within any particular 0.1x0.1 degree grid area, and the nearest 20 events to the grid centre are all <10 km depth but the remaining 80 are all at 50 km depth. The current method for estimating the D90 would yield a value <10 km for the whole grid area, despite the fact that earthquakes are occurring down to 50 km. Therefore, I suspect the method used to compute the seismogenic thickness could yield misleading results. This might not be an issue if the typical earthquake spacing in lat/lon is larger than ~0.1x0.1 degree, but the authors should add new results demonstrating that their interpretations are independent of the gridding scale and sample size used in calculating the D10/D90/seismogenic thickness.

3. **D10 and D90 may not be the relevant metrics for understanding the absolute depth range of earthquake nucleation:** The D10 and D90 parameters have been developed to study temporal variations in the depth of seismicity in regions with dense earthquake catalogues (e.g. California, Japan; see Rolandone et al., [2004]). These metrics do not seem suitable when studying the depth-extent of seismicity in the ISC catalogue. Firstly, the ISC catalogue is relatively sparse, and so choosing D10/D90 may not be robust as it might vary with longer observation intervals. Secondly, the 10% and 90% cut-off are arbitrary values – the more common and logical definition is that the seismogenic thickness is the depth of the deepest observed earthquake in an area [like used in Maggi et al., 2000; Jackson et al., 2004]. Could the authors add some explanation as to why they use D10 and D90 and why it is a relevant metric?

I will also suggest that D10 is unlikely to be a robust metric when calculated from the ISC catalogue, because hypocentral locations derived using travel time data have particularly poor depth resolution within the top 10 km of the crust. Weston et al., [2011] and Wimpenny and Watson, [2021] have demonstrated the poor resolution of ISC-EHB event depths in the upper crust by comparing ISC hypocentral depth estimates with both finite-fault slip solutions derived from modelling InSAR data and more accurate earthquake centroid depths from waveform modelling. Both studies found that there can be differences on the order of 5-10 km, and that the ISC systematically overestimates the depth-range of slip in shallow earthquakes.

4. **The temperature field within the mountains is most likely not in steady state:** The authors have assumed that the temperature field throughout NW South America can be modelled as being in steady-state. The time taken for a system to approach thermal steady state is given by the thermal time-constant $\tau = l^2/\pi^2\kappa$ where $l$ is the thickness of the lithosphere and $\kappa$ is the thermal diffusivity of the lithosphere (~10–100 Ma for normal lithosphere). Given that mountain building will advect heat and move the lithosphere out of thermal steady-state [England and Thompson, 1984], and that mountain building in the NW Andes has taken place more recently than 10-100 Ma, then I would assume that the temperature field within the NW Andes cannot be modelled as being in thermal steady state. I agree that the steady-state assumption is more reasonable within the forelands.

The heat flow and downhole temperature data the author's use to validate the models come from the top 4 km of the crust, and are mostly located offshore and not within the continental lithosphere, therefore are not a good test of whether the model is representative of the temperature field at depth in Region 1, 2 or 3. To address this comment, the authors need to justify why they think the steady-state assumption is valid for the mid/lower crust.

5. **Lab and nature suggest olivine remains seismogenic up to ~600 degrees – not 600-1000 degrees:** There are two strong lines of evidence that suggest olivine-rich rocks can only remain able to nucleate earthquakes up to temperatures of ~600 degrees Celsius. The first comes from the depths of well-constrained earthquakes within the oceanic lithosphere, where the temperature structure is well known. Here the maximum centroid depths of large earthquakes can be seen to deepen with the age of the lithosphere and remain consistently shallower than the 600 degree isotherm [McKenzie et al., 2005; Craig et al., 2014]. Secondly, laboratory experiments show that olivine aggregates switch from deforming through stick-slip, to deforming through stable creep, at temperatures equivalent to ~600 degrees at the strain rates expected within the oceanic lithosphere [Boettcher et al., 2007]. The authors cite Scholz [2019] (a textbook) regarding the seismogenic range of temperatures for olivine being 600-1000 degrees, but do not describe the evidence for this. I would suggest that the author's cite the original papers that came to this conclusion, as I have not been able to check where their logic has come from.

**Line-by-Line Comments**

**Line 11:** "Crustal seismogenic thickness" is a misleading term in my view, because the seismogenic layer can include parts of the crust and upper mantle. I would suggest that just the depth extent of earthquakes in the lithosphere is the more logical description of the seismogenic thickness, and the depth most likely to correlate with a "stability transition" from seismogenic to aseismic deformation processes.

**Line 18:** "Potential temperature" means specifically the temperature at which a rock would be if moved from a particular depth along an adiabat, which I think is different from what the authors mean here. I think they mean just the possible absolute temperature, and would recommend removing the word "potential".

**Line 32:** Why is the crustal seismogenic thickness not just the layer where all earthquakes occur? I'm not sure why it would be defined as where 'most', but not all, earthquakes occur.

**Line 39:** Is 90% of events based on a measure of statistical significance, or is it just defined as an arbitrary value?

**Line 46-48:** I would argue that it is generally accepted that earthquakes *mostly* nucleate within the crust at temperature <350 degrees Celsius, but *can* nucleate in lower crustal rocks at temperatures up to 600 degrees Celsius within certain areas where the lower crust might be dry [see Jackson et al., 2008; Craig et al., 2011; Sloan et al., 2011; Craig et al., 2012]. Examples of lower-crustal seismicity are now extensive, and there are plenty within South America [Assumpcao et al., 1992; Devlin et al., 2012; Wimpenny 2022].

**Line 73:** Typo "as the extent of the CST" not "extend".

**Line 77:** It is not clear to me why flat slab subduction means that you can assume thermal steady state. If anything, I would assume that flat slab subduction would suggest that you need to account for the heat advection in the modelling of the temperature field, because the slab has not always been flat and therefore the temperature boundary condition on the base of the lithosphere has changed through time. Please could the authors elaborate on why a flat slab configuration allows them to make an assumption of thermal steady state?

**Line 92:** What is meant by "large uncertainties" in the Moho? Could the authors give quantitative estimates of the typical Moho uncertainty?

**Figure 1:** There are a number of references to different geographic places (e.g. Panama-Chaco Block) in the text that are not defined on Figure 1. Please could the authors add these place names to the figure to help the reader.

**Line 242-245:** The authors "disregard earthquakes below the Moho...", and use only events above the Moho to compute "... the upper and lower stability transitions". By ignoring earthquakes below the Moho, then the maximum depth of earthquakes the authors calculate is limited by the crustal thickness, and doesn't necessarily correspond to any "stability transition" or mechanical condition in the lithosphere. I would recommend changing the phrasing the authors are using to describe what they are actually measuring, which is more like the fraction of the crust that is seismogenic.

**Line 263:** Why is removing events that are below the magnitude of completeness (Mc) relevant for this analysis, because the authors are not necessarily studying earthquake frequency or moment release through time? Small events below Mc could still be useful in defining the depth distribution of seismicity.

**Line 274-276:** The bootstrapping method of estimating the uncertainties in D10/D90 yield what I would consider unrealistically small values of <1 km. This is presumably because the 'formal' uncertainties in earthquake depths included in the ISC catalogue that are propagated through the bootstrapping are not realistic representations of the true depth errors. For example, comparing earthquake hypocentral depths in the ISC-EHB catalogue with more accurate centroid depths calculated from body-waveform modelling yields a mean difference on the order of ~5±10 km with errors of up to 50 km [Wimpenny and Watson, 2021]. Therefore, I expect a more realistic uncertainty is at least 5-10 km.

In addition, the inference that "*low uncertainties [estimating from bootstrapping] indicate that using 20 earthquakes for each node is [sic] already reliable...* " is not necessarily true in my view. If you used a sample size of 1 earthquake, then you would get low uncertainties, but the uncertainty estimate is not necessarily robust in that changing the sample used would give a different answer. The way to check whether 20 events is a reasonable sample size for giving robust results would be to make sure that increasing the sample size does not alter the conclusions (even if the spatial resolution has to decrease as a product of the greater sample size).

**Line 292:** I recommend quoting the misfit as 5 degrees, not 4.99 degrees, as the latter makes it sound like the model and data have precision to two decimal places, which is unlikely.

**Section 4.1:** A more general comment is that the thermal model validation relies on comparing measurements of the temperature in the very shallowest part of the crust in areas that are mostly offshore and do not overlap with Regions 1, 2, or 3. Could the authors add some text explaining why they think that validating the model using measurements from outside of the region of interest means that it will be valid in the regions of interest, which may itself have a different geological history and therefore different temperature structure (e.g. the effects of mountain building on the geotherms).

**Line 317:** Why is the "regional seismogenic zone" defined as 20 km? This seems arbitrary to me, and different from the seismogenic thickness quoted elsewhere.

**Line 326-329:** I would recommend re-wording this sentence, as on first reading I thought the authors were suggesting that areas with high geothermal gradients were areas with a strongly coupled crust and lithospheric mantle, though they actually mean it is the other way around. Maybe just split it up into a few shorter sentences for clarity.

**Line 348-354:** This is an interesting test, and it is physically intuitive that areas of higher geothermal gradient will have more seismicity because the lithosphere is hotter, therefore weaker and deforms more rapidly in those areas. However, I would be interested to know whether the thermal modelling adds any more useful information on where earthquakes are expected to occur relative to direct surface observables such as topography or fault traces.

**Line 407:** Please cite the original source for the constraint on the seismogenic temperatures for olivine gouge as 600-1000 degrees.

**Line 415:** Why does shallow earthquake hypocentres mean faults are 'not well developed', and what is meant by 'not well developed'? Do you mean 'immature' in that they haven't accommodated a significant amount of relative motion? Is there any evidence from natural fault zones that the hypocentral distribution of earthquakes differs with the 'development' of the fault, or are these inferences based off of laboratory experiments?

**Line 417-419:** This sentence captures much of my concern regarding the results of this study. The authors state: *"However, despite relying only on the best located earthquakes (see Sect. 3.2.1), large errors in the hypocentral depths still remain (up to 30 km, see Fig. S5), and should be considered in the analysis of our results."* I agree, but it's not clear to me how the authors have considered this in their own analysis of their results. In my view, the best way to mitigate this issue is to use more accurate methods for determining earthquake depths, which are more time consuming but will yield more robust results.

**Line 443:** What do the authors mean by "D90 depths of almost 35 km … are in agreement with the crustal-scale structure that these systems likely represent"? Faults can be 'crustal scale' but be aseismic still presumably (e.g. segments of the San Andreas Fault).

**Line 445:** "The D90 values … are clearly bounded by major faults" – I find it hard to see these patterns reflected in Figure 8, mainly because there are a lot of faults that do not seem to bound changes in the colour patterns. Maybe highlight the specific faults thought to bound the changes in D90 in the Figure?

**Line 560-565:** There are a few points where the authors mention the diverse geology of the northern Andes and how mafic/ultramafic terranes at depth might account for the deep seismicity. These arguments would benefit significantly from a simplified geological map of the study that highlights where these ultramafic/mafic terranes are based on surface outcrop and display these data relative to the seismicity and D90 maps.

---

## Author Comment (AC2)

**"Crustal Seismogenic Thickness and Thermal Structure of NW South America"**

**Author's response to review by Dr. Sam Wimpenny.**

We sincerely thank Dr. Wimpenny for his prompt review and insightful comments. We found them very useful, and we very much appreciate his feedback. Please find below a point-by-point reply to the issues raised by him.

Considering his comments and concerns, we plan to improve the manuscript upon its previous version. We hope that this new version supports in more detail the analysis performed and better frames the contribution that the paper makes to the field.

New references (not already cited in the manuscript) are listed in a separate section at the end.

**Manuscript overview**

**Comment:** ..."The authors also find that there is a strong correlation between areas of elevated geothermal gradient and seismicity, but do not provide a physical interpretation for the correlation."

**Response:** Thank you for pointing this out. We will add a short paragraph in this regard.

**Comment:** A criticism I have with this manuscript is that it has mostly overlooked the extensive literature on the relationship between earthquake depths, geology, and temperature within the continents that is directly relevant to the arguments they present [Jackson 2002; Jackson et al., 2004; McKenzie et al., 2005; Jackson et al., 2008; Sloan et al., 2011; Craig et al., 2012]. I recommend the authors more explicitly address how their conclusions advance on the findings made in the aforementioned papers, as this will help contextualise the contribution that this paper makes to the field.

**Response:**

Most of the papers cited by the reviewer have been in turn already cited and discussed by the references included in the manuscript, such as the reviews by Scholz (2019) and Chen et al. (2021). But, of course, we can add more references to better frame the results. Our findings are not at odds with earlier ones, but are based on a much more detailed thermal modelling than typically achieved. For example, the references cited by the reviewer base their temperature estimates inside the Earth on 1-D or 2-D thermal models, using simplified lithospheric structures, instead of more realistic 3D models as implemented in our work. Nevertheless, this will be specifically addressed in the next version. Please see our response to comment #5, as it is also related to this issue.

**General comments**

1. **Inaccuracies in the ISC earthquake hypocentral depths limit the earthquake depth analyses presented:** The authors use the ISC catalogue's hypocentral depth estimates to map out the depth distribution of seismicity. The ISC locates earthquakes using the reported travel times of body waves, and for events less than ~40 km deep the ISC location

procedure can be incorrect by tens of kilometers in depth. These errors are not necessarily reflected in the formal uncertainties in the catalogue, which is a well-known limitation associated with using the ISC earthquake catalogue to study shallow seismicity [Maggi et al., 2000; Chen et al., 2009; Weston et al., 2011] and is something the authors acknowledge. Most studies use waveform modelling of the earthquakes to accurately determine earthquake depths and thereby draw robust conclusions regarding the depth of seismicity in the continents [see examples for South America in: Suarez et al., 1983; Chinn and Isacks 1983; Devlin et al., 2012]. Without accurate earthquake depth estimates, it is unclear to me to what extent variability in the calculated seismogenic thickness, D10 or D90 represent errors in the earthquake depths in the ISC catalogue, or real spatial variability in the depth of earthquake generation. To address this comment, I would recommend that the authors re-analyse the depths of earthquakes in their study area using waveform modelling of teleseismic [e.g. Devlin et al., 2012; Wimpenny 2022] or regional [e.g. Alvarado et al., 2005] seismic data. Alternatively, if the author's believe that the depths of crustal earthquakes in the ISC catalogue are accurate enough for their purposes, then they need to demonstrate this by comparing the ISC event depths with an independent source of earthquake depths (e.g. from local seismic deployment), because in most other settings the ISC event depths are not accurate enough for these purposes.

**Response:**

To summarize our reply to this issue: In this research, we have used the ISC Bulletin because it is the most complete, public and homogeneous seismological catalogue available for such a wide region and temporal period. Large-scale relocation of such an extensive catalogue by full-waveform modelling is beyond what can currently be achieved. Nevertheless, for the next manuscript version we will enhance the working catalogue by replacing the ISC locations for the more accurate ones published elsewhere (for a minority of events). This future improvement of the catalogue is not expected to modify much the overall results or conclusions of this work, because they depend on the overall statistics of a large earthquake sample.

Delving into details, some of the reviewer's criticisms about the data refer to an older version of the ISC catalogue, not the newer and improved one used here. The reviewer pointed out the inaccuracies of the ISC location procedures, particularly regarding the hypocentral depths, and cites several research papers noting these limitations, dated from 1983 to 2012. But, as already cited in the manuscript, the ISC Bulletin has been recently completely rebuilt for the period 1964-2010 (Storchak et al., 2020), adding additional earthquakes and relocating hypocenters with the same location procedures used from 2011 onwards (Bondár and Storchak, 2011). The criticisms stated by those papers mentioned in the review refer to the older version of the ISC database, not to the rebuilt and improved one, for which location accuracy is expected to be significantly improved.

For example, currently ISC does not locate seismicity considering only the arrivals of body waves (*P* phases), as mentioned in the review, but the new algorithm also considers diverse *S* phases (Bondár and Storchak, 2011). When ISC cannot relocate an earthquake with

teleseismic data (for example, because it has a low magnitude), it considers as prime the location provided by an authoritative regional agency, built from phase arrivals measured in the records of permanent or local temporary seismic stations (Di Giacomo & Storchak, 2016). It is already a great achievement of ISC to build a global catalogue with such a level of detail.

Mapping variations in D90 or other depth percentiles requires as many events as possible for statistical analysis. That is our reason for using the ISC Bulletin as the most extensive earthquake catalogue available in this wide region and temporal period, down to its magnitude of completeness. We end up using the most reliable events, over 1400 earthquakes for the analysis.

In any case, we agree with the reviewer regarding that earthquake locations obtained with full-waveform modelling (or from dense, local, monitoring surveys whose data was not submitted to ISC) may be more reliable than those routinely published by ISC. For this reason, we will improve the earthquake catalogue used, by replacing the locations of the ISC Bulletin with more accurate ones, if available, obtained by waveform modelling (e.g. Whimpenny & Watson, 2021; Wimpenny, 2022) or local surveys (e.g. Dimate et al., 2003, if we can obtain the data from the authors, as they are not public). The work by Alvarado *et al*. (2005), also suggested in the comments, does not cover our study area. These catalogue modifications, nevertheless, are expected to affect only a minority of events. So the results after this extra effort are expected to be similar to the ones already shown in the manuscript.

Unfortunately, relocation of the whole catalogue used (with earthquakes even down to magnitude *M*=3.5), using full-waveform analysis is simply not feasible. While phase arrivals have been reported to ISC by a plethora of agencies in the region, most records of full waveforms are not public. Gathering them would imply contacting each agency, and digitizing the seismograms one by one if they are not in digital form (which is the common case before the year ~2000, when most seismograms are in analog form, either in magnetic tapes or paper records).

Full-waveform relocation is typically done only for a few, selected, recent events of moderate to large magnitude (usually $M \geq 5$) for which public digital, teleseismic waveform data (e.g. from IRIS stations) are available. This is actually already illustrated by the catalogue of Whimpenny & Watson (2021, updated online), which contains just 12 earthquakes in our study area (with magnitudes $\geq 5.1$, occurred between 1979 and 2021, including those relocated by Whimpenny, (2022).

2. **The method used to compute the seismogenic thickness does not necessarily account for the real depth-distribution of events in each grid area:** The authors say that the D10/D90 statistics are calculated on a 0.1x0.1 degree grid by considering the nearest 20 events to each grid centre as the sample from which D10/D90 are computed. This method appears biased to me. Consider a case where there are 100 events within any particular 0.1x0.1 degree grid area, and the nearest 20 events to the grid centre are all <10 km depth but the remaining 80 are all at 50 km depth. The current method for estimating the D90 would yield a value <10 km for the whole grid area, despite the fact that earthquakes are

occurring down to 50 km. Therefore, I suspect the method used to compute the seismogenic thickness could yield misleading results. This might not be an issue if the typical earthquake spacing in lat/lon is larger than ~0.1x0.1 degree, but the authors should add new results demonstrating that their interpretations are independent of the gridding scale and sample size used in calculating the D10/D90/seismogenic thickness.

**Response:**

The cell size of 0.1 X 0.1 degrees in the sampling grid used is indeed chosen to avoid the sampling problem mentioned by the reviewer. The problem would have arised if a larger grid spacing had been used. Namely, the minimum search radius to the nearest 20 epicentres in our region (the minimum value actually represented in Fig. 9b) is ~6.4 km, as can be checked in the data repository for the manuscript. Meanwhile, the maximum half-width of a 0.1 X 0.1 degree cell is ~5.5 km. So for each cell of the used grid, the area sampled is at least as large as the cell itself, as it should (e.g. Wiemer & Wyss, 2000; González, 2017). We will mention this issue in the next version of the manuscript for justifying the grid spacing used.

The mapping approach followed is to use, for each grid node (cell centre) the minimum number of earthquakes (20) which produces stable results of D10 or D90. This allows mapping these percentiles with the best spatial detail possible.

Using a different sample size would inevitably change the results. For example, using more earthquakes in the sample around each grid node would imply using larger sampling radii. This would always lead to different results of D10 or D90, as would imply mixing shallower and deeper earthquakes from different locations farther away from each other, and the spatial variations of D10 or D90 would tend to be inevitably smoothed out. Using smaller samples would produce the opposite effect (yielding an apparently higher variability of the results, but with higher uncertainty and thus less statistically meaningful).

Using a different sample size is indeed not justified. For sample sizes < 20, the uncertainty of the 10th (D10) or 90th (D90) percentile increases significantly, as will be further commented on below. And, given that the results are already stable for 20 events in each sample (as determined by bootstrap), a sample size > 20 would be arbitrary, and would need to be subjectively chosen by checking the appearance of the resulting maps. This kind of subjective choice has been faced elsewhere when sampling earthquakes spatially at different resolutions (e.g. see Schorlemmer *et al*., 2004). The procedure used here (using the minimum sample required for the particular statistical analysis, for each grid node, e.g., in González, 2017) avoids such a subjective choice. This will be emphasized in the next version of the manuscript.

3. **D10 and D90 may not be the relevant metrics for understanding the absolute depth range of earthquake nucleation:** The D10 and D90 parameters have been developed to study temporal variations in the depth of seismicity in regions with dense earthquake catalogues (e.g. California, Japan; see Rolandone et al., [2004]). These metrics do not seem suitable when studying the depth-extent of seismicity in the ISC catalogue. Firstly, the ISC catalogue is relatively sparse, and so choosing D10/D90 may not be robust as it might vary with longer observation intervals. Secondly, the 10% and 90% cut-off are arbitrary values – the more common and logical definition is that the seismogenic thickness is the depth of

the deepest observed earthquake in an area [like used in Maggi et al., 2000; Jackson et al., 2004]. Could the authors add some explanation as to why they use D10 and D90 and why it is a relevant metric?

I will also suggest that D10 is unlikely to be a robust metric when calculated from the ISC catalogue, because hypocentral locations derived using travel time data have particularly poor depth resolution within the top 10 km of the crust. Weston et al., [2011] and Wimpenny and Watson, [2021] have demonstrated the poor resolution of ISC-EHB event depths in the upper crust by comparing ISC hypocentral depth estimates with both finite-fault slip solutions derived from modelling InSAR data and more accurate earthquake centroid depths from waveform modelling. Both studies found that there can be differences on the order of 5-10 km, and that the ISC systematically overestimates the depth-range of slip in shallow earthquakes.

**Response:**

Non-reliable values of D10

After the reviewer's comments, and given the shallow values that we obtain for D10, we reckon that they may be more affected by the uncertainties of hypocentral depths. Even if the absolute depth uncertainties were similar for deeper and shallower events, the relative uncertainty (= uncertainty / depth) is larger for shallower ones.

To accommodate the reviewer's comment, we plan to remove the D10 analysis from the manuscript and only focus on D90, which is more interesting from the point of view of the thermal model and the application, e.g. to help constrain the maximum depth of the base of crustal seismic sources, instead of discussing the crustal seismogenic thickness.

As replied above, we will also include published, improved depth determinations in the catalogue (albeit, given that they will be available for a minority of events, the overall results are not expected to change much).

Justification of the use of D90 instead of other percentiles

Focusing on D90, it seems necessary to recall that the lower depth limit of seismicity is a transition, rather than a sharp boundary, for two reasons. One is physical: as depth and temperature increase, progressively fewer minerals are able to deform in a stick-slip fashion, instead of plastically (hence the term "brittle-ductile transition"). The other reason is observational: uncertainties in hypocentral depths blur this transition even further.

Indeed, in some publications, as noted by the reviewer, the largest hypocentral depths (which would be the percentile 100, i.e. D100) are considered. However this approximation is largely unstable, as it depends only on the local, largest, extreme value of the depth distribution, which could be particularly unreliable, due to the depth uncertainties (the largest, single depth in the sample, if unreliable, would make D100 useless). In fact, contrary to what the reviewer argues, the use of D100 is even more prone to temporal variations, as new earthquakes might potentially nucleate deeper than previous observations. These are the

reasons for using a lower percentile, such as the 90% (D90), which is by definition more stable than D100.

For a given sample size, the higher the percentile, the less statistically robust it will be. For example, with a sample of 20 depths, it would make sense to quantify the 90% percentile (2 out of the 20 events, i.e. 20*90/100, are expected to be ≥ than it). It would also be possible to calculate the 95% percentile, but it would be less reliable, as it would depend only on the largest value of the sample (1 out of the 20 events, i.e. 20*95/100, is expected to be ≥ than it), and consequently, the bootstrap uncertainties would be larger than those of D90. Calculating, e.g. a meaningful 99% percentile (D99) would require at least 100 events in the sample and can be attempted only if many small earthquakes are well recorded (e.g. in Taiwan, Wu *et al.*, 2017). And calculating a meaningful 100% (D100) percentile would theoretically require an infinite sample (if a larger observation period is used, there will eventually be deeper earthquakes than observed before, changing D100, but not necessarily D90, for example).

In particular, D90 was initially devised for characterizing spatial, rather than temporal, changes in the hypocentral depth distribution (Sibson, 1982, Marone & Scholz, 1988). It is a robust metric, because it can be calculated even with small samples, and it is the most commonly used proxy to map spatial variations of lower seismogenic depths and to compare with thermal parameters (e.g. Tanaka, 2004; Omuralieva et al., 2012).

Moreover, in Probablilistic Seismic Hazard Analysis (PSHA), the use of D90 is very extended as a reasonable measure of the depth of seismicity, or the maximum depth of seismic sources (e.g. U.S. Nuclear Regulatory Commission, 2012, their chapter 5; Bommer *et al.*, 2023).

Following these arguments, we will briefly justify the use of D90 instead of other percentiles, in the next version of the manuscript.

Mapping  D90 with a sparse catalogue

When mapping the or D90 (or D10) values, we have explicitly represented the mapping resolution (Figure 9b), and considered it in the interpretation, so that we do not over-interpret the results. Of course, in the few regions (such as in Japan or Southern California) with catalogues complete down to much lower magnitude thresholds, and with precise hypocentral depths, it is possible to use many more earthquakes and map these percentiles in greater spatial detail. In any case ,it is necessary to quantify the depth distribution of seismicity elsewhere too (not only in the very best monitored regions), and those limitations, which are accounted for, do not lower the merit of our mapping, which is the first done in the analyzed region.

4. The temperature field within the mountains is most likely not in steady state: The authors have assumed that the temperature field throughout NW South America can be modelled as being in steady-state. The time taken for a system to approach thermal steady state is given by the thermal time-constant $\tau = l2/\pi2\kappa$ where $l$ is the thickness of the lithosphere and $\kappa$ is the thermal diffusivity of the lithosphere (~10–100 Ma for normal lithosphere). Given that mountain building will advect heat and move the lithosphere out of thermal

steady-state [England and Thompson, 1984], and that mountain building in the NW Andes has taken place more recently than 10-100 Ma, then I would assume that the temperature field within the NW Andes cannot be modelled as being in thermal steady state. I agree that the steady-state assumption is more reasonable within the forelands.

The heat flow and downhole temperature data the author's use to validate the models come from the top 4 km of the crust, and are mostly located offshore and not within the continental lithosphere, therefore are not a good test of whether the model is representative of the temperature field at depth in Region 1, 2 or 3. To address this comment, the authors need to justify why they think the steady-state assumption is valid for the mid/lower crust.

**Response:**

Steady-state assumption:

Sensitivity studies (Meeßen, 2019) have shown that the transient effects causing the temperature changes in the upper 50 km of a subducting system are not far from equilibrium. Likewise, Rodriguez Piceda et al. (2022) have explored the effects of the non-steady state and came to the same conclusion.

Although it is true that mountain building processes imply that the system might not be fully in equilibrium, it is generally accepted that the first-order thermal field within the lithosphere is driven by conduction (e.g.: Liu et al., 2021). See response to comment Line 77 for more details.

Observational constraints:

We already acknowledged in the manuscript that there are limited heat flow and temperature measurements in the study area. Precisely for this reason, we made the effort to provide the best thermal model considering a fully 3D thermal approach, by integrating all the available data.

Indeed, the sensitivity test we performed with 25 different models (see Text S2 in the supplements) shows that the fit to the few temperature measurements available is already highly sensitive to the thermal parameters we explored. Figure S1 depicts the residual temperature for the different models we tested. The residuals of the first 19 models are particularly large, which implies that there is a limited range of the parameter values which can properly fit the observations.

In any case, even in an area with abundant high-quality temperature and heat flow observations (e.g. Southern California), those will be always restricted to the uppermost kilometers of the crust. See response to comment Section 4.1 for more details.

5. **Lab and nature suggest olivine remains seismogenic up to ~600 degrees – not 600-1000 degrees:** There are two strong lines of evidence that suggest olivine-rich rocks can only remain able to nucleate earthquakes up to temperatures of ~600 degrees Celsius. The first comes from the depths of well-constrained earthquakes within the oceanic lithosphere,

where the temperature structure is well known. Here the maximum centroid depths of large earthquakes can be seen to deepen with the age of the lithosphere and remain consistently shallower than the 600 degree isotherm [McKenzie et al., 2005; Craig et al., 2014]. Secondly, laboratory experiments show that olivine aggregates switch from deforming through stick-slip, to deforming through stable creep, at temperatures equivalent to ~600 degrees at the strain rates expected within the oceanic lithosphere [Boettcher et al., 2007]. The authors cite Scholz [2019] (a textbook) regarding the seismogenic range of temperatures for olivine being 600-1000 degrees, but do not describe the evidence for this. I would suggest that the author's cite the original papers that came to this conclusion, as I have not been able to check where their logic has come from.

**Response:**

The limiting temperature for seismogenesis in mantle forming materials has been a matter of debate. Some authors define a rather strict limit at 600°C (e.g.: McKenzie et al., 2005; Craig et al., 2014), but new evidence suggests that this limit might occur at higher temperatures. For example King and Marone (2012) explored the deformation of olivine fault gouges in the temperature range of 400–1000°C. Their results suggest that the velocity weakening (negative a-b) occurs at temperatures between 600 and 1000°C, depending on the strain rates applied to the sample (Figure 1).

[Figure]

**Figure 1.** Stability parameter, a – b, for different velocity steps: green 1-10µm/s, orange 10-50µm/s and blue 50-1µm/s. (a) Rate-state model. (b) Two state variable approach. Figures taken from Figures 6 and 8 in King and Marone (2012).

Similarly, Ueda et al., (2020) studied the brittle-ductile transition in peridotites based on a fault system developed in the Balmuccia peridotite body. Their results indicate that the ductile-to-brittle transition in the peridotite occurs at ~720 °C. Lastly, Grose and Afonso (2013) studied the evolution of the oceanic lithospheric using more realistic thermal models than McKenzie et al. (2005), yielding a brittle-ductile transition closer to the 700-800°C isotherms depending on the mantle potential temperature (see Figure 10 in Grose and Afonso, 2013). This implies a higher temperature than a purely thermal, simplified approach,

as their workflow takes into account not only hydrothermal circulation, but also P-T-dependent thermal properties.

Regarding the citation of Scholz (2019), we will modify the text adding the original references compiled by the textbook, and will add some additional works pointing to a higher temperature for seismogenesis of mantle forming materials, as described above.

The original references cited by Scholz (2019) are:

King, D. S. H., & Marone, C. (2012). Frictional properties of olivine at high temperature with applications to the strength and dynamics of the oceanic lithosphere. *Journal of Geophysical Research: Solid Earth*, *117*(12), 1–16. https://doi.org/10.1029/2012JB009511

Boettcher, M. S., Hirth, G., & Evans, B. (2007). Olivine friction at the base of oceanic seismogenic zones. *Journal of Geophysical Research: Solid Earth*, *112*(1), 1–13. https://doi.org/10.1029/2006JB004301

**Line-by-Line Comments**

**Line 11:** "Crustal seismogenic thickness" is a misleading term in my view, because the seismogenic layer can include parts of the crust and upper mantle. I would suggest that just the depth extent of earthquakes in the lithosphere is the more logical description of the seismogenic thickness, and the depth most likely to correlate with a "stability transition" from seismogenic to aseismic deformation processes.

**Response:**

We had used the term "crustal seismogenic thickness" because we are targeting the depth distribution of seismicity at crustal levels, which implies removing from the analysis the sub-crustal seismicity (as done by others before, e.g.: Wu *et al*., 2017 in Taiwan, or Tsuda *et al*. 2019 in Japan).

Our study area includes two subducting flat-slabs, and therefore, there is intermediate to deep seismicity associated with them (see Figure 2 below). In order to disregard events that might be associated with the slab dynamics, we used the Moho from the GEMMA model (Reguzzoni and Sampietro, 2015) to filter out these events.

As can be seen in Figure 2, if we had not disregarded these deep events, the computed seismogenic thickness would reach up to 200 km (or more) in some parts of the study area, which is unrealistic for the purposes of analyzing crustal seismogenesis.

Nevertheless, in the next manuscript version, following the reviewer's comments, we will not analyze D10 and will not use the term "crustal seismogenic thickness". We will rather mention, as Wu et al. (2017) the "seismogenic depths of crustal earthquakes".

[Figure]

**Figure 2.** Seismicity in the Caribbean and northwestern South America (ISC, 2020).

**Line 18:** "Potential temperature" means specifically the temperature at which a rock would be if moved from a particular depth along an adiabat, which I think is different from what the authors mean here. I think they mean just the possible absolute temperature, and would recommend removing the word "potential".

**Response:** Thanks for pointing this out. We will modify the text accordingly.

**Line 32:** Why is the crustal seismogenic thickness not just the layer where all earthquakes occur? I'm not sure why it would be defined as where 'most', but not all, earthquakes occur.

**Response:** Please see response to comment regarding Line 11.

**Line 39:** Is 90% of events based on a measure of statistical significance, or is it just defined as an arbitrary value?

**Response:** Please see response to comment #3.

**Line 46-48:** I would argue that it is generally accepted that earthquakes mostly nucleate within the crust at temperature <350 degrees Celsius, but can nucleate in lower crustal rocks at temperatures up to 600 degrees Celsius within certain areas where the lower crust might be dry [see Jackson et al., 2008; Craig et al., 2011; Sloan et al., 2011; Craig et al., 2012].

Examples of lower-crustal seismicity are now extensive, and there are plenty within South America [Assumpcao et al., 1992; Devlin et al., 2012; Wimpenny 2022].

**Response:** Please see response to comment #5.

**Line 73**: Typo "as the extent of the CST" not "extend".

**Response:** Fixed, thanks.

**Line 77:** It is not clear to me why flat slab subduction means that you can assume thermal steady state. If anything, I would assume that flat slab subduction would suggest that you need to account for the heat advection in the modelling of the temperature field, because the slab has not always been flat and therefore the temperature boundary condition on the base of the lithosphere has changed through time. Please could the authors elaborate on why a flat slab configuration allows them to make an assumption of thermal steady state?

**Response:**

The present-day flat slab geometry has been established since about 6 Ma, when the Nazca tear developed separating the north (flat) and south (step) segments (Wagner et al., 2017). As a result, the volcanic activity has ceased in the continental crust of the overriding plate of the north segment, which spatially corresponds to our study area. This allows us to consider that the propagation of heat within the crust is mainly driven by conduction (e.g.: Liu et al., 2021).

Geodynamic models (e.g.: Schellart and Strak, 2021; Currie and Copeland, 2022) suggest that the subducting velocities are potentially reduced during the evolution from step to flab slab (see Fig 16 in Schellart and Strak, 2021 as an example -Figure 3 below). Considering that we are targeting the thermal field of the uppermost 75 km of the lithosphere, and that our applications are limited to crustal realms, we believe that a steady-state assumption can be applied in the study area to have a first-order estimate on the 3D feedback of the system heterogeneities and their imprint in the resulting thermal field. Moreover, given that the timeframe of the earthquakes we are studying is instantaneous in the geological timescales, our goal is to take a "snapshot" of the present-day thermal configuration considering the 3D system heterogeneities at crustal scales.

[Figure]

**Figure 3.** Resulting velocity field during the evolution of slab flattening (Schellart and Strak, 2021).

**Line 92:** What is meant by "large uncertainties" in the Moho? Could the authors give quantitative estimates of the typical Moho uncertainty?

**Response:**

The Moho uncertainty map is provided in the supplementary Figure S8, as computed by the authors of the GEMMA model (Reguzzoni & Sampietro, 2015) used in this research (see line 500 of published preprint).

**Figure 1:** There are a number of references to different geographic places (e.g. Panama-Chaco Block) in the text that are not defined on Figure 1. Please could the authors add these place names to the figure to help the reader.

**Response:** Thank you. Yes, we will carefully check this issue and improve the figures.

**Line 242-245:** The authors "disregard earthquakes below the Moho…", and use only events above the Moho to compute "… the upper and lower stability transitions". By ignoring earthquakes below the Moho, then the maximum depth of earthquakes the authors calculate is limited by the crustal thickness, and doesn't necessarily correspond to any "stability transition" or mechanical condition in the lithosphere. I would recommend changing the phrasing the authors are using to describe what they are actually measuring, which is more like the fraction of the crust that is seismogenic.

**Response:** Please see response to comment regarding Line 11.

**Line 263:** Why is removing events that are below the magnitude of completeness (Mc) relevant for this analysis, because the authors are not necessarily studying earthquake frequency or moment release through time? Small events below Mc could still be useful in defining the depth distribution of seismicity.

**Response:**

Small events below Mc are not useful for defining the depth distribution of seismicity, because their depths are biased. The deeper an earthquake is, the less likely it is to detect, that is, Mc increases with depth (e.g. Fig. 5 of Schorlemmer *et al*., 2010). If we used an incomplete sample, considering earthquakes below Mc, small deep earthquakes would be preferentially missing and the overall apparent depth distribution would be biased towards shallower values. We will mention these issues explicitly in the next manuscript version.

**Line 274-276:** The bootstrapping method of estimating the uncertainties in D10/D90 yield what I would consider unrealistically small values of <1 km. This is presumably because the 'formal' uncertainties in earthquake depths included in the ISC catalogue that are propagated through the bootstrapping are not realistic representations of the true depth errors. For example, comparing earthquake hypocentral depths in the ISC-EHB catalogue with more accurate centroid depths calculated from body-waveform modelling yields a mean difference on the order of ~5±10 km with errors of up to 50 km [Wimpenny and Watson, 2021]. Therefore, I expect a more realistic uncertainty is at least 5-10 km.

In addition, the inference that "low uncertainties [estimating from bootstrapping] indicate that using 20 earthquakes for each node is [sic] already reliable… " is not necessarily true in my view. If you used a sample size of 1 earthquake, then you would get low uncertainties, but the uncertainty estimate is not necessarily robust in that changing the sample used would give a different answer.

The way to check whether 20 events is a reasonable sample size for giving robust results would be to make sure that increasing the sample size does not alter the conclusions (even if the spatial resolution has to decrease as a product of the greater sample size).

**Response:**

Despite the efforts taken at quantifying the uncertainties in our analysis, certainly, non-formal depth errors, (such as systematic errors due to the velocity model used in the location), cannot be directly quantified. As mentioned earlier, we will consider better earthquake locations, if available, to try to improve the reliability of the results. We will also discuss the additional uncertainties in the next manuscript version.

Low uncertainties estimated from bootstrapping indeed show that the results are stable. If the sample size were reduced, the results of the percentiles (e.g. D90) would be unstable: removing one earthquake from the sample and substituting it with another (i.e. bootstrapping the sample) would significantly change the percentile. The only exception would be the very unlikely case that all the earthquakes in each sample had the same or almost identical depths, so removing one or another would not change the percentile.

We find the comment about using a hypothetical sample of 1 earthquake inadequate, given that the very minimum sample to calculate a 10% or 90% percentile would be 10 cases. Repeating the calculations with samples of 10 earthquakes, the bootstrap uncertainties increase significantly, so such small samples are not as reliable as the value of 20 used here.

As discussed before, modifying the sample size would actually not be justified, and the results would change. Fewer than 20 events would yield more spatially variable but more uncertain D10 or D90 results, and more than 20 events would smooth out spatial variations, as earthquakes farther from each grid node would have to be considered in the corresponding sample.

**Line 292:** I recommend quoting the misfit as 5 degrees, not 4.99 degrees, as the latter makes it sound like the model and data have precision to two decimal places, which is unlikely.

**Response:** We agree, thanks. We will modify both the text and the figure accordingly.

**Section 4.1:** A more general comment is that the thermal model validation relies on comparing measurements of the temperature in the very shallowest part of the crust in areas that are mostly offshore and do not overlap with Regions 1, 2, or 3. Could the authors add some text explaining why they think that validating the model using measurements from outside of the region of interest means that it will be valid in the regions of interest, which may itself have a different geological history and therefore different temperature structure (e.g. the effects of mountain building on the geotherms).

**Response:**

See response to comment #4. Even if the temperature observations at depth are not located at those regions, they are able to constrain the thermal model parameters. Choosing other values for the thermal parameters results in a worse fit, as explored in the sensitivity analysis performed (Text S2).

Thinking about the study area as a 3D fully coupled system, we believe that being able to reproduce the observations is a good sign of the model first-order robustness for the applications we currently consider in our manuscript. A main reason for our chain of arguments is that we not only model temperatures (for example by interpolating those) but consider the structural complexity of the heterogeneous distribution of thermal properties together with the physics of conductive heat transport to predict the 3D thermal field that is consistent with the few observed temperatures. The 3D heterogeneity in physical properties is constrained by other methods, such as seismic data or 3D gravity modelling, as certain seismic velocity- density pairs can be interpreted as lithologies, from where we can conclude on the lithology-dependent thermal properties. In fact, as expected, we do get different temperature structures all over the study area, as the model integrates the complexity of the region.

We should note that the procedure followed, of building a 3D model integrating all available information, is a major improvement with respect to earlier approaches. Moreover, if eventually more data of heat flow or thermal measurements become available, they could be integrated in future models following this philosophy.

**Line 317:** Why is the "regional seismogenic zone" defined as 20 km? This seems arbitrary to me, and different from the seismogenic thickness quoted elsewhere.

**Response:**

The histogram of the hypocentral depths of the selected earthquake catalogue (Figure 7 c) shows that the majority of earthquakes in the study area nucleate at less than 20 km depth. The computed D90 of the entire catalogue is 20.9 km (see line 457). Therefore, we selected the temperature difference between the surface and 20 km as a proxy for estimating the geothermal gradient in the seismogenic zone at regional scale.

**Line 326-329:** I would recommend re-wording this sentence, as on first reading I thought the authors were suggesting that areas with high geothermal gradients were areas with a strongly coupled crust and lithospheric mantle, though they actually mean it is the other way around. Maybe just split it up into a few shorter sentences for clarity.

**Response:** Thank you, we will improve this sentence.

**Line 348-354:** This is an interesting test, and it is physically intuitive that areas of higher geothermal gradient will have more seismicity because the lithosphere is hotter, therefore weaker and deforms more rapidly in those areas. However, I would be interested to know whether the thermal modelling adds any more useful information on where earthquakes are expected to occur relative to direct surface observables such as topography or fault traces.

**Response:**

We did not explore the skillfulness of other geodynamic variables in our study area because testing this systematically would require a whole new paper (as done, e.g. by Becker et al., 2015). It would require dedicated modelling efforts, to check if, for example, strain rates and rates of topography change are skillful geodynamic variables at forecasting the spatial distribution of seismicity, as in the region analyzed by Becker et al. (2015). It is important to note that it is not only the thermal gradient per se, but the contrast between domains of differing gradients that matters. We will add a sentence on these points to the new manuscript.

**Line 407:** Please cite the original source for the constraint on the seismogenic temperatures for olivine gouge as 600-1000 degrees.

**Response:**

The original sources cited by Scholz (2019) are:

King, D. S. H., & Marone, C. (2012). Frictional properties of olivine at high temperature with applications to the strength and dynamics of the oceanic lithosphere. *Journal of Geophysical Research: Solid Earth*, *117*(12), 1–16. https://doi.org/10.1029/2012JB009511

Boettcher, M. S., Hirth, G., & Evans, B. (2007). Olivine friction at the base of oceanic seismogenic zones. *Journal of Geophysical Research: Solid Earth*, *112*(1), 1–13. https://doi.org/10.1029/2006JB004301

Nevertheless, we will modify the text to make it clear that the upper boundary of this temperature range is highly debated, including additional references we have been citing throughout this document.

**Line 415:** Why does shallow earthquake hypocentres mean faults are 'not well developed', and what is meant by 'not well developed'? Do you mean 'immature' in that they haven't accommodated a significant amount of relative motion? Is there any evidence from natural fault zones that the hypocentral distribution of earthquakes differs with the 'development' of the fault, or are these inferences based off of laboratory experiments?

**Response:**

Considering the uncertainties discussed above for shallow hypocentres, we will remove this comment. This interpretation was explained by Scholz (2019, p. 149 and references therein), based on the seismogenic depths observed in different faults worldwide:

*"The presence of this upper transition was first proposed by Scholz, Wyss, and Smith (1969) and is based on the observation that there are both upper and lower seismicity cutoffs on well-developed faults (Figure 3.45). The upper transition is most likely due to the presence of unconsolidated and phyllosilicate-rich gouge in the fault zone at shallow depth – material that is generally found to be velocity strengthening (Section 2.3.4). The observation of an upper cutoff in seismicity at about 2–4 km seems to be limited to well-developed fault zones where a thick gouge layer is likely to be present (Marone and Scholz, 1988). In regions where there are no well-developed faults, such as the central Adirondacks of New York (Blue Mountain Lake), Miramichi, New Brunswick, and Meckering, Australia, earthquakes occur up to very shallow depths (Figure 3.45). In contrast, in the Imperial Valley, California, earthquakes occur only deeper than 5 km, below an unusually thick sequence of unconsolidated sediments (Doser and Kanamori, 1986)." [Underlining is ours.]*

**Line 417-419:** This sentence captures much of my concern regarding the results of this study. The authors state: "However, despite relying only on the best located earthquakes (see Sect. 3.2.1), large errors in the hypocentral depths still remain (up to 30 km, see Fig. S5), and should be considered in the analysis of our results." I agree, but it's not clear to me how the authors have considered this in their own analysis of their results. In my view, the best way to mitigate this issue is to use more accurate methods for determining earthquake depths, which are more time consuming but will yield more robust results.

**Response:**

As we have already discussed above, it is not possible to avoid those depth uncertainties, as a complete relocation of the catalogue is beyond reach. We already considered the measurable, formal, depth uncertainties (which are, on average, 7 km in the selected catalogue, see Figure S7). And we will incorporate better depth determinations if available and take care, as we already did before, of not interpreting the results beyond their limitations. Future updates of the earthquake catalogue will eventually enable successive improvements of this kind of analysis in the region, for which our present manuscript is proposed as a necessary first step.

**Line 443:** What do the authors mean by "D90 depths of almost 35 km … are in agreement with the crustal-scale structure that these systems likely represent"? Faults can be 'crustal scale' but be aseismic still presumably (e.g. segments of the San Andreas Fault).

**Response:**

What we mean is rather obvious: D90 at the fault location is almost 35 km, so its seismicity is distributed along the crust down to such depths, and the fault seems to be a crustal-scale feature. If D90 were, instead, e.g. 5 km, we could not argue that the fault is of crustal scale, as it might be shallow only.

We find this remark important because there are no seismic profiles or detailed tomographic models which could provide alternative evidence of the fault extent at depth.

**Line 445:** "The D90 values … are clearly bounded by major faults" – I find it hard to see these patterns reflected in Figure 8, mainly because there are a lot of faults that do not seem to bound changes in the colour patterns. Maybe highlight the specific faults thought to bound the changes in D90 in the Figure?

**Response:** Thank you; we will improve the figure in the next version of the manuscript.

**Line 560-565:** There are a few points where the authors mention the diverse geology of the northern Andes and how mafic/ultramafic terranes at depth might account for the deep seismicity. These arguments would benefit significantly from a simplified geological map of the study that highlights where these ultramafic/mafic terranes are based on surface outcrop and display these data relative to the seismicity and D90 maps.

**Response:**

Thank you for this recommendation, we will consider it for the next version of the manuscript.

––––––––––

**New references cited (not already included in the manuscript)**

Alvarado, P., Beck, S., Zandt, G., Araujo, M. & Triep, E. (2005): Crustal deformation in the south-central Andes backarc terranes as viewed from regional broad-band seismic waveform modelling. *Geophysical Journal International*, 163: 580-598. doi:10.1111/j.1365-246X.2005.02759.x

Boettcher, M. S., Hirth, G., & Evans, B. (2007). Olivine friction at the base of oceanic seismogenic zones. *Journal of Geophysical Research: Solid Earth*, *112*(1), 1–13. doi:10.1029/2006JB004301

Bommer, J.J.; Ake; J.P. & Munson, C.G. (2023): Seismic source zones for site-specific probabilistic seismic hazard analysis: The very real questions raised by virtual fault ruptures. *Seismological Reseach Letters*, 94, 1900–1911, doi:10.1785/0220230037

Currie, C. A., & Copeland, P. (2022). Numerical models of Farallon plate subduction: Creating and removing a flat slab. *Geosphere*, 18(2), 476–502. doi:10.1130/GES02393.1

Grose, C. J., & Afonso, J. C. (2013). Comprehensive plate models for the thermal evolution of oceanic lithosphere. Geochemistry, Geophysics, Geosystems, 14(9), 3751–3778. https://doi.org/10.1002/ggge.20232

King, D. S. H., & Marone, C. (2012). Frictional properties of olivine at high temperature with applications to the strength and dynamics of the oceanic lithosphere. *Journal of Geophysical Research: Solid Earth*, *117*(12), 1–16. doi.org/10.1029/2012JB009511

Liu, X., Currie, C. A., & Wagner, L. S. (2021). Cooling of the continental plate during flat-slab subduction. *Geosphere*, 18(1), 49–68. doi.org/10.1130/GES02402.1

McKenzie, D., Jackson, J., & Priestley, K. (2005). Thermal structure of oceanic and continental lithosphere. *Earth and Planetary Science Letters*, 233(3–4), 337–349. doi.org/10.1016/j.epsl.2005.02.005

Meeßen, C. (2019). The thermal and rheological state of the Northern Argentinian foreland basins, PhD Thesis, Potsdam: Universität Potsdam, 151 p. https://doi.org/10.25932/publishup-43994

Mora-Bohórquez, J. A., Ibánez-Mejia, M., Oncken, O., de Freitas, M., Vélez, V., Mesa, A., & Serna, L. (2017). Structure and age of the Lower Magdalena Valley basin basement, northern Colombia: New reflection-seismic and U-Pb-Hf insights into the termination of the central Andes against the Caribbean basin. *Journal of South American Earth Sciences*, 74, 1–26. doi.org/10.1016/j.jsames.2017.01.001

Omuralieva, A.M.; Hasegawa, A.; Matsuzawa, T.; Nakajima, J. & Okada, T. (2012): Lateral variation of the cutoff depth of shallow earthquakes beneath the Japan Islands and its implications for seismogenesis. *Tectonophysics*, 518–521, 93-105. doi:10.1016/j.tecto.2011.11.013

Rodriguez Piceda, C., Scheck-Wenderoth, M., Bott, J., Gomez Dacal, M.L., Cacace, M., Pons, M., Prezzi, C., Strecker, M. (2022). Controls of the Lithospheric Thermal Field of an Ocean-Continent Subduction Zone: The Southern Central Andes. *Lithosphere*. 2022 (1): 2237272. doi: https://doi.org/10.2113/2022/2237272

Schellart, W. P., & Strak, V. (2021). Geodynamic models of short-lived, long-lived and periodic flat slab subduction. *Geophysical Journal International*, 226(3), 1517–1541. doi.org/10.1093/gji/ggab126

Schorlemmer, D., Wiemer, S. & Wyss, M. (2004), Earthquake statistics at Parkfield: 1. Stationarity of *b* values, *J. Geophys. Res.*, 109, B12307, doi:10.1029/2004JB003234.

Schorlemmer, D.; Mele, F. & Marzocchi, W. (2010): A completeness analysis of the National Seismic Network of Italy. *Journal of Geophysiscal Research, Solid Earth*, 115, B04308, doi:10.1029/2008JB006097.

Tanaka A. (2004). Geothermal gradient and heat flow data in and around Japan (II): Crustal thermal structure and its relationship to seismogenic layer. *Earth, Planets, and Space* 56, 1195-1199. doi:10.1186/BF03353340

Ueda, T., Obata, M., Ozawa, K., & Shimizu, I. (2020). The Ductile-to-Brittle Transition Recorded in the Balmuccia Peridotite Body, Italy: Ambient Temperature for the Onset of Seismic Rupture in Mantle Rocks. *Journal of Geophysical Research: Solid Earth*, 125(2). doi.org/10.1029/2019JB017385

U.S. Nuclear Regulatory Commission (2012): *Central and Eastern United States Seismic Source Characterization for Nuclear Facilities.* Technical Report. EPRI, Palo Alto, CA, U.S. DOE, and U.S. NRC. http://www.ceus-ssc.com

Wagner, L. S., Jaramillo, J. S., Ramírez-Hoyos, L. F., Monsalve, G., Cardona, A., & Becker, T. W. (2017). Transient slab flattening beneath Colombia. *Geophysical Research Letters*, 44(13), 6616–6623. doi.org/10.1002/2017GL073981

Wimpenny, S. (2022): Weak, seismogenic faults inherited from Mesozoic rifts control mountain building in the Andean foreland. *Geochemistry, Geophysics, Geosystems*, 23, e2021GC010270. doi:10.1029/2021GC010270

Wimpenny, S. & Watson, C.S. (2020): gWFM: A global catalog of moderate-magnitude earthquakes studied using teleseismic body waves. *Seismological Research Letters*, 92 (1), 212–226. doi:10.1785/0220200218

---

## Author Response (AR1)

Barcelona, Spain
October 18[th], 2023

Dear Editor,

We would be very grateful if you could consider our manuscript "**Thermal structure of the southern Caribbean and NW South America: implications for seismogenesis**" for publication in Solid Earth, revised to tackle all the issues raised by the reviewers.

We sincerely appreciate the extra time allotted to us by the Editorial Board for this revision, as it allowed accommodating the work pace to the health issues related to the pregnancy of the first author.

Also, we are very thankful to Dr. Wimpenny and the anonymous Reviewer #2 for their feedback, which helped us to improve the manuscript.

The lithospheric-scale 3D thermal model initially presented remains unchanged in this new manuscript version. Nevertheless, we have better strengthened and explained the basis for its assumptions and improved and clarified its description. We have also better emphasized how it fits all available observations within their uncertainties.

In contrast, to tackle the suggestions raised by Dr. Wimpenny, we have re-compiled the earthquake catalogue, gathering, from a variety of published sources, earthquake locations more accurate than those previously used. This has implied remaking all the calculations, plots and maps dealing with earthquake information. The new results on this regards are similar to the earlier ones, but show clearer patterns, thanks to the enlargement of the earthquake database and its higher precision.

Also, after the precautions noted by Dr. Wimpenny regarding the interpretation of shallow seismicity, we have dismissed our initial calculation of the D10 percentile (a proxy to the upper, shallow, depth limit of seismogenesis). The title, abstract and main text have been modified to accommodate this and all the other comments made by the reviewers.

The manuscript, as before, is accompanied by supplementary electronic materials, updated with new figures. All original data and results have been published in an updated, separate public data repository available online, to enhance the reproducibility of the work. A point-by-point reply to the reviewers is also included.

We hope that you will find the submission in good order. Thank you very much for your consideration and best regards,

Ángela M. Gómez, also on behalf of the co-authors.

**Author's response to review by Dr. Sam Wimpenny**

We sincerely thank Dr. Wimpenny for his prompt review and insightful comments. We found them very useful, as they have helped us to improve the manuscript, so his detailed feedback is appreciated.

Please find below a point-by-point reply to the issues raised by him, updated from our previous one, according to the changes made in the manuscript. We hope that this new manuscript version supports in more detail the analysis performed and better frames the contribution that the paper makes to the field.

References not already cited in the new manuscript are listed in a separate section at the end of this document.

**Manuscript overview**

**Comment:** …"The authors also find that there is a strong correlation between areas of elevated geothermal gradient and seismicity, but do not provide a physical interpretation for the correlation."

**Response:** Thank you for pointing this out. We added this physical interpretation in section 4.2, noting that seismicity is almost absent in cold lithospheric areas such as the Guyana craton and the Caribbean Large Igneous Plateau. Such a correlation evidences that, in these places, the crust and lithospheric mantle may be strongly coupled, implying that the differential stress in these regions might not be high enough to deform the crust in a brittle regime.

**Comment:** A criticism I have with this manuscript is that it has mostly overlooked the extensive literature on the relationship between earthquake depths, geology, and temperature within the continents that is directly relevant to the arguments they present [Jackson 2002; Jackson et al., 2004; McKenzie et al., 2005; Jackson et al., 2008; Sloan et al., 2011; Craig et al., 2012]. I recommend the authors more explicitly address how their conclusions advance on the findings made in the aforementioned papers, as this will help contextualise the contribution that this paper makes to the field.

**Response:**

We have added an extended text about this issue in the introduction, explicitly citing a broader expanse of the literature, as suggested.

Our findings are not at odds with earlier ones, but are based on a much more detailed thermal modelling than typically achieved. For example, the references cited by the reviewer base their temperature estimates inside the Earth on 1-D or 2-D thermal models, using simplified lithospheric structures, instead of more realistic 3D models as implemented in our work. This is now mentioned in the main text.

Please see also our response to comment #5, as it is related to this issue, too.

**General comments**

1. **Inaccuracies in the ISC earthquake hypocentral depths limit the earthquake depth analyses presented:** The authors use the ISC catalogue's hypocentral depth estimates to map out the depth distribution of seismicity. The ISC locates earthquakes using the reported travel times of body waves, and for events less than ~40 km deep the ISC location procedure can be incorrect by tens of kilometers in depth. These errors are not necessarily reflected in the formal uncertainties in the catalogue, which is a well-known limitation associated with using the ISC earthquake catalogue to study shallow seismicity [Maggi et al., 2000; Chen et al., 2009; Weston et al., 2011] and is something the authors acknowledge. Most studies use waveform modelling of the earthquakes to accurately determine earthquake depths and thereby draw robust conclusions regarding the depth of seismicity in the continents [see examples for South America in: Suarez et al., 1983; Chinn and Isacks 1983; Devlin et al., 2012]. Without accurate earthquake depth estimates, it is unclear to me to what extent variability in the calculated seismogenic thickness, D10 or D90 represent errors in the earthquake depths in the ISC catalogue, or real spatial variability in the depth of earthquake generation. To address this comment, I would recommend that the authors re-analyse the depths of earthquakes in their study area using waveform modelling of teleseismic [e.g. Devlin et al., 2012; Wimpenny 2022] or regional [e.g. Alvarado et al., 2005] seismic data. Alternatively, if the author's believe that the depths of crustal earthquakes in the ISC catalogue are accurate enough for their purposes, then they need to demonstrate this by comparing the ISC event depths with an independent source of earthquake depths (e.g. from local seismic deployment), because in most other settings the ISC event depths are not accurate enough for these purposes.

**Response:**

Trying to integrate the reviewer's suggestions as much as possible, for this new manuscript, we have improved the earthquake catalogue (compiling it from a variety of public data sources, trying to gather the best published location for each event). We also remade all the analyses, maps and figures which dealt with earthquake data.

We agree with the reviewer regarding that earthquake locations obtained with full-waveform modelling (or from dense, local, monitoring surveys) may be more reliable than those routinely published by ISC.

Moreover, the previous version of the manuscript used the prime events reported in the ISC Bulleting, which are not the ones with better depth determinations provided by the ISC-EHB dataset.

Mapping variations in D90 or other depth percentiles, or considering the whole depth distribution of regional seismicity, requires as many events as possible for statistical analysis. That is why we have to rely on a regional catalogue, using as many well-located earthquakes as possible above the magnitude of completeness (on the order of thousands).

As mentioned in our initial reply, full-waveform relocation of the whole dataset cannot be presently achieved, as it can be typically done only for a few, selected, recent events of moderate to large magnitude (usually $M \geq 5$) for which public digital, teleseismic waveform data (e.g. from IRIS stations) are available. This is actually already illustrated by the catalogue of Whimpenny & Watson (2021, updated online), which contains just 12 earthquakes in our study area (with magnitudes ≥ 5.1, occurred between 1979 and 2021, including those relocated by Whimpenny, (2022).

The overall patterns in the results with the newly improved catalogue (such as hypocentral temperatures or D90 mapping) are similar as the ones initially found, evidencing that they are robust. Our new results, however, are clearer and more detailed than before, thanks to the catalogue improvements (both in location precision and in the larger number of events included).

The **catalogue improvements** are as follows (further details are provided in section 3.2.1):

Extended period of analysis: The catalogue has been extended over a year; before it covered until March 2020, while now it covers until June 2021 (the last month revised in the ISC Bulletin at the time of writing).

Selection of best-located events from published sources: For each event, the location, with the most reliable depth was selected among a variety of data sources (instead of relying only on the ISC Bulletin). The following order of preference was used (as described in the new manuscript):

1) The gWFM database (Wimpenny and Watson, 2020), based on synthetic body-waveform modeling, and updated to version 1.2, which includes earthquake locations calculated in the region by Wimpenny (2022) and Wimpenny et al. (2018);

2) Locations calculated by full-waveform modelling (with the ISOLA code; Sokos and Zahradnik, 2008) using records obtained at regional or local distances) by the Colombian Geological Survey (Dionicio et al., 2023; Servicio Geológico Colombiano, 2023) and by Quintero et al. (2023);

3) A high-precision hypocentral relocation for the 2008 Quetame mainshock by Dicelis et al. (2016);

4) Locations with free (not fixed) hypocentral depth from the ISC-EHB dataset (Engdahl et al., 2020; Weston et al., 2018), which is compiled and curated by the International Seismological Centre (2023a); and,

5) The prime locations reported in the reviewed ISC Bulletin (International Seismological Centre, 2023b).

Note that the ISC Bulletin (Storchak et al., 2020) and ISC-EHB datasets (Engdahl et al., 2020; Weston et al., 2018) have been recently rebuilt, in order to alleviate the earlier location issues mentioned in the literature cited by the reviewer.

We also explored other data sources, which eventually could not be taken into account. For example, the work by Alvarado *et al*. (2005), suggested in the reviewer's comments, does not cover our study area. Also, we contacted the authors of publications dealing with several relocated earthquake series, but they declined to provide the requested data.

Magnitude selection – An improved scheme for choosing the preferred magnitude for each event was applied, to account for published, improved moment magnitude estimates.

Stricter criteria for discarding poorly located earthquakes – We have chosen to calculate hypocentral temperatures only for earthquakes with reported, formal, hypocentral depth errors ≤ 15 km, instead of ≤ 30 km as before.

Enlarged dataset: Despite the stricter choice just mentioned, thanks to the catalogue improvements, there are now more selected crustal earthquakes (almost 2000) than before (~1400) above the magnitude of completeness.

2. **The method used to compute the seismogenic thickness does not necessarily account for the real depth-distribution of events in each grid area:** The authors say that the D10/D90 statistics are calculated on a 0.1x0.1 degree grid by considering the nearest 20 events to each grid centre as the sample from which D10/D90 are computed. This method appears biased to me. Consider a case where there are 100 events within any particular 0.1x0.1 degree grid area, and the nearest 20 events to the grid centre are all <10 km depth but the remaining 80 are all at 50 km depth. The current method for estimating the D90 would yield a value <10 km for the whole grid area, despite the fact that earthquakes are occurring down to 50 km. Therefore, I suspect the method used to compute the seismogenic thickness could yield misleading results. This might not be an issue if the typical earthquake spacing in lat/lon is larger than ~0.1x0.1 degree, but the authors should add new results demonstrating that their interpretations are independent of the gridding scale and sample size used in calculating the D10/D90/seismogenic thickness.

**Response:**

As mentioned in our initial reply, the grid cell (pixel) size of 0.1 X 0.1 degrees in the sampling grid used was indeed chosen to avoid the sampling problem mentioned by the reviewer.

Now that the new catalogue contains more earthquakes, there are exactly only two pixels in the D90 map (of a total of ~7500) were that problem arose (the pixels contained more than 20 earthquakes each). D10 is no longer computed (see reply to next comment).

We have slightly modified the spatial sampling of earthquakes in order to avoid this issue. Namely, the resolution radius (distance to the furthest earthquake considered in each sample) was set to a minimum of 5 km (in order to cover at least one grid cell of the model). If there are ≥ 20 earthquakes within this distance, all of them are taken into account in the sample for that grid node.

As discussed in our initial reply, using a different sample size is indeed not justified. For sample sizes < 20, the uncertainty of the 90th percentile (D90) increases significantly. And, given that the results are already stable for 20 events in each sample (as determined by bootstrap, now combined with Monte Carlo as explained below), a sample size > 20 would be arbitrary, and would need to be subjectively chosen by checking the appearance of the resulting maps. This kind of subjective choice has been faced elsewhere when sampling earthquakes spatially at different resolutions (e.g. see Schorlemmer *et al.*, 2004). The procedure used here (using the minimum sample required for the particular statistical analysis, for each grid node, e.g., in González, 2017) avoids such a subjective choice.

3. **D10 and D90 may not be the relevant metrics for understanding the absolute depth range of earthquake nucleation:** The D10 and D90 parameters have been developed to study temporal variations in the depth of seismicity in regions with dense earthquake catalogues (e.g. California, Japan; see Rolandone et al., [2004]). These metrics do not seem suitable when studying the depth-extent of seismicity in the ISC catalogue. Firstly, the ISC catalogue is relatively sparse, and so choosing D10/D90 may not be robust as it might vary with longer observation intervals. Secondly, the 10% and 90% cut-off are arbitrary values – the more common and logical definition is that the seismogenic thickness is the depth of the deepest observed earthquake in an area [like used in Maggi et al., 2000; Jackson et al., 2004]. Could the authors add some explanation as to why they use D10 and D90 and why it is a relevant metric?

   I will also suggest that D10 is unlikely to be a robust metric when calculated from the ISC catalogue, because hypocentral locations derived using travel time data have particularly poor depth resolution within the top 10 km of the crust. Weston et al., [2011] and Wimpenny and Watson, [2021] have demonstrated the poor resolution of

ISC-EHB event depths in the upper crust by comparing ISC hypocentral depth estimates with both finite-fault slip solutions derived from modelling InSAR data and more accurate earthquake centroid depths from waveform modelling. Both studies found that there can be differences on the order of 5-10 km, and that the ISC systematically overestimates the depth-range of slip in shallow earthquakes.

**Response:**

Non-reliable values of D10

After the reviewer's comments, as proposed in our initial reply, we have disregarded the calculation of D10. We indeed agree that the depth uncertainties (at least the formal ones, which can be systematically quantified) are larger for shallower earthquakes than for deeper ones, as shown in the new supplementary Figure S5.

Justification of the use of D90 instead of other percentiles

Following our initial reply, we have added to the text a discussion on the statistical basis of focusing on D90 instead of using higher percentiles (which may be less statistically robust, and less stable eventually, as new earthquakes keep being recorded). Particularly, using the deepest observed earthquake, D100 (especially with a small sample), would be unstable, because, eventually, even deeper earthquakes could be recorded.

As initially mentioned in our previous reply, D90 was initially devised for characterizing spatial, rather than temporal, changes in the hypocentral depth distribution (Sibson, 1982, Marone & Scholz, 1988). It is a robust metric, because it can be calculated even with small samples, and it is the most commonly used proxy to map spatial variations of crustal seismogenic depths and to compare with thermal parameters (e.g. Tanaka, 2004; Omuralieva et al., 2012).

Moreover, in Probablilistic Seismic Hazard Analysis (PSHA), the use of D90 is extended as a reasonable measure of the depth of seismicity, or the maximum depth of seismic sources (e.g. U.S. Nuclear Regulatory Commission, 2012, their chapter 5; Bommer *et al*., 2023).

Addition of D95 as supplementary information

Apart from D90, we have now calculated D95, as mentioned in the main text and shown in the supplementary material. Both D90 and D95 are also provided in the public dataset published separately (Gómez-García et al., 2023).

The overall spatial trends of D90 are very similar to those of D95. Albeit, by definition, D95 is deeper than D90, the difference between both is only typically between 1.0 to 1.5 standard deviations to each other (considering their respective uncertainties).

Despite D90 and D95 have similar formal uncertainties (Fig. S9), D90 is preferred in our case because it would be temporally more stable than D95. Future earthquakes will have some chance of being deeper than those already recorded. Given the sample size used, this would affect D95 to a much larger extent than D90, as explained in the main text (section 3.2.2).

Mapping D90 with a sparse catalogue

As noted in our initial reply, when mapping the D90 (or D95) values, we have explicitly represented the mapping resolution (Figs. 8 and S8, respectively), and considered it in the interpretation, so that we do not over-interpret the results.

4. **The temperature field within the mountains is most likely not in steady state**: The authors have assumed that the temperature field throughout NW South America can be modelled as being in steady-state. The time taken for a system to approach thermal steady state is given by the thermal time-constant $\tau = l^2/\pi^2\kappa$ where $l$ is the thickness of the lithosphere and $\kappa$ is the thermal diffusivity of the lithosphere (~10–100 Ma for normal lithosphere). Given that mountain building will advect heat and move the lithosphere out of thermal steady-state [England and Thompson, 1984], and that mountain building in the NW Andes has taken place more recently than 10-100 Ma, then I would assume that the temperature field within the NW Andes cannot be modelled as being in thermal steady state. I agree that the steady-state assumption is more reasonable within the forelands.

The heat flow and downhole temperature data the author's use to validate the models come from the top 4 km of the crust, and are mostly located offshore and not within the continental lithosphere, therefore are not a good test of whether the model is representative of the temperature field at depth in Region 1, 2 or 3. To address this comment, the authors need to justify why they think the steady-state assumption is valid for the mid/lower crust.

**Response:**

Steady-state assumption:

Sensitivity studies (Meeßen, 2019) have shown that the transient effects causing the temperature changes in the upper 50 km of a subducting system are not far from equilibrium. Likewise, Rodriguez Piceda et al. (2022) have explored the effects of the non-steady state and came to the same conclusion.

Although it is true that mountain building processes imply that the system might not be fully in equilibrium, it is generally accepted that the first-order thermal field within the lithosphere is driven by conduction (e.g.: Liu et al., 2021). See response to comment Line 77 for more details.

We have improved the introduction with details about why we consider the steady-state assumption valid for our application in the study area.

Observational constraints:

We already acknowledged in the manuscript that there are limited heat flow and temperature measurements in the study area. Precisely for this reason, we made the effort to provide the best thermal model considering a fully 3D thermal approach, by integrating all the available data.

Indeed, the sensitivity test we performed with 25 different models (see Text S2 in the supplements) shows that the fit to the few temperature measurements available is already highly sensitive to the thermal parameters we explored. Figure S1 depicts the residual temperature for the different models we tested. The residuals of the first 19 models are particularly large, which implies that there is a limited range of the parameter values which can properly fit the observations.

In any case, even in an area with abundant high-quality temperature and heat flow observations (e.g. Southern California), those will be always restricted to the uppermost kilometers of the crust. See response to comment Section 4.1 for more details.

5.  **Lab and nature suggest olivine remains seismogenic up to ~600 degrees – not 600-1000 degrees:** There are two strong lines of evidence that suggest olivine-rich rocks can only remain able to nucleate earthquakes up to temperatures of ~600 degrees Celsius. The first comes from the depths of well-constrained earthquakes within the oceanic lithosphere, where the temperature structure is well known. Here the maximum centroid depths of large earthquakes can be seen to deepen with the age of the lithosphere and remain consistently shallower than the 600 degree isotherm [McKenzie et al., 2005; Craig et al., 2014]. Secondly, laboratory experiments show that olivine aggregates switch from deforming through stick-slip, to deforming through stable creep, at temperatures equivalent to ~600 degrees at the strain rates expected within the oceanic lithosphere [Boettcher et al., 2007]. The authors cite Scholz [2019] (a textbook) regarding the seismogenic range of temperatures for olivine being 600-1000 degrees, but do not describe the evidence for this. I would suggest that the author's cite the original papers that came to this conclusion, as I have not been able to check where their logic has come from.

**Response:**

The limiting temperature for seismogenesis in mantle forming materials has been a matter of debate. Some authors define a rather strict limit at 600°C (e.g.: McKenzie et al., 2005; Craig et al., 2014), but new evidence suggests that this limit might occur at higher temperatures. For example King and Marone (2012) explored the deformation of olivine fault gouges in the temperature range of 400–1000°C. Their results suggest

that the velocity weakening (negative a-b) occurs at temperatures between 600 and 1000°C, depending on the strain rates applied to the sample (Figure 1).

[Figure]

**Figure 1.** Stability parameter, a – b, for different velocity steps: green 1-10μm/s, orange 10-50μm/s and blue 50-1μm/s. (a) Rate-state model. (b) Two state variable approach. Figures taken from Figures 6 and 8 in King and Marone (2012).

Similarly, Ueda et al., (2020) studied the brittle-ductile transition in peridotites based on a fault system developed in the Balmuccia peridotite body. Their results indicate that the ductile-to-brittle transition in the peridotite occurs at ~720 °C. Lastly, Grose and Afonso (2013) studied the evolution of the oceanic lithospheric using more realistic thermal models than McKenzie et al. (2005), yielding a brittle-ductile transition closer to the 700-800°C isotherms depending on the mantle potential temperature (see Figure 10 in Grose and Afonso, 2013). This implies a higher temperature than a purely thermal, simplified approach, as their workflow considers not only hydrothermal circulation, but also P-T-dependent thermal properties.

We have integrated these new references to complement the framework of our analysis, both in the introduction and in the discussion of the results.

Regarding the citation of Scholz (2019), we modified the text adding the original references compiled by the textbook, and adding some additional works pointing to a higher temperature for seismogenesis of mantle forming materials, as described above.

The original references cited by Scholz (2019) are:

Blanpied, M. L., Lockner, D. A., and Byerlee, J. D. 1992. An earthquake mechanism based on rapid sealing of faults. *Nature* 358: 574–6.

Boettcher, M. S., Hirth, G., & Evans, B. 2007. Olivine friction at the base of oceanic seismogenic zones. *Journal of Geophysical Research: Solid Earth*, *112*(1), 1–13. https://doi.org/10.1029/2006JB004301

He, C. R., Wang, Z. L., and Yao, W. M. 2007. Frictional sliding of gabbro gouge under hydrotherynal conditions. *Tectonophysics* 445(3–4): 353–362, doi: 10.1016/j.tecto.2007.09.008

King, D. S. H., & Marone, C. 2012. Frictional properties of olivine at high temperature with applications to the strength and dynamics of the oceanic lithosphere. *Journal of Geophysical Research: Solid Earth*, *117*(12), 1–16. https://doi.org/10.1029/2012JB009511

Mitchell, E. K., Fialko, Y., and Brown, K. M. 2015. Frictional properties of gabbro at conditions corresponding to slow slip events in subduction zones. *Geochem. Geophys. Geosystems* 16 (11): 4006–4020, doi: 10.1002/2015gc006093.

**Line-by-Line Comments**

**Line 11:** "Crustal seismogenic thickness" is a misleading term in my view, because the seismogenic layer can include parts of the crust and upper mantle. I would suggest that just the depth extent of earthquakes in the lithosphere is the more logical description of the seismogenic thickness, and the depth most likely to correlate with a "stability transition" from seismogenic to aseismic deformation processes.

**Response:**

In the new manuscript, following the reviewer's comments, we have dismissed the D10 calculations, and so the seismogenic thickness (difference D90-D10) is no longer discussed.

Throughout the manuscript we now refer to the "crustal seismogenic depths" to refer to the depths up to which most crustal earthquakes occur. Since we are focused on the depth distribution of crustal seismicity, this implies disregarding sub-crustal seismicity in the statistical analysis (as done by others before, e.g.: Wu *et al*., 2017 in Taiwan, or Tsuda *et al*. 2019 in Japan).

**Line 18:** "Potential temperature" means specifically the temperature at which a rock would be if moved from a particular depth along an adiabat, which I think is different from what the authors mean here. I think they mean just the possible absolute temperature, and would recommend removing the word "potential".

**Response:** Thanks for pointing this out. We modified the text accordingly.

**Line 32:** Why is the crustal seismogenic thickness not just the layer where all earthquakes occur? I'm not sure why it would be defined as where 'most', but not all, earthquakes occur.

**Response:** Please see response to comment regarding Line 11.

**Line 39:** Is 90% of events based on a measure of statistical significance, or is it just defined as an arbitrary value?

**Response:** Please see response to comment #3.

**Line 46-48:** I would argue that it is generally accepted that earthquakes mostly nucleate within the crust at temperature <350 degrees Celsius, but can nucleate in lower crustal rocks at temperatures up to 600 degrees Celsius within certain areas where the lower crust might be dry [see Jackson et al., 2008; Craig et al., 2011; Sloan et al., 2011; Craig et al., 2012]. Examples of lower-crustal seismicity are now extensive, and there are plenty within South America [Assumpcao et al., 1992; Devlin et al., 2012; Wimpenny 2022].

**Response:** Please see response to comment #5.

**Line 73**: Typo "as the extent of the CST" not "extend".

**Response:** Fixed, thanks.

**Line 77:** It is not clear to me why flat slab subduction means that you can assume thermal steady state. If anything, I would assume that flat slab subduction would suggest that you need to account for the heat advection in the modelling of the temperature field, because the slab has not always been flat and therefore the temperature boundary condition on the base of the lithosphere has changed through time. Please could the authors elaborate on why a flat slab configuration allows them to make an assumption of thermal steady state?

**Response:**

The present-day flat slab geometry has been established since about 6 Ma, when the Nazca tear developed separating the north (flat) and south (step) segments (Wagner et al., 2017).  As a result, the volcanic activity has ceased in the continental crust of the overriding plate of the north segment, which spatially corresponds to our study area. This allows us to consider that the propagation of heat within the crust is mainly driven by conduction (e.g.: Liu et al., 2021).

Geodynamic models (e.g.: Schellart and Strak, 2021; Currie and Copeland, 2022) suggest that the subducting velocities are potentially reduced during the evolution from step to flab slab (see Fig 16 in Schellart and Strak, 2021 as an example -Figure 3 below). Considering that we are targeting the thermal field of the uppermost 75 km of the lithosphere, and that our applications are limited to crustal realms, we believe that a steady-state assumption can be applied in the study area to have a first-order estimate on the 3D feedback of the system heterogeneities and their imprint in the resulting thermal field. Moreover, given that the timeframe of the earthquakes we are studying is instantaneous in the geological timescales, our goal is to take a "snapshot"

of the present-day thermal configuration considering the 3D system heterogeneities at crustal scales.

[Figure]

**Figure 3.** Resulting velocity field during the evolution of slab flattening (Schellart and Strak, 2021).

We summarized these arguments in the introduction of the new version of the manuscript.

**Line 92:** What is meant by "large uncertainties" in the Moho? Could the authors give quantitative estimates of the typical Moho uncertainty?

**Response:**

The Moho uncertainty map is provided in the supplementary Figure S4, as computed by the authors of the GEMMA model (Reguzzoni & Sampietro, 2015) used in this research.

**Figure 1:** There are a number of references to different geographic places (e.g. Panama-Chaco Block) in the text that are not defined on Figure 1. Please could the authors add these place names to the figure to help the reader.

**Response:** Thank you. We have carefully checked this issue and improved the figures accordingly.

**Line 242-245:** The authors "disregard earthquakes below the Moho…", and use only events above the Moho to compute "… the upper and lower stability transitions". By ignoring earthquakes below the Moho, then the maximum depth of earthquakes the authors calculate is limited by the crustal thickness, and doesn't necessarily correspond to any "stability transition" or mechanical condition in the lithosphere. I would recommend changing the phrasing the authors are using to describe what they are actually measuring, which is more like the fraction of the crust that is seismogenic.

**Response:** Please see response to comment regarding Line 11.

**Line 263:** Why is removing events that are below the magnitude of completeness (Mc) relevant for this analysis, because the authors are not necessarily studying earthquake frequency or moment release through time? Small events below Mc could still be useful in defining the depth distribution of seismicity.

**Response:**

Small events below Mc are not useful for defining the depth distribution of seismicity, because their depths are biased. The deeper an earthquake is, the less likely it is to detect, that is, Mc increases with depth (e.g. Fig. 5 of Schorlemmer *et al.*, 2010). If we used an incomplete sample, considering earthquakes below Mc, small deep earthquakes would be preferentially missing and the overall apparent depth distribution would be biased towards shallower values. We have now mentioned these issues explicitly in section 3.2.1.

**Line 274-276:** The bootstrapping method of estimating the uncertainties in D10/D90 yield what I would consider unrealistically small values of <1 km. This is presumably because the 'formal' uncertainties in earthquake depths included in the ISC catalogue that are propagated through the bootstrapping are not realistic representations of the true depth errors. For example, comparing earthquake hypocentral depths in the ISC-EHB catalogue with more accurate centroid depths calculated from body-waveform modelling yields a mean difference on the order of ~5±10 km with errors of up to 50 km [Wimpenny and Watson, 2021]. Therefore, I expect a more realistic uncertainty is at least 5-10 km.

In addition, the inference that "low uncertainties [estimating from bootstrapping] indicate that using 20 earthquakes for each node is [sic] already reliable… " is not necessarily true in my view. If you used a sample size of 1 earthquake, then you would get low uncertainties, but the uncertainty estimate is not necessarily robust in that changing the sample used would give a different answer.

The way to check whether 20 events is a reasonable sample size for giving robust results would be to make sure that increasing the sample size does not alter the

conclusions (even if the spatial resolution has to decrease as a product of the greater sample size).

**Response:**

We have further extended the uncertainty analysis to better take these issues into account, and checked the robustness of the results. We had to focus on formal depth errors, because non-formal ones (such as systematic errors due to the velocity model used in the location), cannot be directly quantified unless the entire catalogue were relocated.

Improvement of the propagation of errors in the analysis

The former bootstrap method for estimating the D90 uncertainties considered only the effect of the limited sample of earthquakes used. It seems that the reviewer assumed that the hypocentral depth errors were also considered, but they were not.

As now described in Section 3.2.2, we have extended the method into a more computationally intensive one. It combines bootstrapping (to consider the effects of the finite earthquake sample, as before) and a Monte Carlo approach (randomizing each earthquake depth according to its reported uncertainty, in order to propagate this error into the final result).

Resulting uncertainties

The resulting uncertainties of D90 and D95 (Figure S9) are larger than before (typically between 1-2 km, instead of <1 km), but still rather small, on the same order (or even smaller) than the uncertainties of the Moho depth in the region (Figure S4).

This implies that the results on D90 are indeed robust. Using a larger sample size is not justified, and it would require smoothing out spatial variations (since earthquakes farther from each grid node would have to be considered in the corresponding sample). Using a too small sample size (<10 events) is not meaningful for calculating a 90% percentile. In general, using too small sample sizes would be penalized by bootstrap (yielding high uncertainties of D90), since it is sensitive to the robustness (or lack thereof) of the results due to the finite sample size.

Note that, since a sample of at least 20 earthquakes is used for calculating each statistic (D90 or D95), the resulting uncertainty tends to be smaller than the hypocentral depth error of a single earthquake. This is similar to the classical statistical observation that the mean of a sample of measurements tends to have an error smaller than that of each measurement alone.

**Line 292:** I recommend quoting the misfit as 5 degrees, not 4.99 degrees, as the latter makes it sound like the model and data have precision to two decimal places, which is unlikely.

**Response:** We agree, thanks. We have modified both the text and the figure accordingly.

**Section 4.1:** A more general comment is that the thermal model validation relies on comparing measurements of the temperature in the very shallowest part of the crust in areas that are mostly offshore and do not overlap with Regions 1, 2, or 3. Could the authors add some text explaining why they think that validating the model using measurements from outside of the region of interest means that it will be valid in the regions of interest, which may itself have a different geological history and therefore different temperature structure (e.g. the effects of mountain building on the geotherms).

**Response:**

See response to comment #4. The sensitivity test (Text S2 and Fig. S1) shows that, even if the temperature observations are not homogeneously distributed across the region, they are able to constrain the thermal model parameters (because choosing other values for the thermal parameters results in a worse fit).

Thinking about the study area as a 3D fully coupled system, we believe that being able to reproduce the observations is a good sign of the model's first-order robustness for the applications we currently consider in our manuscript. A main reason for our chain of arguments is that, we not only model temperatures, but consider the structural complexity and the heterogeneous distribution of thermal properties, together with the physics of conductive heat transport to predict the 3D thermal field. This field is, in turn, consistent with the few observed temperatures. The 3D heterogeneity in physical properties is constrained by other methods, such as seismic data or 3D gravity modelling, as certain seismic velocity- density pairs can be interpreted as lithologies, from where we can conclude on the lithology-dependent thermal properties. In fact, as expected, we do get different temperature structures all over the study area, as the model integrates the complexity of the region.

We should note that the procedure followed, of building a 3D model integrating all available information, is a major improvement with respect to earlier approaches. Moreover, if eventually more data of heat flow or thermal measurements become available, they could be integrated in future models following this philosophy.

**Line 317:** Why is the "regional seismogenic zone" defined as 20 km? This seems arbitrary to me, and different from the seismogenic thickness quoted elsewhere.

**Response:**

The histogram of the hypocentral depths of the selected earthquake catalogue (Figure 7 c) shows that the majority of crustal earthquakes in the study area nucleate at less than 20 km depth (the computed D90 of all crustal earthquakes is 20.5 km). Therefore, we selected the temperature difference between the surface and 20 km as a proxy for estimating the geothermal gradient in the seismogenic zone at regional scale.

**Line 326-329:** I would recommend re-wording this sentence, as on first reading I thought the authors were suggesting that areas with high geothermal gradients were areas with a strongly coupled crust and lithospheric mantle, though they actually mean it is the other way around. Maybe just split it up into a few shorter sentences for clarity.

**Response:** Thank you, we have now improved this paragraph according to the suggestion. It reads:

"Moreover, the geothermal gradients can also be used as an indirect indicator of crustal rheology. In Fig. 5a, it is possible to observe the correlation between the spatial distribution of seismicity and the geothermal gradients in this region. The crustal earthquakes occur at locations with a mean geothermal gradient of 19.4±1.23 °C/km-1, preferentially clustering in specific zones, e.g. in the North Andes block and the Panama microplate. Seismicity is almost absent in cold lithospheric areas such as the Guyana craton and the Caribbean Large Igneous Plateau. Such correlation indirectly suggests that in these places, the crust and lithospheric mantle may be strongly coupled, and therefore, the differential stress is not high enough to deform the crust in a brittle regime. This again, is an indication that a 1D geotherm approximation will not be robust enough to model the thermal configuration of the heterogeneous study area."

**Line 348-354:** This is an interesting test, and it is physically intuitive that areas of higher geothermal gradient will have more seismicity because the lithosphere is hotter, therefore weaker and deforms more rapidly in those areas. However, I would be interested to know whether the thermal modelling adds any more useful information on where earthquakes are expected to occur relative to direct surface observables such as topography or fault traces.

**Response:**

We did not explore the skillfulness of other geodynamic variables in our study area because testing this systematically would require a whole new paper (as done, e.g. by Becker et al., 2015). It would require dedicated modelling efforts, to check if, for example, strain rates and rates of topography change are skillful geodynamic variables at forecasting the spatial distribution of seismicity, as in the region analyzed by Becker

et al. (2015). It is important to note that it is not only the thermal gradient per se, but the contrast between domains of differing gradients that matters. We have now commented these issues in section 4.2.

**Line 407:** Please cite the original source for the constraint on the seismogenic temperatures for olivine gouge as 600-1000 degrees.

**Response:**

We have added all the original references regarding the cited seismogenic windows of different rocks and minerals.

**Line 415:** Why does shallow earthquake hypocentres mean faults are 'not well developed', and what is meant by 'not well developed'? Do you mean 'immature' in that they haven't accommodated a significant amount of relative motion? Is there any evidence from natural fault zones that the hypocentral distribution of earthquakes differs with the 'development' of the fault, or are these inferences based off of laboratory experiments?

**Response:**

As mentioned in the initial reply, considering the uncertainties discussed above for shallow hypocentres, we have removed this argument. This interpretation was explained by Scholz (2019, p. 149 and references therein), based on the seismogenic depths observed in different faults worldwide.

**Line 417-419:** This sentence captures much of my concern regarding the results of this study. The authors state: "However, despite relying only on the best located earthquakes (see Sect. 3.2.1), large errors in the hypocentral depths still remain (up to 30 km, see Fig. S5), and should be considered in the analysis of our results." I agree, but it's not clear to me how the authors have considered this in their own analysis of their results. In my view, the best way to mitigate this issue is to use more accurate methods for determining earthquake depths, which are more time consuming but will yield more robust results.

**Response:**

As noted above, a complete relocation of the catalogue is beyond reach, but, to deal with these issues, we have improved the earthquake catalogue, and performed a more exhaustive quantification of uncertainties of the D90 (and D95) percentiles.

We have also discussed in more detail (in sections 4.3 and 4.4) the limitations resulting from the uncertainties in focal depths or depth percentiles, being careful of not over-interpreting the results.

In any case, we would like to remark that the results agree well, e.g. with laboratory experiments of rock friction and with other recent publications, so we are not making "extraordinary" claims which would require additional robustness checks.

**Line 443:** What do the authors mean by "D90 depths of almost 35 km … are in agreement with the crustal-scale structure that these systems likely represent"? Faults can be 'crustal scale' but be aseismic still presumably (e.g. segments of the San Andreas Fault).

**Response:**

D90 at the fault location is almost 35 km, so its seismicity is distributed along the crust down to such depths, and the fault seems to be a crustal-scale feature. If D90 were, instead, e.g. 5 km, we could not argue that the fault is of crustal scale, as it might be shallow only.

We find this remark important because there are no seismic profiles or detailed tomographic models which could provide alternative evidence of the fault extent at depth.

**Line 445:** "The D90 values … are clearly bounded by major faults" – I find it hard to see these patterns reflected in Figure 8, mainly because there are a lot of faults that do not seem to bound changes in the colour patterns. Maybe highlight the specific faults thought to bound the changes in D90 in the Figure?

**Response:** We improved Figure 8, adding the general NW Andes terranes (blue polygons in Fig. 8b). We also include clearer statements on the faults that bound certain regions, and refer the readers to Figures 1 and 3, where the names of the main faults mentioned in the text are displayed.

**Line 560-565:** There are a few points where the authors mention the diverse geology of the northern Andes and how mafic/ultramafic terranes at depth might account for the deep seismicity. These arguments would benefit significantly from a simplified geological map of the study that highlights where these ultramafic/mafic terranes are based on surface outcrop and display these data relative to the seismicity and D90 maps.

**Response:**

Thank you. We have added a terrane map (Figure 1b) and referred to it throughout the text to support our arguments.
* * *
**Author's response to Reviewer #2**

We would like to thank reviewer #2 for his/her constructive comments and for appreciating our contribution.

We have tackled all the issues raised by its review, as described point by point below.

References not already cited in the manuscript are added at the end of this document.
* * *
Comment:

In this study the authors focus on the NW South America region to investigate the 3D distribution of the crustal seismogenic thickness as well as the temperature of occurrence of crustal earthquakes. They propose an integrated workflow that includes a lithological and a thermal model and incorporates observed seismicity. This approach is particularly suitable for this complex region, where different tectonic plates interact, there are several subduction zones, thick sedimentary basins and accreted terranes. An important contribution of this kind of study is that it improves the understanding of seismogenesis and has implication for future seismic hazard assessment. This is particularly relevant in the study region, which is seismically very active and has hosted devasting earthquakes.

The manuscript is well written and organized, but my main problem is that I found that many statements were poorly discussed or justified. Similarly, I often could not see in the quoted figures the statements made in the text. Some important conclusions were not sufficiently supported in the text. For example, the authors find that the hottest domains correspond to the deepest values of D90, without giving any physical explanation, and from this correlation they make the conclusion that 'The spatial variation in the geothermal gradient in the uppermost 20 km of the lithosphere has significant predictive power for forecasting the distribution of seismicity'. First, I find that this correlation between high geothermal gradient and deep crustal seismicity deserves further validation and discussion, since it is, at least, counterintuitive. A warmer crust should promote a ductile instead of a brittle (and then seismogenic) behaviour, for this reason the brittle-ductile transition should move to shallower depths and then the base of the seismogenic crustal layer should be shallower, what is the opposite to what obtained in the present study. I consider that this issue and other comments that I elaborate in the following notes can be addressed in a moderate revision.

**Response:**

There are two main concerns raised in this comment:

1. ..." the authors find that the hottest domains correspond to the deepest values of D90, without giving any physical explanation, and from this correlation they

make the conclusion that 'The spatial variation in the geothermal gradient in the uppermost 20 km of the lithosphere has significant predictive power for forecasting the distribution of seismicity'".

We believe this paragraph may reflect a misunderstanding of the methods and interpretations. The analysis of the **geothermal gradient** (sections 3.1.4 and 4.2) is completely different to that of the **D90** (sections 3.2.2 and 4.4).

In the previous version of the manuscript (section 4.4, lines 457-478), we discussed the different possibilities that such high D90 temperatures might imply. Nevertheless, we have significantly improved the discussion and interpretation in the new version of the document, including more clear statements about these deep seismic events, for example:

"Regarding the occurrence of crustal earthquakes at temperatures higher than the seismogenic windows expected for typical crustal rocks, it can be remarked that: 1) The earthquake dataset includes aftershocks (as otherwise the number of events for analysis would be further reduced), which may nucleate at depths larger than the base of the background seismogenic zone (Zielke et al., 2020). Thus, the calculated D90 values may be affected by transient deepening of the seismogenic crust during aftershock sequences. These deeper values would yield a larger temperature for the CSD than the long-term one. 2) The diverse allochthonous terranes accreted to NW South America, and the variety of autochthonous crustal blocks, include (ultra)mafic, olivine-rich rocks, which could host seismicity at larger temperatures. 3) The lower crust under part of the Andes may be mafic, able to host earthquakes at the relatively high modelled temperatures, which are due to a hot upper mantle together with a thick upper crust (which generates additional heat due to the decay of radioactive elements, (Vilà et al., 2010)."

On the other hand, the geothermal gradient calculation uses the modelled temperature at the surface and at 20 km below it (see section 3.1.4), but not the D90 temperatures. Therefore, we are not using the D90 to conclude that 'The spatial variation in the geothermal gradient in the uppermost 20 km of the lithosphere has significant predictive power for forecasting the distribution of seismicity'.

2. …" I find that this correlation between high geothermal gradient and deep crustal seismicity deserves further validation and discussion, since it is, at least, counterintuitive. A warmer crust should promote a ductile instead of a brittle (and then seismogenic) behaviour, for this reason the brittle-ductile transition should move to shallower depths and then the base of the seismogenic crustal layer should be shallower, what is the opposite to what obtained in the present study."

In our manuscript we are not claiming that there is a correlation between high geothermal gradient zones and seismicity. The previous version of the manuscript mentions (section 4.2):

"Moreover, the geothermal gradients can also be used as an indirect indicator of crustal rheology. In Fig. 5a, it is possible to observe the correlation between the spatial distribution of seismicity and the geothermal gradients in this region, as most of the seismicity preferentially clusters around zones with geothermal gradients > 19 °C/km-1: i.e.: the North Andes terranes and Panama microplate. Such correlation indirectly suggests that the later are places where the crust and lithospheric mantle are strongly coupled, and therefore, the differential stress is not high enough to deform the crust in a brittle regime."

Considering that the mean geothermal gradient for continental crust is about 25°C/km-1 (Criss, 2020; DiPietro, 2013), our computed values of maximum 23°C/km-1 should not be treated as high in a general sense (see Figure 5), but we rather referred to local maxima values.

In the new version of the manuscript we have improved section 4.2, to make a clearer interpretation between geothermal gradient and seismicity. In articular, we note that "Seismicity is almost absent in cold lithospheric areas such as the Guyana craton and the Caribbean Large Igneous Plateau. Such correlation indirectly suggests that in these places, the crust and lithospheric mantle may be strongly coupled, and therefore, the differential stress is not high enough to deform the crust in a brittle regime."

Main comments:

The last two paragraphs of the Introduction section have a level of detail about the adopted approach that the reader is not ready to follow at this point of reading (for example the validity of the steady-state assumption…) I suggest keeping this part more general and express the main purpose of this study. The style of the last statement ('the main advantage…') is more appropriate in this regard.

Lines 77-78. 'In order for this approach to be realistic, our thermal model considers only the uppermost 75 km of the lithosphere'. The authors should justify this choice and explain in what sense it is more realistic. The depth of 75 km is shallower than typical lithosphere-asthenosphere boundary (LAB) depths.

**Response:** Thank you. We have modified the introduction, trying to accommodate also the comments from Reviewer 1. We now provide, apart from details about the modelling approach, more background context to help the reader follow our assumptions.

There is not any discussion on the possible effects of the steady-state assumption. The authors argue (lines 75-77) that a steady-state approach is appropriate since they target crustal earthquakes. However, transient effects can also affect the shallower parts of the lithosphere. The authors also mention that the subducting segments of the Nazca and Caribbean slabs are flat in the study area. This is not a direct support for the steady-state assumption and in addition is contradictory with the steep Coiba slab plotted in Figure 10.

**Response:** We included now a short explanation about why we consider valid a steady-state assumption in this region. The new introduction mentions:

"A steady-state approach can be regarded as appropriate for this analysis since: 1) we preferentially target crustal earthquakes. 2) The subducting segments of the Nazca and Caribbean slabs in the study area are flat (Gómez-García et al., 2021; Kellogg et al., 2019); (Sun et al., 2022), implying that the subducting velocities might be lower than in steep slab segments (Currie and Copeland, 2022; Schellart and Strak, 2021). So, the transient effects of dynamic changes and mantle wedge cooling due to subduction occur in much longer temporal scales than those of the heat transport in the upper lithosphere and of the earthquake cycle. And 3) we are already considering the mantle imprint on the temperature field at 75 km depth as a lower boundary condition."

Regarding Figure 10 (now Figure 9), please be aware that it is not to scale, as already described both in the figure and the caption of the previous version of the manuscript. Line 535: … "Vertical scale exaggerated.". We have included the proper value of the vertical exaggeration in the figure (V.E. 8.5X).

Line 175: 'All lateral borders of the model are assumed to be closed' I don't understand this condition. Do the authors mean that the borders are insulating, with a zero horizontal heat flow?

**Response:** No. It means that we are not prescribing lateral boundary conditions. We assign thermal boundary conditions only at the top (Earth's surface) and bottom (75 km depth) of the model.

Concerning the structure, section 3.1.3 'Validation of the modelled temperatures' mentions the method used for this validation, but I do not see how this validation is actually performed. Related to this, I don't see the purpose of section 3.1.4 Geothermal gradient. Can the authors briefly advance the purpose of this calculation?

**Response:** We have renamed sect 3.1.3, now it is called "Data available for validating the thermal model". Moreover, we have updated the text giving more details about the model validation, for example:

"We validated the 3D thermal model (Sect. 4.1) by comparing available measurements of downhole temperatures (ANH, 2020) and surface heat flow (Lucazeau, 2019), not used as model inputs, against the corresponding modelled values. The locations of the control points are shown in Fig. 3. Our goal was to minimize the misfit between the observed and modelled values." In the supplementary material (Text S2 and Fig. S1) it is shown how the chosen model has the parameter values that minimize the misfit of downhole temperatures.

Regarding section 3.1.4 "Geothermal gradient", we are using the spatial variations of this quantity to support that a 3D modelling approach is necessary to describe reallistically the complex thermal behavior of the system. Additionally, as discussed in section 4.2 "Geothermal gradient: 3D variations and correlation with seismicity", we

also found correlations between the geothermal gradient and the spatial distribution of seismicity.

Lines 190-192: the statement:' the geothermal gradient for the crustal seismogenic zone was computed as the temperature difference between the surface and 20 km depth' seems to me just contradictory with what said just few lines before about the geothermal gradient being computed every 3 km. It seems therefore quite inaccurate just using the temperatures at the surface and 20 km depth.

**Response:** As mentioned in section 3.1.4, the geothermal gradient is not constant with depth. For this reason, we computed it following two different approaches:

1. For depths ranging from the surface (z=0) down to z=30 km, with incremental steps of 3 km. These results are presented in the supplementary Figure S6 and briefly discussed in section 4.2.
2. The geothermal gradient for the average crustal seismogenic zone was computed as the temperature difference between the surface and 20 km below it, the latter being approximately the average crustal seismogenic depth at regional scale (see section 4.3 and Figure 7).

Line 194. The statement 'A similar approach was followed by Gholamrezaie et al. (2018)' seems rather ambiguous? Do the authors refer to the approach to compute the thermal gradient of the 3D thermal modelling? Please clarify.

**Response:** We reworded the sentence as follows: "A similar approach for calculating the geothermal gradient based on 3D thermal models was followed by Gholamrezaie et al., (2018), also using a 3D modelling scheme in which the geological heterogeneities of the system were included.".

Line 285. The resulting D10 and D90 values, and their corresponding standard deviations are provided in the data repository (Gomez-Garcia et al., 2022). Why not shown here in a map?

**Response:** Following the reviewer #1 comments, we removed the analysis of the D10. And yes, we now include the uncertainty estimation for the D90 map (Figure 8) in the new version. Please notice that we are also computing D95, and discuss these results as supplementary information. The new text reads: "The resulting D90 (and D95) values and their corresponding standard deviations and resolution radii are provided in the data repository (Gomez-García et al., 2023) and discussed further in Sect. 4.4."

Line 299. The authors mention that 'it is possible to conclude that the model (heat flow) fits the regional trend', but honestly, I don't see this satisfactory fitting in figure 4b. Similarly, modelled temperatures shown in figure 4a seem to systematically underestimate observations at depths < 4 km. Can the authors hypothesize about this (small but systematic) misfit?

**Response:** Surface heat flow estimates are usually not only affected by advective processes (as already mentioned in the paper), but also have high uncertainties. On

the other hand, the typical error of borehole temperatures is around 5 degrees C, which is the average misfit we get for our modelled values (see Figure 4a). In Figure 4a, it is possible to appreciate that the largest misfits occur at depths shallower than 1 km. In general, the uppermost crustal region is prone to advective processes such as fluid circulation, that is out of the scope of this research, especially given the rather small spatial scales at which these processes occur, compared to our regional-scale model. We have modified the text in section 4.1 to clarify this point.

Despite we are using the best-quality heat flow measurements available they still have errors between 10% and 20%. This is the main reason why, as described in the paper, we are only looking at the regional trend of the regions outside known actively advective systems (marked by ovals in the figure below).

[Figure]

Lines 320-321. I don't follow the logic of the argument 'This again is an indication that a 1D approximation is not robust enough to model the thermal configuration of the study area' as the occurrence of earthquakes in regions with a diverse range of geothermal gradients would be also reproduced in 1D or 2D models, although these 1D or 2D approaches become less accurate in areas with large lateral thermal variations.

**Response:** Thank you. We meant a 1D geotherm approximation. We fixed it in the new version.

Lines 32-327. I don't see that in what sense the correlation between seismicity clusters and high geothermal gradient suggests that this geothermal gradient occurs in 'places where the crust and lithospheric mantle are strongly coupled'. Theoretically, a high temperature should lead to the shallowing of the brittle-ductile transition and therefore it should promote decoupling between the upper crust and the lithospheric mantle, with a ductile-aseismic lower crust in between. In summary, keeping the rheological stratification of the continental crust in mind, a strong coupling between the crust and the lithospheric mantle should correspond to a colder crust, and therefore a predominant brittle behaviour and absence of a ductile layer in the lower crust weak enough to mechanically decouple crust and lithospheric mantle.

**Response:** Thank you. We noticed this paragraph was confusing. We updated it as follows:

"Moreover, the geothermal gradients can also be used as an indirect indicator of crustal rheology. In Fig. 5a, it is possible to observe the correlation between the spatial distribution of seismicity and the geothermal gradients in this region. The crustal earthquakes occur at locations with a mean geothermal gradient of 19.4±1.23 °C/km-1, preferentially clustering in specific zones, e.g. in the North Andes block and the Panama microplate. Seismicity is almost absent in cold lithospheric areas such as the Guyana craton and the Caribbean Large Igneous Plateau. Such correlation indirectly suggests that in these places, the crust and lithospheric mantle may be strongly coupled, and therefore, the differential stress is not high enough to deform the crust in a brittle regime. This again, is an indication that a 1D geotherm approximation will not be robust enough to model the thermal configuration of the heterogeneous study area."

Lines 458-459 'Instead, the hottest domains are associated to sedimentary basins (Fig. S3) and correspond to the deepest values of D90'. I think that a simple interpretation for this is that temperature increases with depth and therefore deepest values of D90 occur at higher temperatures. This seems in contradiction with what said in lines 510-513 'However, we observe a general trend between the lithospheric configuration and the seismicity distribution, that is the colder and therefore stronger the lithosphere, the deeper and higher in magnitudes the earthquakes (e.g.: Chen et al.,2013).' Overall I do not understand why a cold lithosphere correlates with deep seismicity, while a high geothermal gradient correlates with deepest D90. I do not see a plausible explanation for this opposite effect of the thermal state in the crust and in the lithospheric mantle on seismicity distribution.

**Response:** Thank you. We clarified this text to avoid confusions. We also removed the phrase in lines 510-513. Please note that the section 4.5 (of the original manuscript) has now been partially merged with section 4.4 "Depths and temperatures at the base of the seismogenic crust (D90)", as we are not discussing the crustal seismogenic thickness anymore. This is because D10 is no longer considered in our paper, as suggested by Reviewer 1.

Lines 475-476 'a thick lower crust together with a relatively hot upper mantle could contribute to large hypocentral temperatures (Sect. 4.5)' I do not understand the relation between the lower crust thickness and the hypocentral temperature. As in my previous comments I suggest the authors to give an explanation for this kind of poorly explained inferences.

**Response:** We modified the text in section 4.4 and added this explanation. For example:

"Variations of the base of the seismogenic crust (magenta dotted line in Fig. 9) are not necessarily correlated with variations in Moho depths. Between ~74°W and ~76°W (approximately corresponding to region 2, Fig. 1a), there is an abrupt deepening of D90, which correlates with a thick lower crust and with the shallowing of the 600°C

isotherm due to the thermal imprint of a hot upper mantle. This deepening of D90 causes its correspondingly high temperatures in region 2, as already discussed. Again, **to explain that crustal earthquakes occur down to deeper locations despite of the hot lower crust, it is necessary to hypothesize that the latter has a mafic composition, with a deeper brittle-ductile transition.**"

In the case of the sentence mentioned in your comment, we modified it as follows:

"The lower crust under part of the Andes may be mafic, able to host earthquakes at the relatively high modelled temperatures, which are due to a hot upper mantle together with a thick upper crust (which generates additional heat due to the decay of radioactive elements (Vilà et al., 2010)."

lines 510-513 'However, we observe a general trend between the lithospheric configuration and the seismicity distribution, that is the colder and therefore stronger the lithosphere, the deeper and higher in magnitudes the earthquakes (e.g.: Chen et al.,2013).' I don't see this in Figure 10. Please, note that deep earthquakes beneath the Coiba slab correlate with a shallowing of the 600 ºC isotherm.

**Response:** Thank you. We removed this sentence, as it was confusing.

Line 579 'the seismogenic crust is thicker and hotter below the thick Middle Magdalena basin'. I see in figure 10 that the seismogenic crust is thicker, but not hotter than beneath the surrounding Murindó and Guaicaramo/Yopal faults.

**Response:** The seismogenic crust is hotter underneath the Middle Magdalena basin as D90 reaches deeper depths compared to Murindó and Guaicaramo/Yopal faults. However, please notice that we removed the discussion of D10 following reviewer #1 suggestions, and also removed this sentence from the conclusions as it was not a strong argument to be included there.

Minor comments:

Line94: remove the typo Reguzzoni & Sampietro,ta 2015

In color code for figure 2a, better say 'lithospheric mantle' instead of 'mantle'. Similarly, in the second line of the caption of Fig. 2, better say 'lithospheric mantle' instead of 'upper mantle'

Line 392. The reference to Fig 2 is wrong.

Line 551 'for calculating of the thermal field' remove 'of'

**Response:** We have corrected them all, thank you.

**References cited (if not already included in the manuscript)**

Alvarado, P., Beck, S., Zandt, G., Araujo, M. & Triep, E. (2005): Crustal deformation in the south-central Andes backarc terranes as viewed from regional broad-band seismic waveform modelling. *Geophysical Journal International*, 163: 580-598. doi:10.1111/j.1365-246X.2005.02759.x

Boettcher, M. S., Hirth, G., & Evans, B. (2007). Olivine friction at the base of oceanic seismogenic zones. *Journal of Geophysical Research: Solid Earth*, *112*(1), 1–13. doi:10.1029/2006JB004301

Criss, R. (2020). Chapter 6 - Thermal models of the continental lithosphere. *Heat Transport and Energetics of the Earth and Rocky Planets*. Elsevier, 151-174. ISBN 9780128184301. https://doi.org/10.1016/B978-0-12-818430-1.00006-9

DiPietro, J. (2013). Chapter 20 - Keys to the Interpretation of Geological History. *Landscape Evolution in the United States*. Elsevier, 327-344. ISBN 9780123977991. https://doi.org/10.1016/B978-0-12-397799-1.00020-8.

Gómez-García, Ángela María; González, Álvaro; Cacace, Mauro; Scheck-Wenderoth, Magdalena; Monsalve, Gaspar. (2023). Hypocentral temperatures, geothermal gradients, crustal seismogenic depths and 3D thermal model of the Southern Caribbean and NW South America. GFZ Data Services. (temporary link: https://dataservices.gfz-potsdam.de/panmetaworks/review/92360098bcf5658b89ffed28d0fbcbf1ebd0f9204f07f5b007ee6961268dd679/)

Liu, X., Currie, C. A., & Wagner, L. S. (2021). Cooling of the continental plate during flat-slab subduction. *Geosphere*, 18(1), 49–68. doi.org/10.1130/GES02402.1

Meeßen, C. (2019). The thermal and rheological state of the Northern Argentinian foreland basins, PhD Thesis, Potsdam: Universität Potsdam, 151 p. https://doi.org/10.25932/publishup-43994

Schellart, W. P., & Strak, V. (2021). Geodynamic models of short-lived, long-lived and periodic flat slab subduction. *Geophysical Journal International*, 226(3), 1517–1541. doi.org/10.1093/gji/ggab126

Schorlemmer, D., Wiemer, S. & Wyss, M. (2004), Earthquake statistics at Parkfield: 1. Stationarity of *b* values, *J. Geophys. Res.*, 109, B12307, doi:10.1029/2004JB003234.

U.S. Nuclear Regulatory Commission (2012): *Central and Eastern United States Seismic Source Characterization for Nuclear Facilities.* Technical Report. EPRI, Palo Alto, CA, U.S. DOE, and U.S. NRC. http://www.ceus-ssc.com

Wagner, L. S., Jaramillo, J. S., Ramírez-Hoyos, L. F., Monsalve, G., Cardona, A., & Becker, T. W. (2017). Transient slab flattening beneath Colombia. *Geophysical Research Letters*, 44(13), 6616–6623. doi.org/10.1002/2017GL073981

---

## Referee Report (RR1)

**Re-review of the paper: "*Thermal structure of the southern Caribbean and NW South America: implications for seismogenesis*" by Gómez-García et al.,**

**Reviewer:** Dr Sam Wimpenny (University of Bristol, UK)

Thank you to the authors for engaging constructively with the first round of reviews and for their responses.

The edits the authors have made have clarified how the uncertainties associated with the earthquake depth estimates might translate into uncertainties in the estimated temperatures at the earthquake hypocentral locations. These new additions do not significantly change the relationships between earthquake depths and temperatures originally presented.

There are several line-by-line comments that need to be addressed. The text can be quite difficult to follow sometimes, with incomplete reasoning (as pointed out by Reviewer 2 in the original reviews), meaning I had to re-read sections multiple times to understand the arguments being presented. I have tried to highlight these in the review.

The only technical comment I have is to include some logic for what the authors think the possible range is for the temperature estimates at each earthquake hypocentral depth.

I recommend the manuscript could be accepted following these minor revisions.

**General Comments:**

1. **Thermal modelling uncertainties:** There is no mention of the uncertainties associated with the temperatures derived from the thermal modelling. Uncertainties will derive from the material parameters (radiogenic heat production, thermal conductivity), as well as the basal boundary condition at 75 km based on converting S-wave velocities to temperatures. The approach the authors take is to select one model that "best-fits" the surface temperature observations from a subset of 25 models in which they have varied the material parameters. However, given that there are vast numbers of variables in these 3-dimensional models, then 25 models as a sensitivity test is unlikely to capture the full range of possible temperature distributions that could match the data. It is also unclear which of the models they discard fit the data slightly worse than the best-fit model, but still fit the data to within its uncertainties, in which case the data cannot be used to infer which model is most accurate.

   The arguments the authors present would be significantly improved by including some discussion of the estimated uncertainties in temperatures from their thermal models, and how these uncertainties translate into the uncertainties in the earthquake hypocentral temperature estimates.

2. **Robustness versus physical meaning of D90:** In the original review I suggested that D90 might not be the relevant metric for mapping the controls on the depth of earthquake generation, because it inherently ignores the deepest events and therefore consistently under-estimates the depth to the base of the seismogenic layer. I agree with the authors that estimating D100 is not robust, as one new earthquake can change the D100 estimate. However, there is an important difference between whether a metric is robust, versus if one is physically relevant. The D90 should track changes in the depth to which the *majority* of the seismicity takes place. It does not

necessarily track the brittle-ductile transition. I would recommend that the authors go through the manuscript and make sure this distinction is made clear.

3. **Use of colons throughout the text**: There are a number of places where colons are used after abbreviations (like e.g.:). I'm not sure the colons are necessary, but this can be confirmed by typesetting of the article.

4. **The role of hydration state:** The majority of the manuscript focuses on how lithology and temperature might be the main control on why some parts of the deep crust are seismogenic but others are not. A number of studies have recognised that the presence of water within minerals and interstitial water is also likely to be important in controlling whether a given material will be seismogenic at a given temperature [e.g. Mackwell et al., 2004; Jackson et al, 2008]. Can the authors explain why they think hydration state of the crust is not important, or why they have not mentioned hydration, in their discussion for the controls on the depth of earthquakes?

**Line by Line Comments:**

**Line 14**: If the authors want to use crustal seismogenic depth (CSD) then, in my opinion, they need to be explicit that it is similar to the brittle-ductile transition, but if seismicity extends through the crust and into the upper mantle, then the CSD does not correspond to the brittle-ductile transition. Statements like "… the CSD is a proxy for the brittle-ductile transition…" are potentially true, but not always.

**Line 15:** "The CSD largely limits the depth down to which crustal earthquakes may rupture …" – the phrasing of this makes it sound like the CSD controls the rupture depths. I'd suggest re-phrasing to: "The CSD represents the depth to which crustal earthquakes occur, and therefore is an important constraint on the seismic hazard in a region because it will be related to the maximum depth of earthquake ruptures".

**Line 32-33:** "The coherence of the hypocentral temperatures with those expected from laboratory measurements provides additional support to the model." – which model? The thermal model? The model in which lab experiments are extrapolated to lithospheric scales?

**Line 40-41:** I would suggest removing the text after "… i.e., temperature at which …" and then merging the second paragraph with the first.

**Line 41**: Change "assemblies" to "assemblages"

**Line 50-53:** It would be worth explaining exactly how Ueda et al., (2020) inferred that earthquakes occur in mafic rocks at temperatures of ~720 degrees C [i.e. they used thermobarometry of mineral assemblages in rocks containing psuedotachylytes to infer the temperatures at which the psuedotachylytes formed]. This is relevant because the thermobarometry results have associated uncertainties of ±50 degrees typically, so the range of temperatures might be more like 670-770 degrees C.

**Line 53:** "Afonso and Grose (2013) … used a more realistic thermal model than … McKenzie et al., (2005)" – more realistic in what way? Answering this question is important for the reader to be convinced that the models were in fact an improvement, and therefore that the existing bounds on the temperature of seismogesis in the mantle may be incorrect. I think Afonso and Grose included the temperature dependence of density, specific heat and conductivity derived from laboratory experiments (though worth double-checking this)?

**Line 64-67:** "intracontinental faults, the brittle to ductile transition seems to be … limited by the 300-350 degree C isotherm" – this statement is likely not true. There is plenty of evidence for earthquakes occurring on intracontinental fault zones at depths where the estimated temperatures far exceed 300-350 degrees C [see Jackson et al., 2008; Sloan et al., 2011; Craig et al., 2012; Emmerson et al., 2006].

**Line 63:** Consider changing "up-scale" to "scale up" and "target" to "determine".

**Line 67:** Can you cite some examples of where the high geothermal gradient correlates with shallowing seismicity to support your argument here?

**Line 71-72:** I agree that the 1-dimensional geotherms are a simplification, but these simplifications are made because it is believed that horizontal diffusion or advection of heat plays a minor role in controlling the temperature field in relatively stable tectonic settings. Similarly, a simple layered geometry is often assumed, because the exact nature of the subsurface lithologies, and their material properties (e.g. radiogenic heat production, thermal properties) are not known precisely. The key point is that the unknowns contribute greater uncertainty to the temperature predictions than does ignoring horizontal diffusion of heat, and therefore the 1-D simplification is justified. Equally, a full parameter sweep can be performed with 1-D models, meaning you can quantify uncertainty more easily.

**Line 79-80:** "Upscaling the seismogenesis from laboratory experiments…" consider changing wording to "… scaling up the predicted conditions of seismogenesis from laboratory experiments to the lithosphere"

**Line 84:** Is the CSD really "influenced… by the local geothermal gradient"? The local geothermal gradient is just a proxy for the absolute temperature at depth, which is most likely the parameter that controls whether faults break in earthquakes or creep aseismically and therefore the CSD.

**Line 88:** "The subducting segments of the … slab … in the study area are flat … implying that the subucting [sic] velocities might be lower than in the steep segments" – is this true? I would assume that, if the slabs are plate-like and do not deform extensively internally, then the subduction velocity and the advection of heat beneath the overriding plate should only vary along the length of the subduction zone due to the variations in relative plate motions about the plate's rotation pole. What evidence is there that the subduction velocity is slower in the flat slab segments compared to in the steep slab segments in the Andes?

**Line 91:** "on much longer timescales" not "in much longer timescales".

**Line 93-95:** I am really struggling to understand what this paragraph means. Maybe consider re-phrasing to: "The novelty of our study is to consider how spatial heterogeneity in the lithology of the lithosphere and mantle temperature influences the temperature distribution and seismicity within the crust"?

**Line 96-102:** This statement about seismic hazard is important, but the authors need to be more explicit of exactly how their work can update our understanding of seismic hazard in the region. Specifically, it will provide an estimate of the spatial variation in the thickness of the seismogenic crust and its links with surface observables. The thickness of the seismogenic crust sets the seismogenic area of faults, and therefore the possible maximum magnitude of earthquakes these faults can host.

**Line 105:** "results" not "resulted".

**Line 108:** Spelling error – "steep" not "step".

**Line 110:** Are there any studies to cite that have looked at the timing of volcanism across the region from absolute dating, and which have argued that the presence of a flat slab led to the termination of volcanism, to support this argument?

**Line 113:** "remainder" not "remaining".

**Line 130:** "dominated by plateau and magmatic arc terranes" – as in oceanic plateau rocks? Please clarify what is meant by plateau rocks.

**Line 140-144:** A general question about this section that it might be worth trying to address in the text: why is it relevant what geologically-inferred sutures and fault systems run through the study area? I can see why the geological terranes are important, but less so the specific faults.

**Line 184-186:** "The best model was selected as the one that independently best reproduced the temperatures measured in the boreholes". It seems important to consider all the models that match the borehole temperature data to within the data's uncertainties (±10 degrees?), as opposed to just the one model that best-fits the data, especially if the differences in data fits are small. The reason I say this is because models with the same near-surface temperatures and mantle temperatures could have very different temperatures in the mid- and lower-crust, so just considering a single model might not be reflective of the range of possible temperatures at the depths of earthquakes.

**Line 238:** I still think that "Depth of Crustal Earthquakes" is clearer than "Crustal Seismogenic Depth", but I'll leave it up to the authors to decide what they want to use.

**Line 246-247:** You can remove the citations for Wimpenny et al., 2018 and Wimpenny 2022, as their results are included in the gWFM catalogue that you have already cited.

**Line 264:** A better citation than Wimpenny (2022) here would be something like McCaffrey and Abers (1988) or Nabalek (1984), as these were really the papers that demonstrated how waveform-modelling methods provided more accurate estimates of earthquake centroid depths.

**Line 312:** Change to "… first converted to Mw using the relations…"

**Line 313:** I would recommend just leaving this sentence as "using the relations detailed in Text S6." – all the citations are difficult to follow and can be found in the Supplement.

**Line 409-410:** I would recommend removing the point about whether the crust and lithospheric mantle are coupled or not – I don't see how the lack of seismicity can tell us that.

**Line 469:** Should be "… towards lower hypocentral temperatures in the Falcon Basin"?

**Line 493:** Should this say "regional average seismogenic depth" as opposed to just "regional seismogenic depth"?

**Line 495-500:** The authors argue that the Murindo earthquake had a hypocentre near the base of the seismogenic layer and that the earthquake ruptured to the surface. Can the authors cite the evidence that this earthquake did in fact rupture to the surface? From what I can tell from the literature, there were earthquake environmental effects, but no recorded primary surface ruptures.

**Line 510:** Maybe change the subtitle to: "Temperature at the base of the seismogenic crust"?

**Line 515:** The blue polygon in Figure 8b is very difficult to see, at least on my screen. Maybe make it clearer by making the line thicker, or putting a white background behind it?

**Line 515:** It is not entirely clear what "sheared continental affinity" actually means. Why is the inference that it has been sheared relevant? Is it not just that the terrane is formed from continentally-derived rocks that's relevant?

**Figure 8:** There are a lot of references in the text to fault zones that are shown on Figure 3, but not on Figure 8. Because the D90 information is on Figure 8, then it is difficult to follow exactly where the authors are referring to in the text without having both Figure 3 and Figure 8 next to one another. I would recommend either adding the relevant fault zone names, or adding some annotations. They could also remove the fault zones that are not relevant from the line map to focus the readers eye on the relevant information. These are just some suggestions.

**Line 512-515:** Here the authors say that there is "… a transition from shallow D90 depths and cold temperature associated to [sic] oceanic terranes and island arc affinity in Western South America, towards deeper and hotter values in terranes with a more sheared continental affinity in the east." – The authors need to clarify exactly which part of their study region "west" and "east" are referring to here. The trends they point out are subtle and only hold in certain parts of the map area. For example, if you look at a west-east traverse for two profiles extracted from Figure 8a (see Figure R1 below), for the northern most profile the D90 gets deeper beneath the Middle Magdalena Basin compared to the Western Cordillera. However, for the profile further south the D90 is deeper beneath the Western Cordillera than it is beneath the Middle Magdalena Basin (MMB).

[Figure]

**Figure R1:** Figure 8a from the manuscript with two E-W profiles shown as white dashed lines.

**Line 520-522:** Can I suggest re-phrasing these sentences to: "We interpret the observed variability in D90 between the Central and Eastern Cordilleras, and the Middle Magdalena Basin, to suggest there is significant rheological contrasts between these areas. These major terranes are likely separated by crustal-scale faults." At the moment, I find it hard to understand what these sentences are trying to say.

**Line 547:** Spelling error? Should say "CSD" not "SCD".

**Line 628:** Why can a hot upper mantle explain why there are earthquakes at high temperatures? The hot upper mantle just explains why the temperature is high, not necessarily why there is seismicity in the rocks at these high temperatures. For example, Iceland has a hot upper mantle, has a predominantly mafic crust because it has formed from MOR volcanism, but there is very little seismicity deeper than 10-15 km (if any).

---

## Author Response (AR2)

Barcelona, Spain
November 29th, 2023

Dear Editor,

We would be very grateful if you could consider our manuscript "Thermal structure of the southern Caribbean and NW South America: implications for seismogenesis" for publication in Solid Earth, revised to tackle all the minor issues raised by the reviewers.

We are again very thankful to Dr. Wimpenny and the anonymous reviewer #2 for their useful feedback, which has contributed to significantly improving the manuscript.

The latter, as before, is accompanied by supplementary electronic materials, and all original data and results have been published in a separate public data repository that will be available online if the paper is accepted for publication.

A point-by-point reply to the reviewers follows.

We hope that you will find the submission in good order. Thank you very much for your consideration and best regards,

Ángela M. Gómez, also on behalf of the co-authors.

**Reply to comments by Dr. Wimpenny**

**General Comments:**

**1. Thermal modelling uncertainties:** There is no mention of the uncertainties associated with the temperatures derived from the thermal modelling. Uncertainties will derive from the material parameters (radiogenic heat production, thermal conductivity), as well as the basal boundary condition at 75 km based on converting S-wave velocities to temperatures. The approach the authors take is to select one model that "best-fits" the surface temperature observations from a subset of 25 models in which they have varied the material parameters. However, given that there are vast numbers of variables in these 3-dimensional models, then 25 models as a sensitivity test is unlikely to capture the full range of possible temperature distributions that could match the data. It is also unclear which of the models they discard fit the data slightly worse than the best-fit model, but still fit the data to within its uncertainties, in which case the data cannot be used to infer which model is most accurate.

The arguments the authors present would be significantly improved by including some discussion of the estimated uncertainties in temperatures from their thermal models, and how these uncertainties translate into the uncertainties in the earthquake hypocentral temperature estimates.

**Response:**

We would like to thank the reviewer for this comment. We agree that 25 models do not capture the full range of sensitivity due to different thermal parameters. However, this is out of the scope of our manuscript, as dedicated efforts are required to properly achieve such uncertainty quantification. This is further explained in the response to the comment "Line 71-72".

Regarding the uncertainties in the hypocentral depth temperatures, we have added the following text in Sec. 4.3:

Given a thermal model, errors in focal depths propagate into uncertainties in the hypocentral temperatures. The values represented in Figs. 6 and 7 are the most likely ones, corresponding to the best estimates of hypocentral locations; uncertainties have been omitted for clarity. For each earthquake, the possible temperature range can be measured directly from the 3D thermal model (Gómez-García et al., 2023), considering the depth range resulting from the best depth estimate plus/minus the formal 90% depth error. Also, an approximate estimate of its temperature uncertainty can be obtained by multiplying the depth error times the local geothermal gradient at the hypocentral location (e.g. Figs. 5 and S6). For deeper crustal earthquakes, both the formal depth errors (Fig. S5) and the local geothermal gradients (Fig. S6) are typically smaller than those for shallower events, implying typically smaller temperature uncertainties too. Note that real hypocentral depth errors may be larger than the formal ones reported in the catalogues (e.g. Wimpenny and Watson, 2020), due to systematic errors earthquake location procedures, such as in the assumed seismic

velocity model (e.g. Husen and Hardebeck, 2010). Consequently, eventual improvements in velocity models and earthquake location accuracy will directly reduce the uncertainties in hypocentral temperature estimates.

**2. Robustness versus physical meaning of D90:** In the original review I suggested that D90 might not be the relevant metric for mapping the controls on the depth of earthquake generation, because it inherently ignores the deepest events and therefore consistently under-estimates the depth to the base of the seismogenic layer. I agree with the authors that estimating D100 is not robust, as one new earthquake can change the D100 estimate. However, there is an important difference between whether a metric is robust, versus if one is physically relevant. The D90 should track changes in the depth to which the *majority* of the seismicity takes place. It does not necessarily track the brittle-ductile transition. I would recommend that the authors go through the manuscript and make sure this distinction is made clear.

**Response:**
Thank you. We have removed from the text the relation with the brittle-ductile transition to avoid potential misunderstandings.

The robustness of D90 has also been recently highlighted by Ellis et al. (2023) when estimating the maximum depth of seismic rupture on New Zealand's active faults: "We have used D90 rather than D95 as a more robust estimate of H, because D95 (the 95% seismicity cutoff depth) will be more sensitive to location and depth errors for regions with sparse seismicity in which depth uncertainties are about 5–10% of total depth." We now cite this reference in the text.

**3. Use of colons throughout the text**: There are a number of places where colons are used after abbreviations (like e.g.:). I'm not sure the colons are necessary, but this can be confirmed by typesetting of the article.

**Response:**
Thank you. We removed the colons as it seems to be the editorial style.

**4. The role of hydration state:** The majority of the manuscript focuses on how lithology and temperature might be the main control on why some parts of the deep crust are seismogenic but others are not. A number of studies have recognised that the presence of water within minerals and interstitial water is also likely to be important in controlling whether a given material will be seismogenic at a given temperature [e.g. Mackwell et al., 2004; Jackson et al, 2008]. Can the authors explain why they think hydration state of the crust is not important, or why they have not mentioned hydration, in their discussion for the controls on the depth of earthquakes?

**Response:**
Thank you for your comment. We briefly mentioned the role of dehydration reactions in section 4.4:

"Alternatively, the occurrence of upper mantle earthquakes is nowadays broadly recognized (e.g. Chen et al., 2013) as also dehydration reactions can trigger seismicity at temperatures above the normal brittle-ductile transition (e.g. Bishop et al., 2023; Rodriguez Piceda et al., 2022)."

However, as our modelling scheme does not take into account water content, we decided to keep this discussion out of the paper's scope.

We have added the new references suggested by the reviewer to the paragraph above.

**Line by Line Comments:**

**Line 14**: If the authors want to use crustal seismogenic depth (CSD) then, in my opinion, they need to be explicit that it is similar to the brittle-ductile transition, but if seismicity extends through the crust and into the upper mantle, then the CSD does not correspond to the brittle-ductile transition. Statements like "… the CSD is a proxy for the brittle-ductile transition…" are potentially true, but not always.

**Response:**
We removed from the text the relation with the brittle-ductile transition to avoid potential misunderstandings.

**Line 15:** "The CSD largely limits the depth down to which crustal earthquakes may rupture …" – the phrasing of this makes it sound like the CSD controls the rupture depths. I'd suggest re-phrasing to: "The CSD represents the depth to which crustal earthquakes occur, and therefore is an important constraint on the seismic hazard in a region because it will be related to the maximum depth of earthquake ruptures".

**Response:**
Considering the use given to CSD in the seismic hazard literature, we have rewritten the sentence as "In particular, most earthquakes in the crust nucleate down to the crustal seismogenic depth (CSD), which is a proxy to the maximum depth of crustal earthquake ruptures in seismic hazard assessments."

**Line 32-33:** "The coherence of the hypocentral temperatures with those expected from laboratory measurements provides additional support to the model." – which model? The thermal model? The model in which lab experiments are extrapolated to lithospheric scales?

**Response:**
We modified the text as follows:
"The coherence of the calculated hypocentral temperatures with those expected from laboratory measurements provides additional support **to our modelling workflow**."

**Line 40-41:** I would suggest removing the text after "… i.e., temperature at which …" and then merging the second paragraph with the first.

**Response:**
Thank you. We changed the text accordingly.

**Line 41**: Change "assemblies" to "assemblages"

**Response:**
Thanks for pointing out this typo. We fixed it through the text.

**Line 50-53:** It would be worth explaining exactly how Ueda et al., (2020) inferred that earthquakes occur in mafic rocks at temperatures of ~720 degrees C [i.e. they used thermobarometry of mineral assemblages in rocks containing psuedotachylytes to infer the temperatures at which the psuedotachylytes formed]. This is relevant because the thermobarometry results have associated uncertainties of ±50 degrees typically, so the range of temperatures might be more like 670-770 degrees C.

**Response:**
We modified the text as follows:
"For example, Ueda et al. (2020) found that the brittle-to-ductile transition in peridotite occurs at ~720 °C, **based on thermobarometry of equilibrium mineral assemblages in fault-related deformed rocks (pseudotachylytes, cataclasites, and mylonites)**."

**Line 53:** "Afonso and Grose (2013) … used a more realistic thermal model than … McKenzie et al., (2005)" – more realistic in what way? Answering this question is important for the reader to be convinced that the models were in fact an improvement, and therefore that the existing bounds on the temperature of seismogesis in the mantle may be incorrect. I think Afonso and Grose included the temperature dependence of density, specific heat and conductivity derived from laboratory experiments (though worth double-checking this)?

**Response:**
Thank you. We clarified this sentence and updated it as follows:
"Similarly, Grose and Afonso (2013) studied the evolution of the oceanic lithosphere using more realistic thermal models than those assumed by McKenzie et al. (2005) **(i.e. including the effects of hydrothermal circulation, oceanic crust, and temperature-pressure-dependent thermal properties, as well as mineral physics)**, and found a brittle-ductile transition closer to the 700-800°C isotherms, depending on the estimated mantle temperature."

**Line 64-67:** "intracontinental faults, the brittle to ductile transition seems to be … limited by the 300-350 degree C isotherm" – this statement is likely not true. There is plenty of evidence for earthquakes occurring on intracontinental fault zones at depths where the estimated temperatures far exceed 300-350 degrees C [see Jackson et al., 2008; Sloan et al., 2011; Craig et al., 2012; Emmerson et al., 2006].

**Response:**

We agree in this point with the reviewer. Therefore, we removed the details about the bounding isotherms, as they were a result for a particular region studied by Zuza and Cao (2020). The new sentence reads as follows:
"The results from these efforts indicate that in intracontinental faults, the brittle-ductile transition seems to be controlled by variations in the geothermal gradient (Zuza and Cao, 2020)."

**Line 63:** Consider changing "up-scale" to "scale up" and "target" to "determine".

**Response:**
Thank you. We modified the text accordingly.

**Line 67:** Can you cite some examples of where the high geothermal gradient correlates with shallowing seismicity to support your argument here?

**Response:**
One example is the work by Tanaka (2004) -already cited in the manuscript-, who studied the relationship between heat flow, geothermal gradient and D90 depths in Japan. Figure 1 in Tanaka (2004) (see below) shows that as the geothermal gradient increases, the D90 depths become shallow, particularly for geothermal gradients > 100 K/km.

[Figure]

**Line 71-72:** I agree that the 1-dimensional geotherms are a simplification, but these simplifications are made because it is believed that horizontal diffusion or advection of heat plays a minor role in controlling the temperature field in relatively stable tectonic settings. Similarly, a simple layered geometry is often assumed, because the exact nature of the subsurface lithologies, and their material properties (e.g. radiogenic heat production, thermal properties) are not known precisely. The key point is that the unknowns contribute greater uncertainty to the temperature predictions than does ignoring horizontal diffusion of heat, and therefore the 1-D simplification is justified. Equally, a full parameter sweep can be performed with 1-D models, meaning you can quantify uncertainty more easily.

**Response:**
We would like to thank the reviewer for the comment, to which we only agree in part.
While discussing the limitations of considering (multi) one-dimensional thermal
modelling, it is worth to note that these studies not only assume that advection and
lateral diffusion of heat can be neglected, but also the effects of 3D heterogeneities in
the rock properties as driven by an heterogeneous lithospheric configuration (which
can be hardly captured by 1D approximations).
Heat refraction from material contrasts as well as thermal blanketing by sediments are
some examples. Those are indeed important processes for building up lateral and
vertical variations that are genetically linked to the configuration of the plate, including
structural inheritance. These effects can only be captured and described by
considering a 3D model that integrates as close to reality as possible (given the data
availability) structural heterogeneities between first order geological domains
"amalgamated" through time into the present-day lithospheric configuration.

On the second point raised by reviewer#1, we agree that simplified 1D modelling is
computationally less expensive and as such, offers capabilities to ensemble modelling.
Here the reviewer does however disregard recent progress made in the field of model
order reduction modelling, which nowadays enables to run multifidelity ensemble
simulations (for both global and local sensitivity analysis and uncertainty
quantification) without imposing stringent limitation of the model geometry (1D vs 2D
vs 3D) as well as the driving physics (whether thermal diffusion or more complex non-
linear physics). Some of the co-authors have indeed demonstrated how a family of
such surrogate models, based on a reduced basis approximation of the lower order
dimension, can indeed be used for complex 3D geology and non-linear physics (e.g.
Degen et al., 2021; Degen et al., 2022).

**Line 79-80:** "Upscaling the seismogenesis from laboratory experiments…" consider
changing wording to "… scaling up the predicted conditions of seismogenesis from
laboratory experiments to the lithosphere"

**Response:**
Done.

**Line 84:** Is the CSD really "influenced… by the local geothermal gradient"? The local
geothermal gradient is just a proxy for the absolute temperature at depth, which is
most likely the parameter that controls whether faults break in earthquakes or creep
aseismically and therefore the CSD.

**Response:**
To clarify the sentence, we refer now to temperature instead of geothermal gradient:
to "As the CSD is influenced by factors that vary in space, such as lithology and
**temperature**…"

**Line 88:** "The subducting segments of the … slab … in the study area are flat … implying
that the subucting [sic] velocities might be lower than in the steep segments" – is this
true? I would assume that, if the slabs are plate-like and do not deform extensively

internally, then the subduction velocity and the advection of heat beneath the overriding plate should only vary along the length of the subduction zone due to the variations in relative plate motions about the plate's rotation pole. What evidence is there that the subduction velocity is slower in the flat slab segments compared to in the steep slab segments in the Andes?

**Response:**
The reviewer's interpretation would imply that the slab is rigid, which is not expected to be the case. Slower subduction velocities in flat slabs have been noted in the references cited: Currie and Copeland (2022) noted this in a different subduction zone (Farallon plate), and Schellart and Strak (2021) found this in geodynamic simulations (where flat subduction occurs when subduction velocity reaches a minimum).

**Line 91:** "on much longer timescales" not "in much longer timescales".

**Response:**
Done.

**Line 93-95:** I am really struggling to understand what this paragraph means. Maybe consider re-phrasing to: "The novelty of our study is to consider how spatial heterogeneity in the lithology of the lithosphere and mantle temperature influences the temperature distribution and seismicity within the crust"?

**Response:**
Thank you. We added this suggestion and connected this sentence with the paragraph above it.

**Line 96-102:** This statement about seismic hazard is important, but the authors need to be more explicit of exactly how their work can update our understanding of seismic hazard in the region. Specifically, it will provide an estimate of the spatial variation in the thickness of the seismogenic crust and its links with surface observables. The thickness of the seismogenic crust sets the seismogenic area of faults, and therefore the possible maximum magnitude of earthquakes these faults can host.

**Response:**
Thank you for this suggestion. We modified the text as follows:
"In particular, the CSD is a proxy to the maximum depth of seismic ruptures in crustal faults (e.g. Ellis et al., 2023; Zhang et al., 2022), which in turn may limit the rupture areas and the maximum magnitudes of the earthquakes that these faults may host."
The new references cited have been added to the bibliography.

**Line 105:** "results" not "resulted".

**Response:**
Done.

**Line 108:** Spelling error – "steep" not "step".

**Response:**
Done.

**Line 110:** Are there any studies to cite that have looked at the timing of volcanism across the region from absolute dating, and which have argued that the presence of a flat slab led to the termination of volcanism, to support this argument?

**Response:**
Yes, the work by Wagner et al. (2017) focuses on the timing of volcanism and the evolution of the Nazca subduction through time. We misplaced the citation to this paper in the previous version of the manuscript. We corrected the paragraph as follows:
"The present-day flat slab geometry has been established since about 6 Ma, when the Nazca tear developed separating the north (flat) and south (steep) segments.  As a result, the volcanic activity has ceased in the continental crust of the overriding plate of the north segment, which spatially corresponds to our study area **(Wagner et al., 2017)**."

**Line 113:** "remainder" not "remaining".

**Response:**
Done.

**Line 130:** "dominated by plateau and magmatic arc terranes" – as in oceanic plateau rocks? Please clarify what is meant by plateau rocks.

**Response:**
Yes, we meant **oceanic** plateau, and have modified the text accordingly.

**Line 140-144:** A general question about this section that it might be worth trying to address in the text: why is it relevant what geologically-inferred sutures and fault systems run through the study area? I can see why the geological terranes are important, but less so the specific faults.

**Response:**
We had already mentioned (former lines 119-120) that "As a consequence, large-scale sutures (faults) act as major boundaries between these terranes (Kennan and Pindell, 2009)". To further clarify our motivation we have added to that sentence "so they have to be addressed, as they may potentially limit domains with different thermal and/or seismogenic behavior."

**Line 184-186:** "The best model was selected as the one that independently best reproduced the temperatures measured in the boreholes". It seems important to consider all the models that match the borehole temperature data to within the data's uncertainties (±10 degrees?), as opposed to just the one model that best-fits the data,

especially if the differences in data fits are small. The reason I say this is because models with the same near-surface temperatures and mantle temperatures could have very different temperatures in the mid and lower-crust, so just considering a single model might not be reflective of the range of possible temperatures at the depths of earthquakes.

**Response:**
We agree with the main reasoning, that is, models having different properties can fit a single observable (considering the range of uncertainty in the observable measurements). However, we would like to add that the range in the thermal parameters, as discussed in the manuscript, have been chosen based on the insights from the 3D data-integrative geological modelling ( i.e. regional geology, gravity, seismic profiles, etc) and it has been chosen also to be consistent with our knowledge of the tectonic evolution of the study area.  This is reflected in the fact that we ended up with a finite range of variations for each parameter, or in other words, we performed a local sensitivity analysis instead of a global one.

It is also important to note that a more adequate uncertainty characterization of the thermal modelling approach requires an independent effort out of the scope of the current research. As we have been mentioning in previous responses, order reduction modelling is one of the most recent and promising approaches to achieve such systematic sensitivity analyses.

To the main comment by reviewer#1, the range of variations in the thermal properties of the crustal layers could hardly explain having a systematically different thermal configuration, while at the same time fitting the measured temperature (see supplementary Figure S1). This is the main reason for our choice to discuss in detail only the best fitting model, while still discussing the mismatch of all the other members of our analysis as SI materials.

In order to better clarify the approach, we have noted in the text that:
"The model fitting approach followed, for simplicity, a local optimization in which the initial average values of some thermal properties were tuned only if necessary, in order to reproduce with minimum misfit the independent measurements of temperatures in boreholes (as discussed in Sect 4.1)."

**Line 238:** I still think that "Depth of Crustal Earthquakes" is clearer than "Crustal Seismogenic Depth", but I'll leave it up to the authors to decide what they want to use.

**Response:**
There is no agreement in the literature on how this depth should be called. For example, Zhang et al. (2022) refer to the "lower seismogenic depth" in the title but start their introduction as "We investigate crustal seismogenic depths of the western United States [...]".

Since, to be precise, we are mapping D90, we have now mentioned already in the abstract that CSD is "mapped as D90, the 90% percentile of hypocentral depths".

**Line 246-247:** You can remove the citations for Wimpenny et al., 2018 and Wimpenny 2022, as their results are included in the gWFM catalogue that you have already cited.

**Response:**
The only data of the gWFM catalogue within the limits of our study are those provided by Wimpenny et al. (2018) and Wimpenny (2022), so it seems better to keep mentioning these original sources too.

**Line 264:** A better citation than Wimpenny (2022) here would be something like McCaffrey and Abers (1988) or Nabalek (1984), as these were really the papers that demonstrated how waveform-modelling methods provided more accurate estimates of earthquake centroid depths.

**Response:**
Thank you for this recommendation. We replaced the citation to Wimpenny (2022) with McCaffrey and Abers (1988) and Nabalek (1984).

**Line 312:** Change to "… first converted to Mw using the relations…"

**Response:**
Changed.

**Line 313:** I would recommend just leaving this sentence as "using the relations detailed in Text S6." – all the citations are difficult to follow and can be found in the Supplement.

**Response:**
Thank you, we modified the text accordingly. We also removed the references to Di Giacomo et al. (2015) and Salazar et al. (2013), which are no longer cited in the main text and can be found in the supplement.

**Line 409-410:** I would recommend removing the point about whether the crust and lithospheric mantle are coupled or not – I don't see how the lack of seismicity can tell us that.

**Response:**
The reviewer is right in that as it is written the sentence leads to some confusion and misunderstanding in the reader. What we wanted to highlight is that a cold lithosphere is also mechanically compliant, a typical feature of long-lived geological features as cratons.

However, considering also the comment from reviewer#2, we decided to remove the interpretation about crust-mantle coupling.

**Line 469:** Should be "… towards lower hypocentral temperatures in the Falcon Basin"?

**Response:**
Thank you, we modified the text accordingly.

**Line 493:** Should this say "regional average seismogenic depth" as opposed to just "regional seismogenic depth"?

**Response:**
Yes, we added it to the text.

**Line 495-500:** The authors argue that the Murindo earthquake had a hypocentre near the base of the seismogenic layer and that the earthquake ruptured to the surface. Can the authors cite the evidence that this earthquake did in fact rupture to the surface? From what I can tell from the literature, there were earthquake environmental effects, but no recorded primary surface ruptures.

**Response:**
Thanks for this point. We have now softened the wording, indicating that "Its geological effects **suggest** a surface rupture exceeding 100 km in length" (former line 499) and "the mainshock **most likely** ruptured…" (former line 502).

Mosquera-Machado et al. (2009) reviewed that "No evidence of surface faulting has been reported in the literature." but "surface faulting probably occurred". "There was great uncertainty about the exact location and rupture length of the causative faults for the earthquake sequence, because the tropical fluvial setting and difficult access in the epicentral area do [sic] not allow [a] reasonably accurate identification of earthquake fault scarps."

Nevertheless, when listing the ground effects at the Murindó site they indeed described "Two east–west oval sectors on each side of the [Murindó] fault, uplift in the west were sand and ground water were ejected, and subsidence to the east". This evidences coseismic ground deformation at both sides of the fault, so most likely the rupture reached the ground or almost did.

**Line 510:** Maybe change the subtitle to: "Temperature at the base of the seismogenic crust"?

**Response:**
Since we describe the results of depths too, we prefer not to remove this word from the subtitle. We have slightly modified it to "Depths and temperatures of the base of the seismogenic crust".

**Line 515:** The blue polygon in Figure 8b is very difficult to see, at least on my screen. Maybe make it clearer by making the line thicker, or putting a white background behind it?

**Response:**

We have made the blue lines thicker, both in Figure 8 and in Supplementary Figure S8.

**Line 515:** It is not entirely clear what "sheared continental affinity" actually means. Why is the inference that it has been sheared relevant? Is it not just that the terrane is formed from continentally-derived rocks that's relevant?

**Response:**

We complemented the description of the "sheared continental margin" terrane in the study area as follows:

"The collision of the C-LIP with the continental margin of South America defined not only a broad sheared margin **(with remnants of continental slivers and ophiolitic sutures, Kennan and Pindell, 2009)**, but also extended fragments of mafic and ultramafic rocks associated to mantle-plume processes, and emplaced oceanic crust and remnants of island arcs (see Boschman et al., 2014; Kennan and Pindell, 2009; Montes et al., 2019)."

This said, for the interpretation of the results it is important to note the presence of high-density rocks within this sheared continental terrane.

**Figure 8:** There are a lot of references in the text to fault zones that are shown on Figure 3, but not on Figure 8. Because the D90 information is on Figure 8, then it is difficult to follow exactly where the authors are referring to in the text without having both Figure 3 and Figure 8 next to one another. I would recommend either adding the relevant fault zone names, or adding some annotations. They could also remove the fault zones that are not relevant from the line map to focus the readers eye on the relevant information. These are just some suggestions.

**Response:**
We agree that the display is not ideal, but adding names or annotations to Fig. 8 would clutter it excessively, because of its reduced size. We expect that the interested reader will be able to familiarize herself or himself with the names annotated in earlier figures (1 and 3).

**Line 512-515:** Here the authors say that there is "... a transition from shallow D90 depths and cold temperature associated to [sic] oceanic terranes and island arc affinity in Western South America, towards deeper and hotter values in terranes with a more sheared continental affinity in the east." – The authors need to clarify exactly which part of their study region "west" and "east" are referring to here. The trends they point out are subtle and only hold in certain parts of the map area. For example, if you look at a west-east traverse for two profiles extracted from Figure 8a (see Figure R1 below), for the northern most profile the D90 gets deeper beneath the Middle Magdalena Basin compared to the Western Cordillera. However, for the profile further south the D90 is deeper beneath the Western Cordillera than it is beneath the Middle Magdalena Basin (MMB).

**Response:**

This trend should now be clearer to spot in the figures, thanks to the improvement of the visibility of the terrane contours in figures 8 and S8. The location mentioned by the reviewer seems actually the only, small, exception to the general trend.

To further clarify the text, we have rewritten slightly former lines 513-515, specifying that "Our results suggest a trend from shallower D90 depths and colder temperatures in the Greater Panama terrane (oceanic, with island arc affinity) towards deeper and hotter values in the sheared continental margin (Fig 1b and blue polygons in Fig. 8b)."

**Line 520-522:** Can I suggest re-phrasing these sentences to: "We interpret the observed variability in D90 between the Central and Eastern Cordilleras, and the Middle Magdalena Basin, to suggest there is significant rheological contrasts between these areas. These major terranes are likely separated by crustal-scale faults." At the moment, I find it hard to understand what these sentences are trying to say.

**Response:**
Thank you. The new paragraph says:

"We interpret that the observed variability in D90 between the Central and Eastern Cordilleras and the Middle Magdalena Basin (MMB, Fig. 1b) evidences significant rheological contrasts between these areas. These major terranes are likely separated by crustal-scale faults (Kennan and Pindell, 2009)."

**Line 547:** Spelling error? Should say "CSD" not "SCD".

**Response:**
Yes, it was a typo. Thank you. We fixed it.

**Line 628:** Why can a hot upper mantle explain why there are earthquakes at high temperatures? The hot upper mantle just explains why the temperature is high, not necessarily why there is seismicity in the rocks at these high temperatures. For example, Iceland has a hot upper mantle, has a predominantly mafic crust because it has formed from MOR volcanism, but there is very little seismicity deeper than 10-15 km (if any).

**Response:**
Thanks for pointing this out. Our reasoning was that, having a mafic lower crust, seismicity could occur at higher temperatures as they have a deeper brittle-ductile transition. In this case, the high temperatures are supported by a relatively hot upper mantle. Of course, "hot" here does not mean as hot as Iceland (which is a pretty unique case).

Nevertheless, we decided to remove the "hot upper mantle" sentence from the paragraph, aiming to avoid misunderstandings.

**Reply to comment by Reviewer #2**

'The crustal earthquakes occur at locations with a mean geothermal gradient of 19.4±1.23 °C/km-1, preferentially clustering in specific zones, e.g. in the North Andes block and the Panama microplate. Seismicity is almost absent in cold lithospheric areas such as the Guyana craton and the Caribbean Large Igneous Plateau. Such correlation indirectly suggests that in these places, the crust and lithospheric mantle may be strongly coupled, and therefore, the differential stress is not high enough to deform the crust in a brittle regime.'

First, better say 19.4±1.2 instead of 19.4±1.23. Second, following the Byerlee law, the brittle strength does not depend on temperature and therefore the same stress is needed to produce brittle failure and earthquakes. Instead I find a more logic explanation that the cold undeformed areas (e.g. the Guyana craton) may have less inhereted structural weakeness (faults) and are therefore more difficult to localize strain. Also compositional differences can have an effect. For example cratons are usually to be compositionally strengthened (via loss of volatiles).
Overall I would suggest 'relaxing' their interpretation, or adding alternative explanation.

**Response:**
Thank you very much for your comment. We have modified the paragraph following your recommendation:

"The crustal earthquakes occur at locations with a mean geothermal gradient of 19.4±1.2 °C/km-1, preferentially clustering in specific zones, e.g. in the North Andes block and the Panama microplate. Seismicity is almost absent in cold lithospheric areas such as the Guyana craton and the Caribbean Large Igneous Plateau. This again, is an indication that a 1D geotherm approximation will not be robust enough to model the thermal configuration of the heterogeneous study area."

**References not cited in the manuscript**

Degen, D., Cacace, M. & Wellmann, F. 3D multi-physics uncertainty quantification using physics-based machine learning. Sci Rep 12, 17491 (2022). https://doi.org/10.1038/s41598-022-21739-7

Degen, D., Veroy, K., Scheck-Wenderoth, M. et al. Crustal-scale thermal models: revisiting the influence of deep boundary conditions. Environ Earth Sci 81, 88 (2022). https://doi.org/10.1007/s12665-022-10202-5